# 11-year record of wintertime snow surface energy balance and sublimation at 4863 m a.s.l. on Chhota Shigri Glacier moraine (western Himalaya, India)

Arindan Mandal[1], Thupstan Angchuk[1,2], Mohd Farooq Azam[3], Alagappan Ramanathan[1], Patrick Wagnon[4], Mohd Soheb[1], Chetan Singh[1]

[1]School of Environmental Sciences, Jawaharlal Nehru University, New Delhi 110067, India
[2]DST's Centre of Excellence, Department of Geology, Sikkim University, Gangtok 737102, India
[3]Department of Civil Engineering, Indian Institute of Technology Indore, Simrol 453552, India
[4]Université Grenoble Alpes, CNRS, IRD, IGE, F-38000 Grenoble, France

*Correspondence to*: Arindan Mandal (arindan.141@gmail.com)

**Abstract.** Analysis of surface energy balance (SEB) at the glacier/snow surface is the most comprehensive way to explain the atmosphere-glacier/snow interactions but that requires extensive data. In this study, we have analysed an 11-year (2009-2020) record of the meteorological dataset from an automatic weather station installed at 4863 m a.s.l. on a lateral moraine of the Chhota Shigri Glacier, western Himalaya. The study was carried out over the winter months (December to April) to understand SEB drivers and snow loses through sublimation. Furthermore, this study examines the role of cloud cover on SEB and turbulent heat fluxes. The turbulent heat fluxes were calculated using the bulk-aerodynamic method, including stability corrections. The net short-wave radiation was the primary energy source. However, the turbulent heat fluxes dissipated a significant amount of energy. The cloud cover plays an important role in limiting the incoming short-wave radiation by about 70%. It also restricts the turbulent heat fluxes by larger than 60%, resulting in lower snow sublimation. During winter, turbulent latent heat flux contributed the largest proportion (64%) in the total SEB, followed by net radiation (25%) and sensible heat flux (11%). Sublimation rates were three times higher in clear-sky conditions than overcast, indicating a strong role of cloud cover in shaping favourable conditions for turbulent latent heat flux by modulating the near-surface boundary layer conditions. Dry air, along with high snow surface temperature and wind speed favours sublimation. Besides, we also observed that strong and cold winds, possibly through mid-latitude western disturbances, impede sublimation by bringing high moisture content to the region and cooling the snow surface. The estimated snow sublimation fraction was 16-42% of the total winter snowfall at the study site. This study substantiates that the snow sublimation is an essential variable to be considered in glaciohydrological modelling at the high mountain Himalayan glacierised catchments.

## 1 Introduction

The widespread global glacier imbalance (Slater et al., 2020; Zemp et al., 2019; IPCC, 2019) is a manifestation of ablation dominance compared to accumulation over the last few decades. Ablation processes —including surface melting, sublimation, evaporation, and wind-driven transport/erosion— lead to the loss of snow and ice mass (Bintanja, 1995; Nicholson et al., 2013; Giesen and Andreassen, 2009; Schaefer et al., 2020; Van den Broeke et al., 2005; Oerlemans, 2000; Conway and Cullen, 2016). Among these, sublimation from snow and ice surfaces is one of the significant contributors to the total ablation (Stigter et al., 2018; Huintjes et al., 2015a) yet are seldom quantified, especially in the Himalaya-Karakoram (HK) region (Azam et al., 2021). Sublimation can be calculated from the surface energy balance (SEB), which requires several meteorological inputs to describe the physical relationship between the glacier/snow surface and meteorological variables (Oerlemans, 2001).

SEB studies are rare in the HK region due to the extreme terrain and the lack of high-altitude meteorological data from glacier and snow-covered sites. SEB studies have been conducted on nearly eleven glacier/snow-covered sites across the HK region (see supplementary material; Table S1 and Fig. S1). However, SEB studies on Tibetan glaciers are relatively more abundant (~17 investigated glaciers/ice-covered sites; Table S1), including direct turbulent heat flux measurements (Yang et al., 2011; Zhu et al., 2018) except in Pamir and Kunlun Mountains (Zhu et al., 2020). Glaciers in the Pamir and Kunlun Mountains are extreme continental type, with cold temperature and low annual precipitation (Zhu et al., 2020; Li et al., 2019), thus their SEB characteristics are expected to behave differently than majority of HK glaciers which are alpine type, with relatively higher precipitation and temperature. In the HK region, a few SEB experiments have been carried out recently, most of them being in the central Himalaya in Nepal, yet at a smaller temporal range, from a month to a few seasons/years (Rounce et al., 2015; Steiner et al., 2018; Acharya and Kayastha, 2019; Litt et al., 2019; Matthews et al., 2020; Steiner et al., 2021). SEB studies in the Indian Himalaya are few. Only a single on-glacier SEB experiment was conducted at the Chhota Shigri Glacier in the western Himalaya (Azam et al., 2014a). Recently, Singh et al. (2020) conducted a SEB experiment on a moraine surface with ephemeral snow cover near the Pindari Glacier in Uttarakhand using two-year data from a weather station. Glacier-wide applications of SEB remain rare to date in the HK region (Srivastava and Azam, 2022).

Apart from the limited number of SEB sampled sites in the HK region, the available literature has mostly focused on the radiative or net radiation fluxes. Net radiation plays a greater role in supplying melt energy to snow/ice than turbulent heat fluxes (Smith et al., 2020). Turbulent fluxes can contribute about 20% of SEB globally and sometimes above 70% for a shorter timescale (Thibert et al., 2018). The higher contribution of turbulent heat flux is common in the high-latitudinal glaciers having low altitude, where snow-ice surfaces are exposed to higher air temperatures and dry conditions. The contribution of turbulent heat fluxes on some of the Tibetan and Nepalese glaciers/snow-covered sites are also higher, being well larger than 20%, e.g., Chongche Ice Cap in the Kunlun Mountains, South Col of the Everest (Table S1). The SEB experiment on the Everest summit shows that a decrease in turbulent heat flux boosts short-wave radiation efficiency, which results in surface melting despite air temperatures being below freezing point (Matthews et al., 2020). Overall, the turbulent heat fluxes and their involvement in SEB of the HK glaciers are rarely studied and thus, poorly understood.

Snow sublimation is expected to be a significant component of the glacier surface mass balance in the HK region (Azam et al., 2021). Stigter et al. (2018) showed that sublimation loss on the central Himalayan Yala Glacier in Nepal is larger than 20% of winter snowfall. Srivastava and Azam (2022) studied the glacier-wide SEB on the Chhota Shigri and Dokriani glaciers in the Indian Himalaya and estimated a mass loss through sublimation up to 20% of the total annual ablation, with strong spatial and temporal variability. Sublimation contribution is observed to be up to 66% of the total mass loss on the Purogangri ice cap of the north-central Tibetan Plateau (Huintjes et al., 2015b). In the Muji Glacier in northeast Pamir, cold season's evaposublimation loss is > 70% of the corresponding snowfall (Zhu et al., 2020). In the Qilian Mountains at the August-One Glacier in north-east Tibetan Plateau, evaposublimation loss is lower but accounts for about 15% of annual precipitation (Guo et al., 2021). Recently, Gascoin (2021) reported that the basin-wide mean snow sublimation is ~11% of the total snow ablation in the Indus Basin, with more than 60% in Ladakh and western Tibet areas based on satellite-derived datasets (HMASR v1). The HK region's high-altitude meteorological conditions, such as high wind, low atmospheric pressure, and dry air are expected to support sublimation (Wagnon et al., 2013; Shea et al., 2015; Mandal et al., 2020; Matthews et al., 2020; Azam et al., 2018). Therefore, the quantification of high-altitude sublimation is important to improve our understanding of the glacier mass balance components in the HK region.

Direct sublimation measurement requires the use of an eddy covariance system or pan sublimation technique. The eddy covariance system is advanced and precise (Sexstone et al., 2016) but expensive, hence it has been used only in two sites in the HK region: Yala and Lirung glaciers in Nepal (Stigter et al., 2018; Steiner et al., 2018). The pan sublimation or lysimeter measurements are rare in the HK region, likely due to inaccessibility and harsh weather conditions. Alternatively, the bulk-aerodynamic method is widely used for calculating turbulent heat fluxes and thus sublimation. On the Yala Glacier, Stigter et al. (2018) evaluated multiple methods (e.g., bulk-aerodynamic, the Penman-Monteith equation and an empirical relation) with eddy covariance-based sublimation. Results obtained show that the bulk method estimate is similar to observed eddy covariance-based sublimation. However, parameterisation of the bulk-exchange coefficient and surface roughness length is critical for precisely modelling the turbulent heat fluxes (Smith et al., 2020; Stigter et al., 2018).

This research presents an 11-year long SEB study on the snow-covered side moraine of the Chhota Shigri Glacier in the western Himalaya using an off-glacier automatic weather station (hereafter AWS-M) installed at 4863 m a.s.l. The AWS-M records round the year data, but for this study, we considered the snow-covered period between December and April of each hydrological year over 2009-2020. Our primary focus here is to better understand the turbulent heat fluxes and their role in SEB during the winter season when the atmospheric conditions are windier and drier. We also attempt to quantify the snow sublimation and its meteorological drivers. Special attention is given to identify the role of cloud cover on the SEB components and sublimation. We also estimated the fraction of snow sublimation to the winter snowfall at the AWS-M site.

## 2 Study area and AWS

### 2.1 Chhota Shigri moraine site and AWS description

Chhota Shigri Glacier is located in the Chandra Basin (sub-basin of the Indus) of the Lahaul-Spiti Valley situated in the western Himalaya (Fig. 1). The Chandra Basin (~30% glacierised) is located in the monsoon-arid transition zone and is influenced by the Indian Summer Monsoon (ISM) during summer and the Western Disturbances (WDs) during winter (Bookhagen and Burbank, 2010). The mean annual precipitation at the Chhota Shigri base camp was 922 mm, of which 67% was during the winter season (November-April) and the remaining 33% during the summer-monsoon (May-October) (Mandal et al., 2020).

Chhota Shigri is among the most-studied glaciers in the HK region in terms of surface mass balance and glacial processes. The mean annual glacier-wide mass balance was -0.46 ± 0.40 m w.e. $a^{-1}$ (water equivalent) over 2002-2019 (Mandal et al., 2020). Azam et al. (2014a) carried out a SEB experiment on this glacier using an on-glacier AWS (hereafter AWS-G; Fig. 1) during 2012-2013 but could not conduct a full-year SEB analysis due to AWS-G failure in winter. They estimated that the net radiation ($R_{net}$) was the primary energy source with about 80% energy flux to SEB, while the turbulent and conductive heat fluxes shared

the rest of the total energy flux.

For this study, the meteorological data were collected on the side moraine of the Chhota Shigri Glacier using the AWS-M (32.23° N, 77.51° E) installed at 4863 m a.s.l. (Fig. 1). The AWS-M has been positioned ~50 m away from the Chhota Shigri Glacier margin and on a relatively flat hill-top site. The surface at the AWS-M site remains snow-covered during winter and sand/sediment exposed during summer (Fig. 1). The AWS-M has been operating since October 2009. Air temperature ($T_{air}$),

surface temperature ($T_s$), relative humidity ($RH$), wind speed ($u$) and direction ($WD$), incoming and outgoing short-wave ($S_{in}$ and $S_{out}$) and long-wave ($L_{in}$ and $L_{out}$) radiations were being recorded at a frequency of 30 seconds and stored as half-hourly averages by a Campbell CR1000 data logger. Data before 23 May 2010 was recorded at an hourly time-step. Precipitation was recorded at the base camp at 3850 m a.s.l. using a Geonor-T200B sensor since July 2012. Description and specifications of the sensors for AWS-M and Geonor gauge are provided in Table 1.

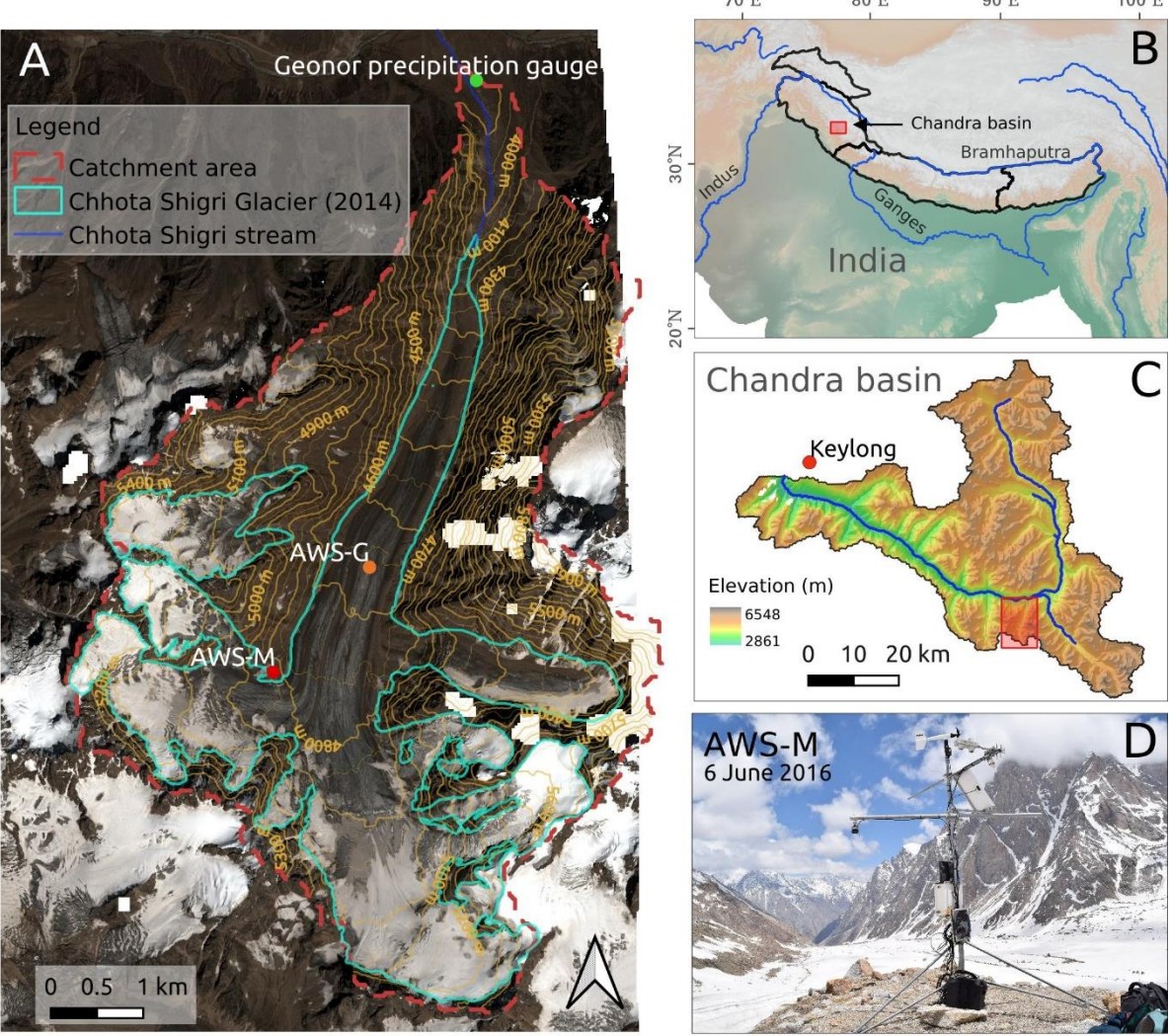

**Figure 1. (A)** Chhota Shigri Glacier catchment showing the location of the AWS-M (red dot), AWS-G (orange dot; middle ablation zone) and Geonor T-200B automatic precipitation gauge (green dot). Glacier outline was derived using the 2014 Pléiades image (Azam et al., 2016). The background is the Pléiades ortho-image of 12 September 2020 (copyright CNES 2020, distribution Airbus D&S). **(B)** Location of the Chhota Shigri Glacier region in the western

Himalaya. **(C)** Map of the Chandra Basin, with Chhota Shigri catchment marked (red rectangle). Elevation based on the Shuttle Radar Topographic Mission (SRTM) Digital Elevation Model (DEM) obtained from the United States Geological Survey (USGS). **(D)** Photo of the AWS-M on the lateral moraine (Photo credit: A Mandal).

**Table 1. Sensor details of the AWS-M (4863 m a.s.l.) and Geonor precipitation gauge at the base camp (3850 m a.s.l.)**

**of the Chhota Shigri Glacier. Variable symbols are also given. Sensor heights indicate the distances to the surface without snow. Long-wave radiation sensors have been operational since May 2010. The snow depth sensor was operational until October 2015.**

| Meteorological variable | Symbol (unit) | Sensor | Height (m) | Accuracy |
|---|---|---|---|---|
| **AWS-M** | | | | |
| Air temperature | $T_{air}$ (°C) | Campbell H3-S3-XT | 1.5 | ±0.1 at 0 °C |
| Surface temperature | $T_s$ (°C) | Apogee SI-111[a] | 2.5 | ±0.2 (-10 to +65 °C) ±0.5 (-40 to +70 °C) |
| Relative humidity | $RH$ (%) | Campbell H3-S3-XT | 1.5 | ±1.5% RH at 23 °C |
| Wind speed | $u$ (m s⁻¹) | Campbell 05103-10-L | 3 | ±0.3 m s⁻¹ |
| Wind direction | $WD$ (degree) | Campbell 05103-10-L | 3 | ±3 degree |
| Incoming and outgoing short-wave radiations | $S_{in}$, $S_{out}$ (W m⁻²) | Kipp & Zonen CNR-1 | 2.5 | ±10% day total |
| Incoming and outgoing long-wave radiations | $L_{in}$, $L_{out}$ (W m⁻²) | Kipp & Zonen CNR-1 | 2.5 | ±10% day total |
| Snow depth | SR50A (m) | Campbell SR50A | 2 | ±0.01 m or 0.4% to target |
| **Precipitation** | $P$ (mm) | Geonor T-200B | 1.7[b] | ±0.6 mm |

[a]Infrared radiometer; [b]Inlet height

## 3 Datasets and methodology

### 3.1 Meteorological data and gaps

The meteorological data from the AWS-M has been used between 1 December and 30 April (DJFMA) of each hydrological year for 2009-2020. We filtered the snow-covered period for SEB based on the daytime surface albedo threshold value above 0.4 at the AWS-M (the mean bare-ground/snow-free surface albedo was less than 0.25 for July-August; 2009-2020). Additionally, we discarded the data of 74 days (2975 data points) out of a total of 1664 days (76248 data points; DJFMA 2009-2020) when albedo was below 0.4 (refer to Table S2 for snow-free dates). This albedo threshold value is similar to the minimum albedo (0.41 to 0.46) of continuous snow cover at the Ganja La and Yala sites in Nepal (Stigter et al., 2021; Kirkham et al., 2019).

There was a gap in observation of all variables in the AWS-M data during the night (18:00 to 06:00 Indian Standard Time; IST) between 22 February 2015 and 2 October 2016 (220 days of DJFMA) due to a disconnected wire between the solar panel and the battery. These gaps were filled using the mean value of the respective variables from available records (1 December 2009 – 21 February 2015 and 1 December 2016 – 30 April 2020) of the AWS-M for the particular time-steps on the same day. To identify the reliability of the gap-filling method, we applied the same method for non-missing year data by removing the night values (18:00 - 06:00 IST) and filled them with mean values from other years. The root mean square error (RMSE) and mean absolute error (MAE) between the original (with night values) and the filled dataset was found to be 3.3°C and 2.6°C for $T_{air}$, 4.1°C and 3.3°C for $T_s$, 27% and 22% for $RH$, and 2.7 m s⁻¹ and 2.1 m s⁻¹ for $u$, respectively for the test year, 2017/18.

Precipitation data was used from the single-Alter-shielded Geonor gauge operated at the glacier base camp at ~3850 m a.s.l. since July 2012 (Fig. 1). All-weather precipitation gauges are known to undercatch precipitation in case of snow (Kochendorfer

et al., 2017), and since our precipitation measurements have not been corrected yet, we suspect that precipitation magnitude is underestimated during the snow season (i.e., winter, spring). But those values have only been used to compare with cumulative sublimation in corresponding years, and this does not impact our results. Geonor gauge has a data gap between October 2013 and July 2014 due to battery failure. Therefore, for the gap period, we used monthly precipitation records from the nearest Indian Meteorological Department's (IMD) Keylong station which is located at 3119 m a.s.l. ([https://weathershimla.nic.in/en-IN/climatedata.html](https://weathershimla.nic.in/en-IN/climatedata.html), last access: 15 November 2021). Precipitation data from the Keylong station is used because it is the only existing observatory close to the study area (~60 km from the AWS-M site; Fig. 1). Geonor and Keylong precipitation gauges cannot differentiate between snow and rain. Since the daily and monthly $T_{air}$ did not rise above 0°C during DJFMA (Fig. 2; Table S3), we considered DJFMA precipitation as snowfall at both sites. Moreover, the AWS-M site is located 1013 m higher than the Geonor gauge altitude. The measured precipitation of Geonor and Keylong were well correlated ($r^2 = 0.82$); however, the RMSE was higher: 274 mm (Fig. S4). Therefore, we applied a precipitation gradient of 0.1 m km$^{-1}$ following Azam et al. (2014b) to extrapolate Keylong's precipitation to the AWS-M altitude (RMSE reduced to 139 mm). For this study, in-situ precipitation data from Geonor gauge is available for only five hydrological years (2012-2018; discontinuous).

### 3.2 Surface energy balance (SEB)

SEB has been calculated at a point location for the skin layer using the AWS-M data at a half-hourly time-step between 1 December and 30 April (~151 days) of each hydrological year over 2009-2020 (hourly time-step for 2009/10). The SEB at the snow surface can be written as (Van den Broeke et al., 2005; Hock, 2005; Oke, 1987):

$$F_{surface} = S_{in} + S_{out} + L_{in} + L_{out} + H + +G + P, \tag{1}$$

where $F_{surface}$ [W m$^{-2}$] is the net energy balance of all energy fluxes at the snow surface, $S_{in}$ and $S_{out}$ are the incoming and outgoing short-wave radiation, $L_{in}$ and $L_{out}$ are the incoming and outgoing long-wave radiation, $H$ and $LE$ are the sensible and latent turbulent heat fluxes, $G$ and $P$ are the conductive heat flux and heat advected by precipitation, respectively.

Compared to other fluxes, $P$ on glacier/snow is negligible (Hock, 2005; Kayastha et al., 1999) therefore neglected here. $G$ was found to be negligible or close to $0.0 \pm 1.0$ W m$^{-2}$ at the on-glacier AWS-G site on the Chhota Shigri Glacier during winter 2012/13 (Azam et al., 2014a), thus neglected in the present study. Also, $G$ was neglected in SEB of transient snow cover at the Ganja La and Yala sites, considering inadequate measurement and information of $G$ in the HK region (Stigter et al., 2021). All fluxes are expressed in W m$^{-2}$ and defined as positive when directed towards the surface and negative when away from the surface. When $F_{surface}$ is larger than 0 W m$^{-2}$, it will get directed towards the surface/snowpack and warm it up until it reaches the melting point ($T_s = 0$°C), and then surplus $F_{surface}$ will cause melting (Hock, 2005).

### 3.2.1 Radiative fluxes

$S_{net}$ [W m$^{-2}$] and $L_{net}$ [W m$^{-2}$] are represented as $S_{in}$ - $S_{out}$ and $L_{in}$ - $L_{out}$, respectively and all together can be expressed as net radiation, $R_{net} = S_{net} + L_{net}$. However, several corrections were applied to $S_{in}$ and $S_{out}$ datasets before using them for SEB. All

the night values (determined based on the solar elevation angle) of $S_{in}$ and $S_{out}$ were set to be zero. The measured $S_{out}$ was higher than $S_{in}$ (1.6 % of total data) during the morning and evening, mainly due to the low solar angle because of poor cosine response of the upward-looking pyranometer ($S_{in}$) or due to covering up of the pyranometer by snowfall (Nicholson et al., 2013; Favier et al., 2004). In such cases, $S_{in}$ was corrected using $S_{out}$ (raw) and accumulated albedo ($\alpha_{acc}$) (Van den Broeke et al., 2004). $\alpha_{acc}$ is the 24-hour sum of $S_{out}$ divided by the sum of $S_{in}$ centred around the moment of observation and calculated following Van den Broeke et al. (2004):

$$\alpha_{acc} = \frac{\sum_{24} S_{out}}{\sum_{24} S_{in}}, \qquad (2)$$

$L_{net}$ was calculated from the difference between observed $L_{in}$ and $L_{out}$. We used raw data from up and down pyrgeometers (CG3) of the radiation sensor (CNR-1) to compute the final $L_{in}$ and $L_{out}$ at the AWS-M site.

### 3.2.2 Turbulent energy flux

The vertical turbulent heat fluxes, $H$ and $LE$, are calculated using the bulk-aerodynamic method, including stability correction (Brutsaert, 1982). This method is widely used for its applicability because it allows estimating $H$ and $LE$ from one level of measurement (Chambers et al., 2020; Radić et al., 2017). The bulk-aerodynamic method has already been applied on this glacier at the AWS-G site (on-glacier; Fig. 1) to conduct a SEB experiment during 2012/13, where the SEB-derived ablation showed a good agreement with stake ablation (Azam et al., 2014a). Further, the bulk method showed a good agreement with the eddy covariance observations over a snow-covered central Himalayan glacier (Stigter et al., 2018). In addition, Denby and Greuell (2000) showed that the bulk-aerodynamic method gives reasonable results in high wind speeds, even in katabatic wind conditions. Therefore, the bulk-aerodynamic method is applied in the present study as it has already been applied in this glacier and several other studies in the HK region, where atmospheric conditions are similar with high winds (Litt et al., 2019; Stigter et al., 2021; Guo et al., 2022; Azam et al., 2014a). The bulk Richardson number, $R_{ib}$, describes the stability of the surface layer (Eq. 3), which relates the relative effects of buoyancy to mechanical forces (e.g., Brutsaert, 1982). Therefore, the stability effects were accounted based on $R_{ib}$:

$$R_{ib} = g \frac{\frac{(T_{air} - T_s)}{(z_t - z_{0t})}}{T_{air} \left(\frac{u}{z_u - z_{0m}}\right)^2}, \qquad (3)$$

where $g$ is the acceleration due to gravity [$g = 9.81$ m s$^{-2}$]; $T_{air}$ and $u$ are the air temperature [K] and horizontal wind speed [m s$^{-1}$] at the measurement height, respectively; $T_s$ is the surface temperature [K]. $z_u$ and $z_t$ are the measurement heights [m] for wind speed and air temperature, respectively. $z_{0m}$, $z_{0t}$ and $z_{0q}$ are the surface roughness lengths [m] for momentum, temperature, and humidity, respectively. $R_{ib}$ is positive in a stable atmosphere. Assuming that local gradients of mean horizontal wind speed, temperature and specific humidity are equal to the finite differences between the measurement height and the surface, the turbulent fluxes, $H$ and $LE$ are (Brutsaert, 1982):

$$H = \rho \frac{C_P k^2 u (T_{air} - T_s)}{ln\left(\frac{z_u}{z_{0m}}\right) ln\left(\frac{z_t}{z_{0t}}\right)} (\Phi_m \Phi_h)^{-1}, \tag{4}$$

$$LE = \rho \frac{L_s k^2 u (q - q_s)}{ln\left(\frac{z_u}{z_{0m}}\right) ln\left(\frac{z_t}{z_{0q}}\right)} (\Phi_m \Phi_v)^{-1}, \tag{5}$$

where $\rho$ is the air density at 4863 m a.s.l. [kg m$^{-3}$] calculated as $\rho = \rho_0 \frac{p_{air}}{p_0}$ where $\rho_0$ is the density [kg m$^{-3}$] at standard sea level pressure $p_0$ [1013.25 hPa] and $p_{air}$ is atmospheric pressure [hPa] measured at the site (Cuffey and Paterson, 2010). $C_P$ is the specific heat capacity of air [J kg$^{-1}$ K$^{-1}$] ($C_p = C_{pd}(1+0.84q)$ with $C_{pd} = 1005$ J kg$^{-1}$ K$^{-1}$, the specific heat capacity for dry air at constant pressure), $L_s$ is the latent heat of sublimation for $T_s < 0$°C ($2.849 \times 10^6$ J kg$^{-1}$), $q$ and $q_s$ [kg kg$^{-1}$] are the specific humidity at height $z$ and surface, respectively. $q$ and $q_s$ were calculated using the measured $T_{air}$, $T_s$ and $RH$. $\Phi_{m/h/v}$ are the non-dimensional stability functions for momentum, heat, and vapor/moisture, respectively. The stability functions are given by Brutsaert (1982) and previously applied in several glacier SEB studies (e.g., Reid and Brock, 2010; Conway et al., 2022) and on the Chhota Shigri Glacier (Azam et al., 2014a). $\Phi_{m/h/v}$ expressed in terms of $R_{ib}$:

For $R_{ib} > 0$ (stable case):

$$\left(\Phi_m \Phi_{h/v}\right)^{-1} = (1 - 5R_{ib})^2, \tag{6}$$

For $R_{ib} < 0$ (unstable case):

$$\left(\Phi_m \Phi_{h/v}\right)^{-1} = (1 - 16R_{ib})^{0.75}, \tag{7}$$

Half-hourly data of $u$, $T_{air}$, $T_s$ and $RH$ were used to apply the bulk-aerodynamic method when the AWS-M surface was snow-covered ($\alpha_{acc} > 0.4$). $T_s$ was directly used from the measurement by an infrared radiometer (Table 1). The correlation between infrared measured $T_s$ and $T_s$ derived from $L_{out}$ (using Stefan-Boltzmann equation for the snow surface with emissivity of 1 following Hock and Holmgren, 2005) was $r^2 = 0.99$ ($p < 0.001$) with RMSE = 0.23°C. The lower and upper limits of $R_{ib}$ were fixed at -0.40 and 0.23, respectively, beyond which all turbulence is suppressed (Denby and Greuell, 2000; Favier et al., 2011). In this way, we discarded about 11% of the data points beyond the $R_{ib}$ range.

The aerodynamic ($z_{0m}$) and scalar surface roughness lengths ($z_{0t}$) play a pivotal role in the bulk method as the turbulent fluxes are very sensitive to the values of these surface roughness lengths (Chambers et al., 2020; Smith et al., 2020; Nicholson and Stiperski, 2020; Wagnon et al., 1999). Therefore, in this study $z_{0m}$ for snow surface is taken as 0.001 m which was calculated for the AWS-G site between 16 September 2012 and 17 January 2013 when the AWS-G surface was snow-covered (Azam et al., 2014a). This value was calculated using wind measurements at two different levels following a conventional logarithmic profile (e.g., Moore, 1983). Similarly, $z_{0t}$ and $z_{0q}$ for snow surface are considered as 0.001 m following Azam et al. (2014a).

Due to the limitations in the data availability, direct validation of the bulk model used in this study was not possible, therefore, our results are based on Azam et al (2014a)'s bulk model validation done on this glacier in 2012/13 and it proved to deliver

robust results compared to observations. We also conducted a sensitivity analysis of our bulk model including surface roughness lengths (Sect. 5.2).

Sublimation ($S$) was estimated for every DJFMA period between 2009 and 2020 (excluded days are listed in Table S2). $S$ [$10^{-3}$ kg m$^{-2}$ or mm w.e.] was calculated at a half-hourly time-step (hourly time-step for 2009/10) from negative $LE,$ according to:

$$S = \frac{dt}{L_s},$$ (8)

where $L_s$ denotes latent heat of sublimation and $dt$ is the time-step [seconds].

### 3.3 Cloud factor

Cloud cover is a good indicator of the contribution of radiation to the surface (Favier et al., 2004). In this study, the cloud factor ($CF$) is calculated at the AWS-M site between 09:00 and 16:00 IST to avoid the steep valley wall's shading effect during morning and evening. $CF$ is calculated by comparing short-wave incoming ($S_{in}$) with the short-wave radiation at the top of the atmosphere ($S_{TOA}$) following Favier et al. (2004):

$$CF = 1.3 - 1.4 \left( \frac{S_{in}}{S_{TOA}} \right),$$ (9)

which represents a quantitative cloud cover estimate and ranges from 0 to 1. The values 1.3 (offset) and 1.4 (scale factor) were derived from a simple linear optimisation process (Favier et al., 2004). $S_{in}$ was used from the direct measurement from the AWS-M, whereas the theoretical value of $S_{TOA}$ for a horizontal surface is calculated following Iqbal (1983).

### 3.4 Statistical analysis

The standard correlation coefficient ($r$) and coefficient of determination ($r^2$) were estimated to assess the relationship between various meteorological variables, SEB, and sublimation. The two-tailed Student $t$-test was used to measure the significance of the $r$ and $r^2$. RMSE is calculated to identify the bias/deviation. The K-fold cross-validation method was applied for linear and multiple regression analysis, performed using the 'caret' package (Kuhn, 2021) of the R environment (R Core Team, 2021). Cross-validation is a machine learning technique that is used to protect the predictive model against overfitting for better accuracy. We used this method to estimate the meteorological variable's variance in sublimation.

### 4 Results

### 4.1 DJFMA meteorological characteristics

The range of the meteorological variables measured at the AWS-M for DJFMA (2009-2020) is given in Table S3 to provide an overview of the prevailing weather conditions in the study region. During DJFMA, the mean monthly $T_{air}$ ranged from -15.5°C in January to -6.9°C in April, with a mean of -12.1°C (Fig. 2). Daily mean $T_{air}$ was below 0°C except during late April in 2010/11 and 2016/17 when daily $T_{air}$ slightly exceeded 0°C (Fig. S2). The highest daily $T_{air}$ was 0.1°C on 27 April 2011 and

the lowest was -21.9°C on 26 January 2019. The mean monthly $T_s$ ranged from -17.7°C in January to -7.5°C in April, with a mean of -13.7°C. Daily mean $T_s$ was below 0ºC across DJFMA. However, half-hourly $T_s$ was higher than $T_{air}$ for about 45% of the data points.

The mean monthly $RH$ ranged from 31% in January to 49% in April, with a mean of 43% (Fig. 2). But for a few days, the mean daily $RH$ in DJFMA was higher than 60%. The mean daily $RH$ was below 30% (assumed as dry air) for 29% of days and above 60% (humid air) for 24% of days during the study period.

The mean monthly $u$ ranged from 3.7 m s$^{-1}$ in April to 6.0 m s$^{-1}$ in February, with a mean of 5.0 m s$^{-1}$ during DJFMA (Fig. 2). Based on half-hourly records, $u < 5.0$ m s$^{-1}$ occurred for 56% of the data points during the study period, while $u > 10.0$ m s$^{-1}$ were observed for only 7%. The half-hourly mean $u$ reached up to 24.2 m s$^{-1}$ on 21 February 2019. The highest recorded mean daily $u$ was 15.9 m s$^{-1}$ on 20 March 2012. The windrose shows that there is a persistent down-valley wind along the glacier flowline coming from the south-east (90°-135°) during DJFMA with high wind speed (Fig. 3).

Precipitation records from the Geonor gauge were available only for five complete DJFMA periods but discontinuous (Fig. 4). During DJFMA, most of the precipitation in the Chhota Shigri catchment falls due to the WDs storms, accounting for about 67% of its annual total of ~900 mm (Mandal et al., 2020). The total mean precipitation during DJFMA was 659 mm (2012-2018; Table S3). March received the highest, with 150 mm corresponding to 26% of total winter precipitation and least in December, with 56 mm corresponding to 10%. The highest single-day precipitation was observed to be 61 mm w.e. recorded on 30 March 2015.

The daily mean variability of incoming and outgoing radiation components and $CF$ are shown in Fig. 5. About 62% of $S_{TOA}$ reached the surface at the AWS-M during DJFMA, indicating the remainder was absorbed and scattered by the cloud cover and atmospheric constituents (e.g., gases, water vapour). Daily mean $S_{in}$ varied between 28 and 414 W m$^{-2}$ corresponding to a mean of 191 W m$^{-2}$ (Table S3). $S_{in}$ was highest in April with a daily mean of 295 W m$^{-2}$. The persistent snow cover, especially during the peak winter period, resulted in a strong reflection of $S_{in}$ (Fig. 5). $S_{out}$ was the largest in March-April due to the accumulated snow cover ($\alpha_{acc} = 0.69$). $L_{in}$ followed the $CF$ pattern (Fig. 5). Low $L_{in}$ attributed to the low $CF$ (clear-sky) conditions. Daily mean $L_{in}$ varied between 123 and 290 W m$^{-2}$, corresponding to a mean of 203 W m$^{-2}$ (Table S3). $L_{in}$ was highest in April with a daily mean of 226 W m$^{-2}$. $L_{out}$ was relatively stable throughout DJFMA, ranging from 243 W m$^{-2}$ to 285 W m$^{-2}$, with a mean of 260 W m$^{-2}$ (Table S3).

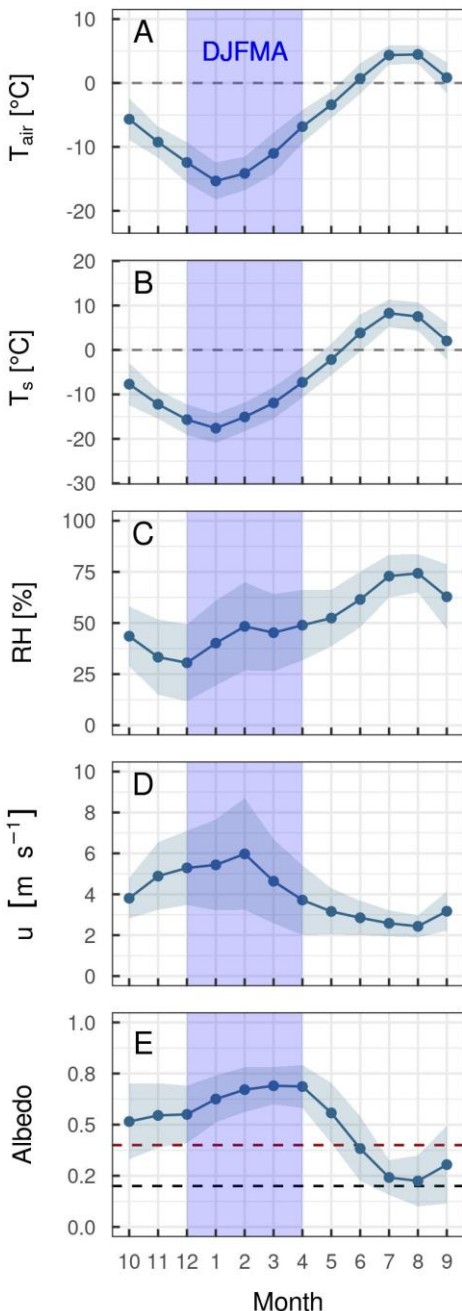

**Figure 2. Monthly climatology of air ($T_{air}$) and surface temperature ($T_s$), relative humidity ($RH$), wind speed ($u$) and surface albedo ($α_{acc}$) at the AWS-M for 2009-2020. DJFMA (1 December to 30 April) period is highlighted with a light blue rectangle in each panel. The shades around the line and scatter points represent the standard deviation (SD). Dashed lines in panel E refer to snow-surface albedo ($α_{acc}$ = 0.4; red line) for SEB analysis and bare-surface albedo ($α_{acc}$ = 0.2; black line). Daily values of $T_{air}$, $T_s$, $RH$, $u$ and albedo for the study period are shown in Fig. S2. Mean yearly values of different variables are provided in Table S4.**

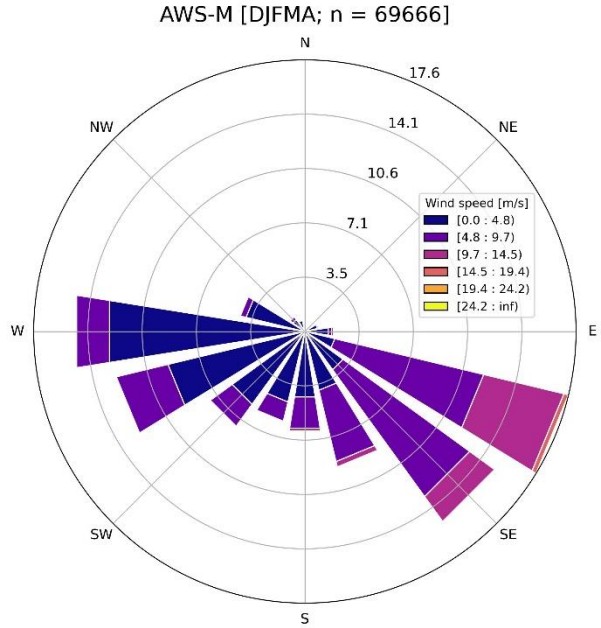

**Figure 3. Windrose of the AWS-M for DJFMA (2009-2020). The frequency of wind direction is expressed as a percentage based on n = 69666 half-hourly data points.**

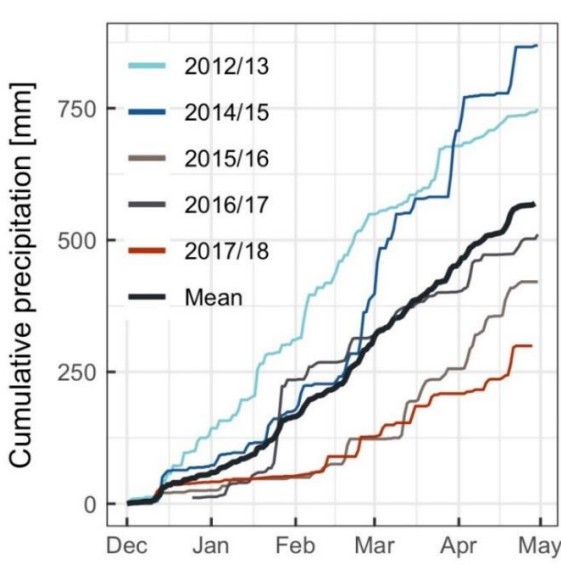

**Figure 4. Cumulative precipitation at the glacier base camp at 3850 m a.s.l. for DJFMA, 2012-2018. No data for 2013/14.**
**The bold line is the mean of all years.**

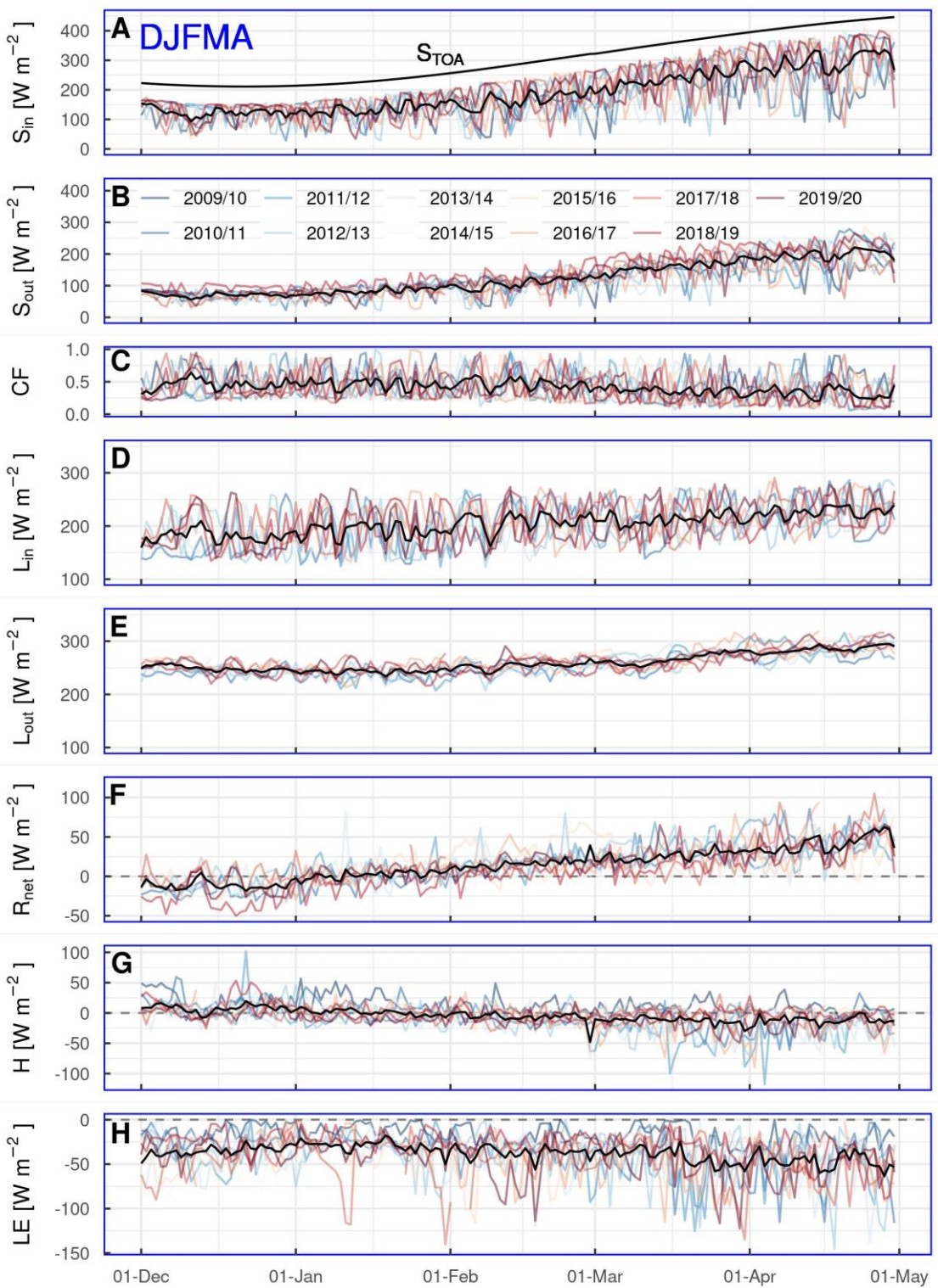

**Figure 5. The daily mean of short-wave radiation at the top of the atmosphere ($S_{TOA}$), short-wave incoming ($S_{in}$) and outgoing ($S_{out}$), cloud factor ($CF$), long-wave incoming ($L_{in}$) and outgoing ($L_{out}$), net radiation ($R_{net}$), turbulent sensible ($H$) and latent ($LE$) heat fluxes at the AWS-M for DJFMA, 2009-2020. $L_{in}$ and $R_{net}$ start from 1 December 2010. The black line highlights the mean of 2009-2020.**

## 4.2 Diurnal cycle of the meteorological variables and SEB components

Fig. 6 shows the mean diurnal cycle of meteorological variables and SEB components at the AWS-M for DJFMA (2009-2020). The mean diurnal cycle of $T_{air}$ and $T_s$ was well below 0°C. However, on certain days $T_s$ was above 0°C (6% of the half-hourly data points) but limited to peak daytime hours between 11:00 and 14:00 IST. Positive $T_s$ was observed when the snowpack was about 20 cm or lower (based on n = 38965 half-hourly SR50A data points between December 2009 and April 2015). $RH$ was the lowest around late-morning at ~10:00 IST and the highest in the evening at ~18:00 IST. $u$ was maximum during the afternoon (~14:00 IST), which corresponded well with the steep drop of $T_{air}$ in the afternoon, a typical valley glacier phenomenon (Greuell and Smeets, 2001).

$S_{in}$, $S_{out}$, $L_{in}$ and $L_{out}$ were the largest at noon, when the solar zenith angle was at its maximum, and the diurnal cycle for $CF$ was reversed. During the daytime, the energy from $S_{net}$ (balance between $S_{in}$ and $S_{out}$) was absorbed by the skin layer of the snow surface. $S_{net}$ was compensated by the energy loss through negative $L_{net}$ (balance between $L_{in}$ and $L_{out}$). The energy balance between $S_{net}$ and $L_{net}$, $R_{net}$ was then used to increase the turbulence of the surface boundary layer resulting in unstable conditions of the surface boundary layer (Fig. 6). The turbulent heat flux cycle was opposite of $S_{in}$, whereas same as $R_{ib}$ (stability). $H$ was positive throughout the night, then it started to sink to negative values for a few hours in the afternoon as the surface was heated up and again became positive in the evening. The unstable condition of the surface boundary layer ($R_{ib} < 0$; $T_{air}$ - $T_s$ < 0°C) was linked to the negative values of $H$ (Fig. 6). $LE$, unlike $H$, was always negative, although less negative in the morning and evening. $R_{ib}$ was mostly positive and small, corresponding to moderately stable surface boundary layer conditions except for about eight hours in the daytime between 9:00 and 16:00 due to the unstable surface boundary layer. $R_{net}$ was negative across the night and early morning, whereas positive during the daytime following the $S_{in}$ cycle. The negative $R_{net}$ indicated radiative cooling of the surface at night, while the positive $R_{net}$ suggested the heat transfer into the snow during the daytime. The $F_{surface}$ was consistently negative, being nearly zero during the late afternoons. Despite strong positive $R_{net}$ during peak daytime, $F_{surface}$ remained negative (Fig. 6). This was because a higher magnitude of negative $H$ + $LE$ considerably compensated a positive $R_{net}$. We found a high negative correlation between half-hourly values of $R_{net}$ and $H + LE$ ($r$ = -0.78; $p$ < 0.001, n = 59131), indicating that $R_{net}$ is responsible for the diurnal variation of $H$ and $LE$.

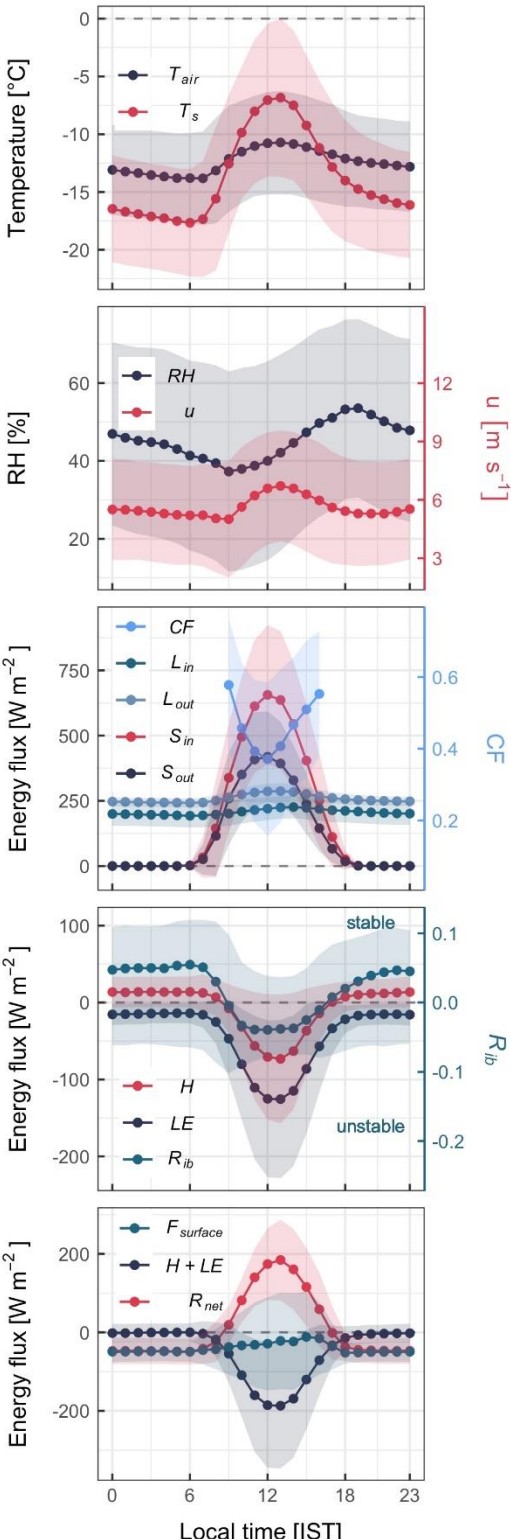


**Figure 6. Mean diurnal cycle of meteorological and SEB variables at the AWS-M for DJFMA. Half-hourly data were used between 2009 and 2020. *CF* was calculated between 09:00 and 16:00 IST. Shading is the SD.**

## 4.3 Seasonal and interannual variation of SEB components

$R_{net}$ was negative in December with a mean of -11 W m$^{-2}$, which gave rise to near-surface air cooling, whereas acted as a heat

supplier from February to April with a mean value of 15, 26 and 41 W m$^{-2}$, respectively (Fig. 5; Table S3). *H* was positive for 56% of the half-hourly values during December-January, suggesting that heat was carried from the atmosphere to the surface. *H* was negative for 44% of the half-hourly values during February-April, down to the monthly mean of -12 W m$^{-2}$. *LE* was always negative across DJFMA, suggesting mass loss through sublimation (Fig. 5; refer to Sect. 4.6). The mean monthly *LE* was most negative in April at -47 W m$^{-2}$, with a mean DJFMA value of -38 W m$^{-2}$ (Table S3). We analysed interannual

correlations between $R_{net}$ and turbulent fluxes to determine their inter-relationship. The correlations were strong and significant for both $R_{net}$ and *H* ($r = -0.70$; $p < 0.05$), and $R_{net}$ and *LE* ($r = -0.80$; $p < 0.05$). This further confirms that $R_{net}$ played an essential role in governing the turbulent fluxes at the AWS-M. The increased energy from $R_{net}$ combined with the longer duration of daylight hours along with the progression of winter results in more unstable boundary layer conditions, which supports stronger negative magnitudes of *H* and *LE*. $S_{in}$ showed stronger indirect relationship with *LE* and *H* ($r = -0.80$ and $-0.61$, respectively;

$p < 0.05$) than $L_{in}$ ($r = -0.36$ and $-0.39$, respectively; not significant). Together, $H+LE$ contributed a negative budget across DJFMA, with a mean monthly value of -40 W m$^{-2}$ (Table S3). As a result, the net energy ($F_{surface}$) was negative across DJFMA.

Fig. 7 presents the contributions of energy fluxes to SEB. During DJFMA, the proportional contribution of all SEB components showed that *LE* dominated the contribution (64%), followed by $R_{net}$ (25%) and *H* (11%) (Fig. S3). The mean monthly contribution showed an increasing contribution of $R_{net}$ with decreasing *LE* and *H* (Table S3). The largest contribution of $R_{net}$

in SEB is well noted across the HMA glaciers (Table S1). However, in this study, during the winter season, such a high contribution of *LE* (> 60%) is unique and contrary to the previous findings (e.g., Zhang et al., 2013; Azam et al., 2014a).

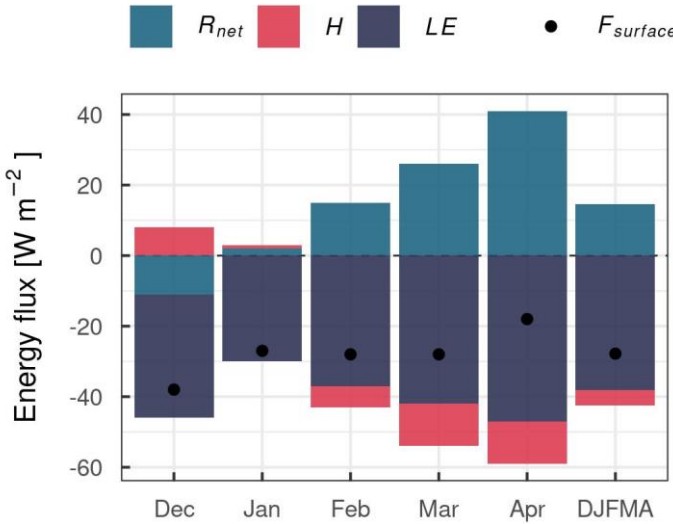

**Figure 7. Mean monthly energy flux density of $R_{net}$, $H$, $LE$ and $F_{surface}$ for DJFMA, 2009-2020. Monthly proportional contribution [%] of all SEB fluxes are shown in Fig. S3.**

### 4.4 Influence of could cover on SEB components in the daytime

We used *CF* values to differentiate between clear-sky when $CF \leq 0.2$ and overcast condition when $CF \geq 0.8$, following Chen et al. (2018). Around 24% of the data were categorised as clear-sky, while 10% in the overcast conditions based on n = 23903 half-hourly data points (09:00-16:00 IST; 2009-2020; Fig. 8). Overcast conditions decrease from January to April with increasing clear-sky conditions.

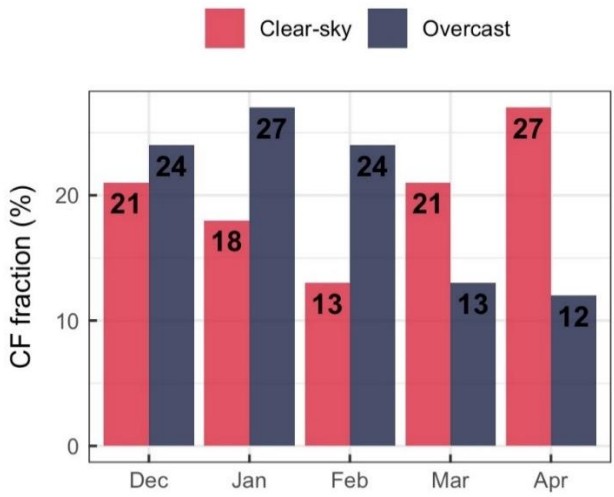

**Figure 8. Monthly fraction of clear-sky ($CF \leq 0.2$) and overcast ($CF \geq 0.8$) conditions at the AWS-M. Fraction percentage is calculated from n = 5810 clear-sky and n = 2381 overcast observations from total n = 23903 half-hourly values between 09:00 and 16:00 IST (DJFMA, 2009-2020).**

Fig. 9 shows the daytime half-hourly variation of $T_{air}$, $u$, $R_{ib}$ and SEB components. The stability of the surface boundary layer is notable in overcast conditions. Due to comparatively lower temperature (both $T_{air}$ and $T_s$) and higher $u$ in overcast conditions, the surface boundary layer remains near-neutral ($R_{ib}$ close to 0 due to low vertical temperature difference; $T_{air}$ - $T_s$). Conversely, high negative $R_{ib}$ values (unstable) were observed in clear-sky conditions. All the SEB components were considerably higher in clear-sky than in overcast conditions. On average, cloud cover subdued about 70% of the daytime mean $S_{in}$ (744 W m$^{-2}$ in

clear-sky as compared to 228 W m$^{-2}$ in overcast conditions). Unlike $S_{in}$, cloud cover increased the daytime mean $L_{in}$ by about 25% (201 W m$^{-2}$ in clear-sky as compared to 250 W m$^{-2}$ in overcast conditions).

    Turbulent heat fluxes were generally higher in clear-sky conditions due to higher instability of the surface boundary layer (Fig. 9). In clear-sky, the mean daytime $H$ was -66 W m$^{-2}$ which is three times more negative compared to overcast conditions (-21 W m$^{-2}$), corresponding to 68% reduction of $H$ in overcast conditions than in clear-sky. Similarly, the mean daytime $LE$ was

also higher in clear-sky, with -136 W m$^{-2}$ compared to -47 W m$^{-2}$ in overcast conditions (65% reduction). The reduced magnitude of turbulent heat fluxes in overcast/cloudy conditions was due to the neutral stability of the surface boundary layer (Fig. 9B; $R_{ib} \approx 0$). In neutral stability conditions, cold temperature ($T_{air} - T_s$ close to 0) restricts the magnitude of $H$ and $LE$ (Fig. 9). In clear-sky conditions, more negative $LE$ was due to the surface's intense heating ($T_{air}$ - $T_s < 0°C$), which creates a stronger vertical moisture gradient ($q$ - $q_s$) than overcast conditions. $F_{surface}$ showed a slight daytime variation during clear-sky,

but no significant variation in overcast conditions.

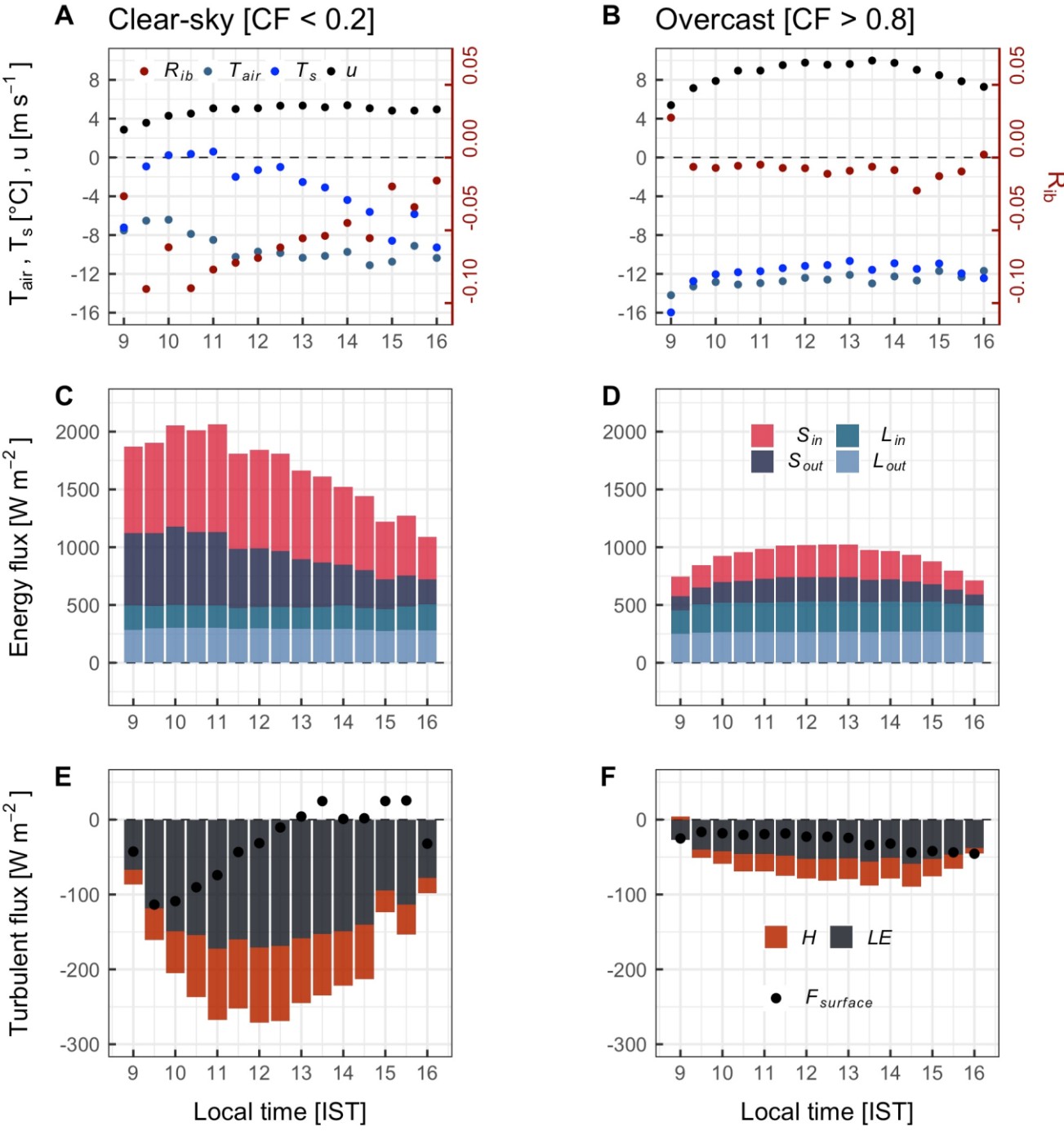

**Figure 9. Daytime (09:00-16:00 IST) diurnal cycle of $T_{air}$, $T_s$, $u$, $R_{ib}$ and SEB components under the clear-sky ($CF \leq 0.2$) and overcast ($CF \geq 0.8$) conditions.**

## 4.5 Relationship of turbulent heat fluxes and meteorological variables under different cloud conditions

Sub-hourly scale correlations were developed to better understand the relationship between $H$ and $LE$ and meteorological variables (Fig. 10). $H$ was strongly and positively correlated with $T_{air} - T_s$ in clear-sky ($r = 0.96$; $p < 0.001$) and overcast conditions ($r = 0.83$; $p < 0.001$). That means $H$ increases as the difference in vertical temperature increases towards negative direction (Fig. 11). Similarly, $LE$ was strongly correlated with $T_{air} - T_s$ in clear-sky ($r = 0.84$; $p < 0.001$) but moderately correlated in overcast conditions ($r = 0.50$; $p < 0.001$), which suggests that the vertical temperature difference significantly controls the near-surface vertical moisture gradient (one of the primary drivers of $LE$). This attributes to a significantly higher negative $LE$ in clear-sky than in overcast conditions (Fig. 11). The correlation of $H$ with $q - q_s$ was moderate in clear-sky ($r = 0.57$; $p < 0.001$) and weak in overcast conditions ($r = 0.10$; $p < 0.001$). Correlation of $LE$ with $q - q_s$ was strong in clear-sky ($r = 0.82$; $p < 0.001$) as well as in overcast conditions ($r = 0.70$; $p < 0.001$). That means $LE$ increases as the vertical difference in moisture increases towards negative direction (Fig. 11). Due to higher near-surface heating and convection, the near-surface moisture gradient is steeper in clear-sky than in overcast conditions (Fig. 11). There is a clear pattern of more negative $LE$ with an increasing $q - q_s$; however, the correlations were not very strong ($r = 0.82$ in clear-sky; $r = 0.70$ in overcast). This could be partly explained by the overestimation of $LE$ in near-neutral conditions ($R_{ib} \approx 0$), which increases the stability function ($\Phi_{m/h/v}$), resulting in a higher magnitude of $LE$. The difference in atmospheric stability in clear-sky and overcast conditions explains the difference in correlations. In this regard, Steiner et al. (2018) discussed that atmospheric stability correction is crucial to estimate $H$ and $LE$ accurately under different cloud conditions and tricky to handle for a rapidly changing mountain atmosphere. No strong correlation was observed between $H$ and $u$ in both clear-sky ($r = -0.27$; $p < 0.001$) and overcast conditions ($r = -0.35$; $p < 0.001$). Similarly, $LE$ and $u$ were also not strongly correlated both in clear-sky ($r = -0.37$; $p < 0.001$) and overcast conditions ($r = -0.33$; $p < 0.001$). However, the negative correlation with $u$ suggests that $H$ and $LE$ increase towards negative direction as $u$ increase and vice versa. The weak correlation between $LE$ and $u$ could be explained in part by the very strong $u > \sim10$ m s$^{-1}$ at the AWS-M site, with high $RH > \sim70\%$ (Fig. S5), which limits the magnitude of $LE$. Such strong winds were often observed in overcast conditions with high cloud cover (Fig. S5A) and precipitation (Fig. S5B) and were likely associated with WDs storms. WDs events are most dominant during winter months around the Chhota Shigri region. This was observed from the ERA5's horizontal wind fields and vertically integrated moisture divergence datasets at 500 hPa from 2009 to 2020 (Fig. S6). Zhu et al. (2021a) and Liu et al. (2020) also indicated that during the winter months in the western Himalaya and western Tibetan regions, WDs storm activities transport a significant amount of moisture and influence the precipitation. A very strong $u$ during WDs kept the snow surface cool and maintained a reduced $T_{air} - T_s$ and $q - q_s$ (close to zero), resulted in a low magnitude of $LE$ (Fig. S7). At the sub-hourly scale, neither $R_{net}$ nor $S_{in}$ and $L_{in}$ can adequately explain turbulent fluxes in both overcast and clear-sky conditions ($r = < 0.50$; Fig. 11). Overall, we noted that at the sub-hourly

scale near-surface moisture availability (through $q - q_s$) plays a bigger role in determining the magnitude of $LE$, with the combined effects from several meteorological variables, particularly $q_s$, $T_s$ and $u$.

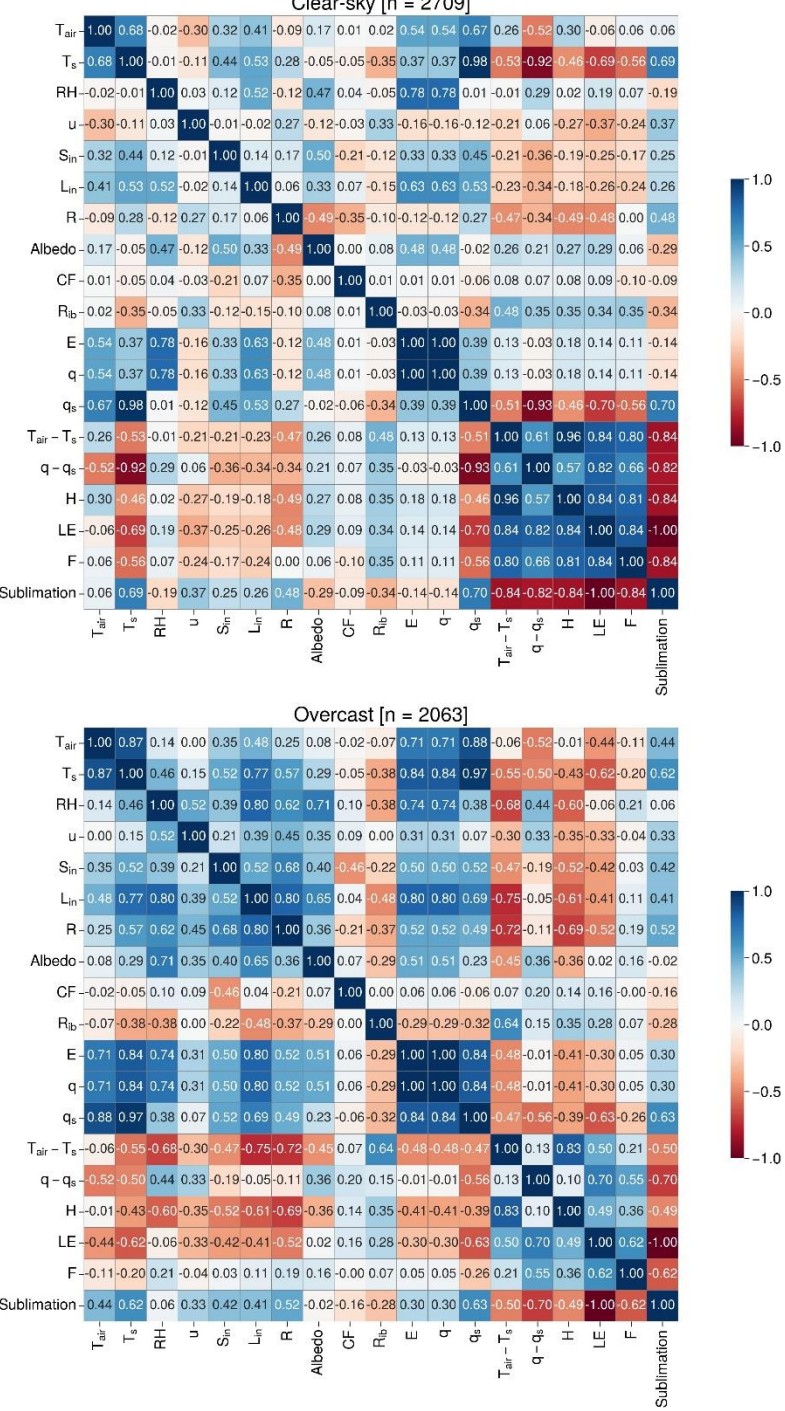

**Figure 10. Pearson correlation coefficient (*r*) matrix of various meteorological and SEB components at the AWS-M in clear-sky and overcast conditions between 09:00 and 16:00 IST, 2009-2020. Number (n) of half-hourly data points are shown on top of the panels.**

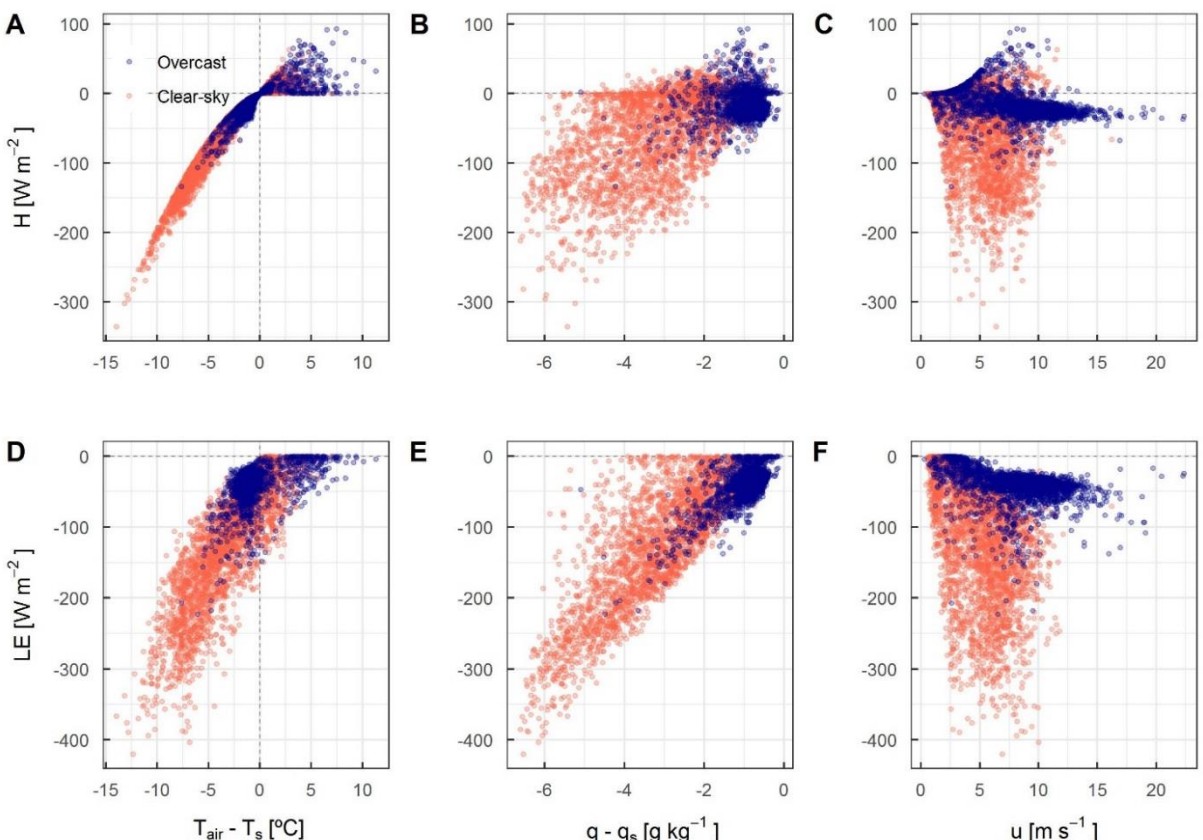

**Figure 11. Half-hourly values of the vertical temperature difference ($T_{air}$ - $T_s$), vertical specific humidity difference ($q$ - $q_s$) and $u$ compared with $H$ and $LE$ between 09:00 and 16:00 IST for DJFMA, 2009-2020. Red circle represents clear-sky (n = 2709), and blue represents overcast conditions (n = 2063).**

### 4.6 Sublimation and its relationship with meteorological variables

Half-hourly $LE$ fluxes were converted to sublimation following Eq. 8. At the AWS-M site, the mean daily sublimation was 1.1 ± 0.5 mm w.e., corresponding to a cumulative DJFMA mean sublimation loss of 145 ± 25 mm w.e. a$^{-1}$ over 2009-2020 (Table 2). The mean daily sublimation rate was almost three times higher in clear-sky (3.7 ± 2.6 mm w.e.) conditions than in overcast (1.3 ± 0.8 mm w.e.), indicating the critical role of cloud cover. The mean monthly sublimation was the highest in March, at 32 ± 7 mm w.e., and the lowest in January, at 26 ± 7 mm w.e., with intra-annual variability in different years. The yearly cumulative sublimation varied across the study period, from a minimum of 85 mm w.e. in 2009/10 to 172 mm w.e. in 2019/20 (Table 2). Notably, the snowfall amounts were often similar over these years (e.g., 2009/10, 2017/18, 2019/20), suggesting a stronger control of other meteorological variables in sublimation, particularly $RH$, $T_s$ and $S_{in}$ than snowfall (Table 2). For

example, in 2009/10, cumulative sublimation was the lowest (85 mm w.e.), which was associated with the lowest $T_s$ (-13.4ºC)

and the highest frequency of $RH > 80\%$ (8.9%) during the study period (Table 2; Fig. S8). Further, $S_{in}$ was also the lowest in 2009/10 (Table 2). The opposite condition prevailed during 2017/18 and 2019/20 when $T_s$ were considerably higher at -6.7ºC and -7.4ºC, respectively, $RH > 80\%$ was the lowest at 3.6% and 4.4%, respectively and $S_{in}$ was the highest at 491 and 494 W m$^{-2}$, respectively. We further assessed the relationship through the interannual correlation analysis based on 11-year sublimation and primary meteorological variables. Interannual correlation between cumulative sublimation and $T_s$ was the

highest ($r = 0.85$; $p < 0.01$) followed by $S_{in}$ ($r = 0.79$; $p < 0.05$) and $RH > 80\%$ ($r = -0.76$; $p < 0.01$) (Table S5). This suggests that on an interannual scale, high $T_s$ (through higher $S_{in}$) and low near-surface moisture conditions support sublimation. Cloud cover, on the other hand, has a significant impact on the primary meteorological variables, particularly $S_{in}$, $T_s$ and $q_s$ (Fig. 9; Sect. 4.4).

**Table 2. Monthly sum of sublimation (mm w.e.), cumulative sublimation ($S_c$; mm w.e.), snowfall (mm w.e.) and the fraction of sublimation to snowfall ($S_{fra}$; %) at the AWS-M during 2009-2020. Snowfall is based on Geonor and Keylong ('*' marked) precipitation data (see Sect. 3.1). Mean DJFMA meteorology for daytime (08:00 and 16:00 IST) is also shown for corresponding years. $RH > 80\%$ is the frequency of $RH > 80$ in a particular year.**

| Month | 2009/10 | 2010/11 | 2011/12 | 2012/13 | 2013/14 | 2014/15 | 2015/16 | 2016/17 | 2017/18 | 2018/19 | 2019/20 | Mean±SD |
|---|---|---|---|---|---|---|---|---|---|---|---|---|
| **December** | 14 | 20 | 30 | 30 | 32 | 49 | 34 | 9 | 39 | 21 | 34 | 28±12 |
| **January** | 11 | 22 | 26 | 37 | 27 | 29 | 31 | 23 | 31 | 22 | 30 | 26±7 |
| **February** | 16 | 27 | 31 | 27 | 38 | 26 | 37 | 35 | 31 | 23 | 38 | 30±7 |
| **March** | 19 | 34 | 42 | 27 | 31 | 27 | 27 | 40 | 36 | 28 | 39 | 32±7 |
| **April** | 25 | 30 | 27 | 29 | 31 | 23 | 29 | 30 | 29 | 27 | 31 | 28±3 |
| $S_c$ [mm w.e.] | 85 | 133 | 156 | 150 | 159 | 153 | 159 | 138 | 167 | 121 | 172 | 145±25 |
| **Snowfall [mm w.e.]** | 485* | 474* | 415* | 850 | 458* | 971 | 522 | 613 | 402 | 675* | 451* | 574±187 |
| $S_{fra}$ [%] | 18 | 28 | 38 | 18 | 35 | 16 | 30 | 23 | 42 | 18 | 38 | 27±10 |
| $T_{air}$ [ºC] | -9.8 | -11.2 | -12.0 | -10.6 | -11.6 | -10.2 | -9.8 | -9.5 | -9.3 | -11.2 | -11.1 | -10.6±0.9 |
| $T_s$ [ºC] | -13.4 | -10.5 | -8.5 | -8.1 | -8.7 | -7.7 | -7.3 | -6.2 | -6.7 | -10.9 | -7.4 | -8.7±2.1 |
| $u$ [m s$^{-1}$] | 5.0 | 5.2 | 5.9 | 4.9 | 5.5 | 4.7 | 4.9 | 5.3 | 4.8 | 5.0 | 4.5 | 5.0±0.4 |
| $RH$ [%] | 41 | 40 | 43 | 39 | 40 | 38 | 36 | 39 | 34 | 41 | 38 | 39±3 |
| $RH > 80\%$ [%] | 8.9 | 8.2 | 5.5 | 7.5 | 5.2 | 7.1 | 5.0 | 6.9 | 3.6 | 5.8 | 4.4 | 6.2±1.7 |
| $S_{in}$ [W m$^{-2}$] | 382 | 481 | 462 | 480 | 465 | 476 | 490 | 465 | 491 | 485 | 494 | 470±31 |

Fig. 12 presents the daytime diurnal cycle of sublimation, $u$ and $q$ for six different meteorological clusters: (1) no filter, (2) high $u$ ($> 10$ m s$^{-1}$), (3) high $q$ ($> 2$ g kg$^{-1}$), (4) low $q$ ($< 1$ g kg$^{-1}$), (5) higher $T_s$ ($> -10$°C) and (6) lower $T_s$ ($< -10$°C). We omitted measurements during the night when sublimation is negligible. Sublimation peaks in the early afternoon between 12:00 and 14:00 hours (Fig. 12), soon after the AWS-M site was sunlit. High insolation during the late morning (10:00 - 12:00

IST; Fig. 7) increases $T_{air}$ - $T_s$, resulting in stronger convection in the early afternoon, which favours sublimation. Once the

snow surface is heated up, the sublimation is conditioned by $q$ - $q_s$. A low $q$ below 1 g kg$^{-1}$ and high $T_s$ above -10°C enhance sublimation (Fig. 12 and 13). Higher $q$ restricts sublimation because the near-surface atmosphere is saturated; consequently, the vertical water vapour pressure gradient is weak. Sublimation was the largest when $T_{air}$ ranged between -5°C and -10°C and also when $T_s$ ranged between 0°C and -10°C (Fig. 12; Fig. 13B and C). Sublimation was considerably lower when moisture availability was higher, $T_s$ was significantly lower, with very strong $u$ (Fig. 12; Fig. 13). This was probably associated with the

cold storm events through WDs, which brings high moisture (Fig. S5) and cold winds in the region (Fig. S7; discussed in Sect. 4.5). Thus, very strong and cold winds with higher moisture from WDs impedes sublimation in the region. Guo et al. (2022) observed a similar phenomenon in the August-One Glacier in the north-east Tibetan Plateau, where sublimation was significantly constrained by extremely low $T_s$ during strong westerlies.

To further understand the combined effect of meteorological variables on sublimation, a multiple linear regression analysis

was performed (Table 3). The multiple regression shows that $q - q_s$, $T_{air} - T_s$, $u$ and $T_s$ were the best sublimation predictors in clear-sky and overcast conditions as well as in all-data conditions (without $CF$ filter). Considering two combined predictors, $q$ - $q_s$ and $u$ explained the highest variance (> 80%) in sublimation in clear-sky and overcast conditions as well as in all-data conditions. When three predictors were considered, it is the combination of $T_{air}$ - $T_s$, $q$ - $q_s$, $u$ explained the highest variance, with 95% in clear-sky and > 90% in overcast and all-data conditions. However, it is noteworthy that individually $u$ explains

the poor variance in sublimation (< 40% in clear-sky and overcast conditions; Fig. 10). Stigter et al. (2018) noted a slightly higher variance of $u$ (48%) in sublimation at the Yala Glacier in the central Himalaya.

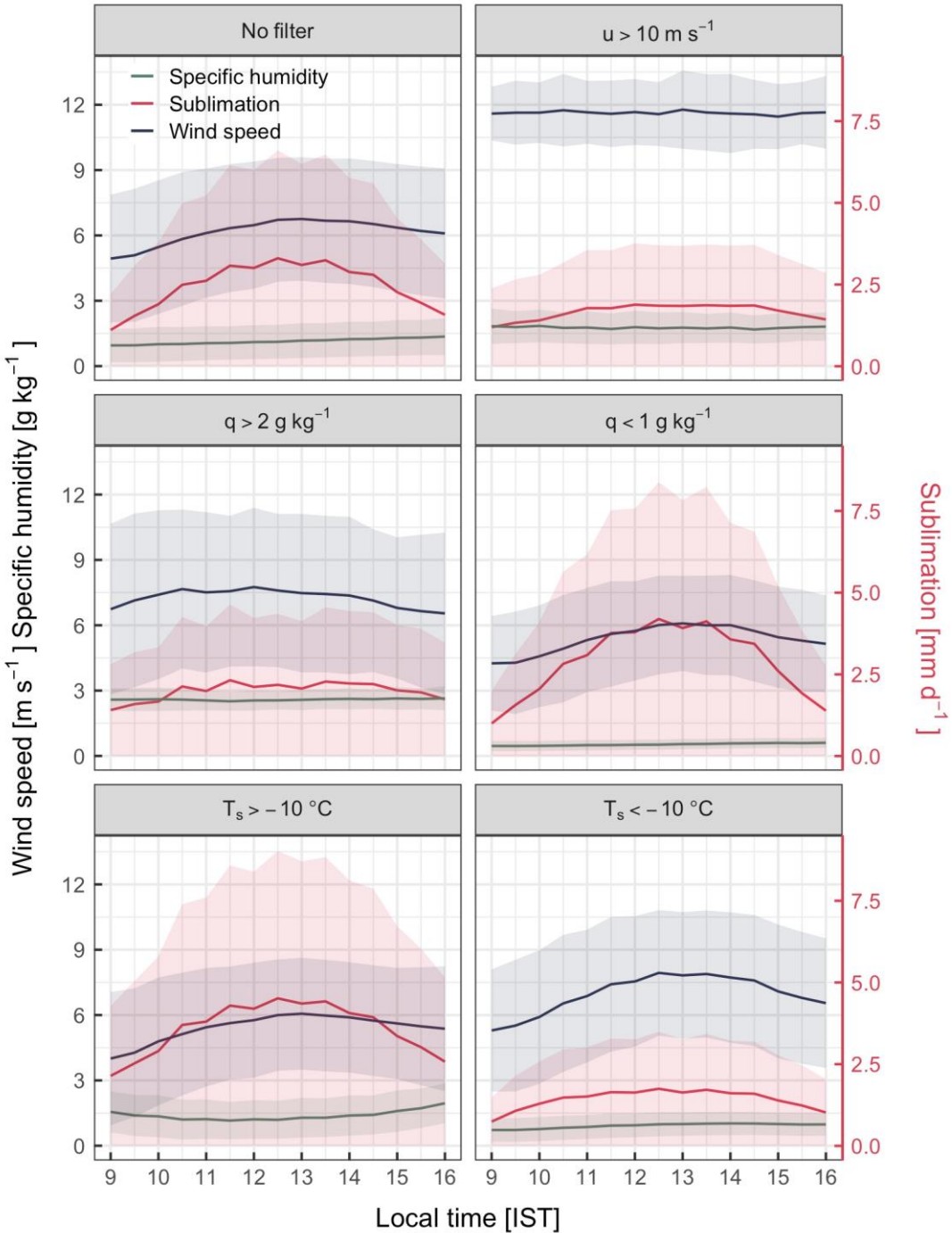

**Figure 12. Half-hourly daytime (09:00-16:00) records of sublimation (red), wind speed (blue) and specific humidity (green) at the AWS-M for different clusters: no filter, $u > 10$ m sec$^{-1}$, $q > 2$ g kg$^{-1}$, $< 1$ g kg$^{-1}$, $T_s > -10°$C and $T_s < -10°$C.**
**Data period: DJFMA, 2009-2020. Number of data-points n=30257, 2347, 12295, 9762, 10552 and 12734 for no filter, $u$ $> 10$ m sec$^{-1}$, $q > 2$ g kg$^{-1}$, $< 1$ g kg$^{-1}$, $T_s > -10°$C and $T_s < -10°$C, respectively.**

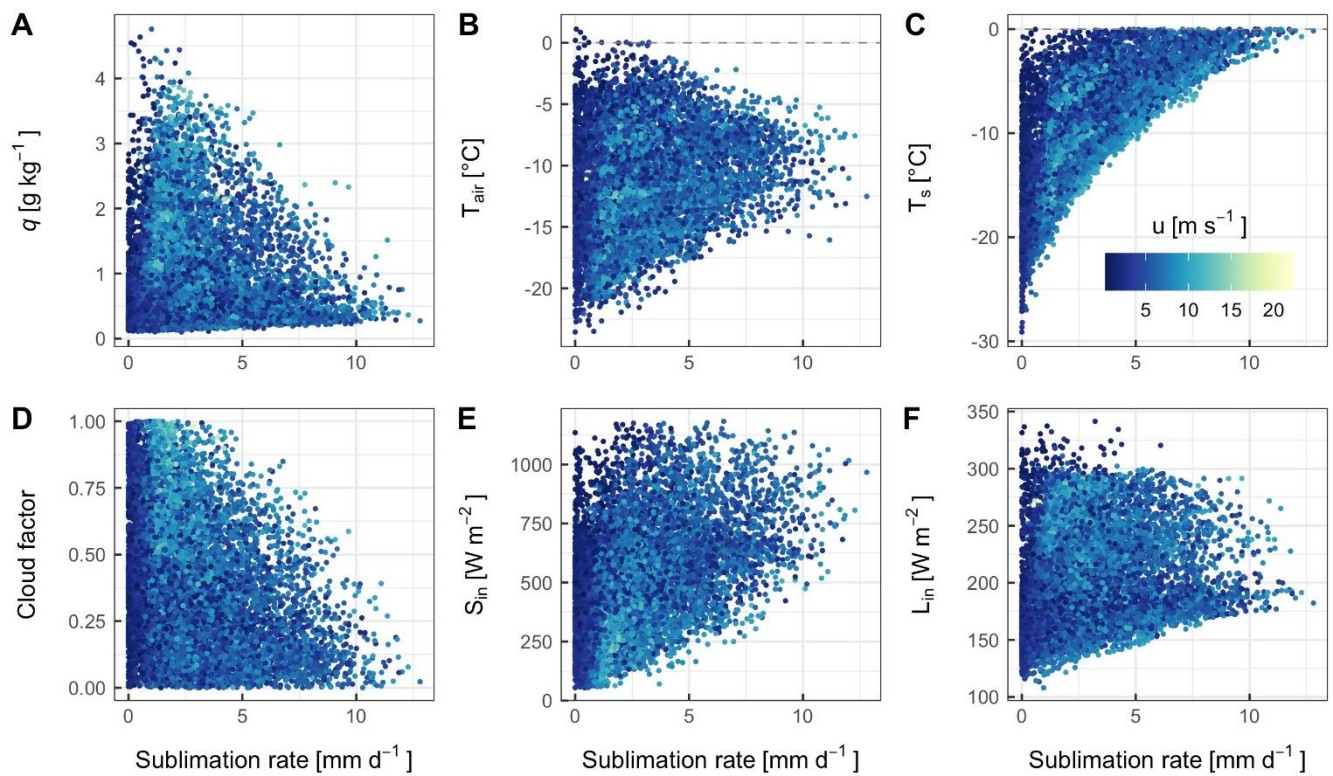

**Figure 13. Scatter plot of $u$, $q$, $T_{air}$, $T_s$, $CF$, $S_{in}$ and $L_{in}$ against sublimation rate at the AWS-M. The colour of the data**
**points refers to the measured wind speed ($u$). Total n = 14088 half-hourly data points between 09:00 and 16:00 IST for DJFMA (2009-2020).**

**Table 3. Summary of the multiple linear regression analysis (k-fold (k = 10) cross-validation) of sublimation rate and combined meteorological variables. Total n = 13217, 2708 and 2063 half-hourly data points for all-data, clear-sky and overcast conditions, respectively, between 09:00 and 16:00 IST for DJFMA (2009-2020). The $p$-value of $r^2$ was always**
**< 0.001.**

| Variable | $r^2$ cross-validation | | |
|---|---|---|---|
| | **All-data** | **Clear-sky** | **Overcast** |
| $T_s$, $u$ | 0.53 | 0.69 | 0.44 |
| $T_{air}$, $u$ | 0.10 | 0.17 | 0.30 |
| $q$, $u$ | 0.03 | 0.15 | 0.15 |
| $q_s$, $u$ | 0.58 | 0.71 | 0.47 |
| $u$, $T_{air}$-$T_s$ | 0.58 | 0.75 | 0.29 |
| $u$, $q$-$q_s$ | 0.86 | 0.85 | 0.84 |
| $q$, $u$, $T_{air}$ | 0.26 | 0.21 | 0.34 |

| | | | |
|---|---|---|---|
| $q, u, T_s$ | 0.79 | 0.82 | 0.71 |
| $q_s, u, T_{air}$ | 0.77 | 0.90 | 0.51 |
| $q_s, u, T_s$ | 0.59 | 0.71 | 0.48 |
| $T_{air}-T_s, q-q_s, u$ | 0.92 | 0.95 | 0.89 |
| $T_{air}-T_s, q-q_s, S_{in}$ | 0.85 | 0.85 | 0.67 |
| $T_{air}-T_s, q-q_s, L_{in}$ | 0.84 | 0.85 | 0.67 |
| $T_{air}-T_s, q-q_s, R_{net}$ | 0.85 | 0.86 | 0.70 |

## 5 Discussion

### 5.1 Factors controlling the latent heat flux/sublimation

We note that the magnitude of *LE* is governed by a combined effect of different meteorological variables, primarily the vertical
moisture and temperature gradients, wind speed and the state of the surface boundary layer (stability). The relationship between
*LE* and meteorological variables, on the other hand, varied in temporal scale, making it complex. Despite *LE* and $R_{net}$ having
a strong relationship on an interannual scale, we did not find a strong relationship in the sub-hourly scale, emphasising the
importance of temporal scale in understanding sublimation. On a sub-hourly scale, we found no strong correlation between
*LE*/sublimation and individual meteorological variables (Fig. 10). The absence of strong correlation between sublimation rate
and one or the other meteorological variables is expected because a conducive environment for enhanced sublimation is created
by a combination of meteorological variables. For example, cloud cover shapes the prevailing weather conditions at the study
site by influencing the stability of the surface boundary layer (Fig. 9). In a stable stratification ($T_{air} - T_s > 0°C$), the snow surface
remains cooler than the air, which attributes to a gentle near-surface moisture gradient and a lower *LE*, whereas in an unstable
stratification ($T_{air} - T_s < 0°C$), steep near-surface moisture gradient results in a high negative *LE*. The other important aspect is
the availability of moisture content in the air which is a function of various meteorological variables, such as precipitation, the
vapour pressure at the surface or above, etc., all of which have a role in promoting sublimation. For example, a low *q* helps to
create a steeper negative moisture gradient which increases sublimation (Fig. 13). Stigter et al. (2018) and Guo et al. (2021)
also reported a similar process where an integrated effect was responsible for higher sublimation in the Yala and August-One
glaciers. The integrated effect of different meteorological variables in supporting sublimation also explains the weak
correlation between *LE*/sublimation and *u* ($r = < 0.40$; Fig. 10). Stigter et al. (2018) and Guo et al. (2021) noted a strong direct
relationship between *LE* and *u,* which does not agree with the present study. This could be partly explained by the highly
heterogeneous *u* at the AWS-M (Fig. 13). For example, the available observations from different sites showed that *u* generally
decreases in overcast conditions (e.g., Stigter et al., 2018; Guo et al., 2021; Conway et al., 2022). However, at the AWS-M, *u*
was often higher in overcast conditions (Fig. 9; Fig. S5) due to westerly activities (discussed in Sect. 4.5 and 4.6). This
heterogeneity is likely the cause of the weak correlation between *u* and sublimation in part. However, the highest multiple
regression variance in combination with *u* (~90%; Table 3) in clear-sky and overcast conditions emphasise the importance of
*u* in driving *LE*/sublimation. Fugger et al. (2022) also observed that the relationship between *LE* and meteorological variables

is highly unpredictable, and $u$ fails to explain the variability of $LE$ at five on-glacier sites in the central and eastern Himalaya (see their Fig. 9A). We note the importance of cloud cover in modulating the surface atmosphere at the AWS-M site which favours sublimation, however, the correlation coefficient between $CF$ and $LE$ was poor ($r$ = -0.09 and -0.16 in clear-sky and overcast conditions, respectively; Fig. 10). This is most likely due to the complex influence of cloud cover on meteorological variables, particularly $S_{in}$ and $L_{in}$. Cloud cover reduces $S_{in}$, which impede sublimation, but at the same time it also increases $L_{in}$, which promotes sublimation partly by raising the $T_s$. This is well-supported by the higher correlations between sublimation and $S_{in}$ and $L_{in}$, particularly in overcast condition (Fig. 10). Although Stigter et al. (2018) did not discuss the correlation between sublimation and cloud cover/factor at the Yala Glacier, they did indicate that sublimation was negligible on overcast days when $RH$ was higher. This is supported by the poor correlation of determination ($r^2$ = 0.08) between sublimation and $RH$ at the Yala Glacier. Guo et al. (2021) also did not obtain a statistical relationship between sublimation and cloud cover, but they also noted a weak sublimation rate during cloudy months due to high moisture and warm conditions. Conway et al. (2022) also found that an increase in cloud cover decreases the magnitude of $LE$ at four on-glacier Himalayan sites, including the Chhota Shigri Glacier. Overall, we conclude that the near-surface moisture availability (through $q$ - $q_s$) plays a major role in governing the magnitude of $LE$ at the AWS-M at different temporal scales, while moisture availability was influenced and conditioned by a number of meteorological variables, notably $S_{in}$, $u$, $q_s$, and $T_s$.

**5.2 Sublimation sensitivity to meteorology and roughness and uncertainty sources**

To test the sensitivity of the calculated sublimation to changes in the input data, we prescribed perturbations of $T_{air}$ ($\pm$ 1°C), $T_s$ ($\pm$ 1°C), $u$ ($\pm$ 10%), $RH$ ($\pm$ 10%) and $z_{0m}$ (0.0005 m, 0.002 m, 0.003 m, and 0.004 m) and re-calculated sublimation for DJFMA, 2009-2020. Similar perturbations for the meteorological variables were applied in previous studies (Andreassen et al., 2008; Zhang et al., 2013; Steiner et al., 2018; Liu et al., 2021). For $z_{0m}$, we chose higher and lower order perturbation values considering the high SD of in-situ calculated snow $z_{0m}$ at the AWS-G (0.001 $\pm$ 0.003 m; Azam et al., 2014a). Results show that sublimation is most sensitive to $z_{0m}$ and $T_s$ (Fig. 14) because they are the direct drivers of $LE$. Perturbation of higher order $z_{0m}$ (0.004 m) and +1°C change in $T_s$ increase the mean cumulative sublimation by 21% (30 mm w.e.). For a much lower order $z_{0m}$ (0.0005 m), the mean cumulative sublimation decreases by 8% (12 mm w.e.). Perturbation to $\pm$ 10% change in $u$ yields a $\pm$ 8% change in sublimation. The mean cumulative sublimation is roughly three times more sensitive to a $\pm$ 1°C change in $T_s$ than a $\pm$ 10% change in $RH$ and $u$.

Sublimation/$LE$ sensitivity in this study is similar to that reported for the Lirung Glacier, which, however, has a debris-covered surface (Steiner et al., 2018). They noted that a $\pm$ 1°C change in $T_s$ results in a -42% and 23% change in $LE$. They also note that $LE$ is less sensitive ($\pm$ 8%) to a $\pm$ 10% change in $u$. Liu et al. (2021) also reported that $LE$ is considerably less sensitive to change in $u$ and $RH$ (< $\pm$ 10% sensitive) than $T_s$ and $z_{0m}$ (> $\pm$ 20% sensitive) on the clean-ice East Rongbuk Glacier in the Everest region. In general, sublimation is less sensitive to the meteorological variables ($T_{air}$, $RH$ and $u$) than $z_{0m}$. However, it could be higher or significant as the change of $\pm$ 1°C in $T_{air}$ or $\pm$10% of $RH$ and $u$ can equally be caused by sensor inaccuracies

provided by the manufacturer (Table 1). That means the sensitivity to $T_{air}$, $RH$ and $u$ could be roughly equal to $z_{0m}$ or $T_s$. Sensitivities reported in this study have crucial implications in improving the existing hydrological models and distributed SEB models, where sublimation loss is ignored (Srivastava and Azam, 2022; Patel et al., 2021). Another important aspect of the sensitivity to meteorological variables is related to the future atmospheric warming and its consequences to sublimation. $T_s$ exhibited a higher sublimation sensitivity than $T_{air}$ (Fig. 14), but under melting condition $T_s$ will not change much because

the temperature of the snow/ice surface cannot rise above the melting point ($T_s = 0°C$). However, relative potential changes in $T_{air}$ are likely to be higher across the globe including in the Himalayan region (Hock et al., 2019; Krishnan et al., 2019). Therefore, sublimation sensitivity with respect to $T_{air}$ could be a major concern in future, due to the expected warming. Considering a future $T_{air}$ increase of ~0.3 ± 0.2°C decade$^{-1}$ for the Himalayan region (Ren et al., 2017; Krishnan et al., 2019), a crude estimate suggests a ~5% decrease in sublimation per decade from the snow/glacier surfaces. This could possibly lead

to a lower energy sink through the $LE$ flux, which will boost the efficiency of $S_{in}/R_{net}$ resulting in more surface melt. However, since sublimation is a process driven by the combined effect of multiple meteorological variables, it remains to be seen how the sensitivity of a single variable influences the overall sublimation and associated processes. The bulk method was already used in the HK region (Table 4), where the climate setting was similar to that of the Chhota Shigri region, with strong wind and dry conditions. We used $z_{0m}$ (0.001 m) which was calculated at the AWS-G site applying a logarithmic profile based on

wind speed data from two levels (Azam et al., 2014a), which might have reduced the potential bias from choosing a random $z_{0m}$ or from the existing literature. However, $z_{0m}$ could be higher or lower depending on the snow redistribution at the AWS-M site, which is expected at such a high altitude. Another important uncertainty source of sublimation is blowing snow and erosion (Wagnon et al., 2013), especially over a strong wind-prone site. A wide variation of blowing snow sublimation rates is reported in the literature, depending on the climate and snow blow model setup (Zwaaftink et al., 2013). However, modelling

of blowing snow sublimation is beyond the scope of this study and might have led to an underestimation of the sublimation. Nevertheless, considering all the above uncertainties, the mean daily sublimation at the AWS-M site (1.1 mm d$^{-1}$) agrees well with the eddy-covariance-based sublimation of 1 mm d$^{-1}$ at the Yala Glacier (Stigter et al., 2018), where the reported meteorological condition is similar as at the AWS-M.

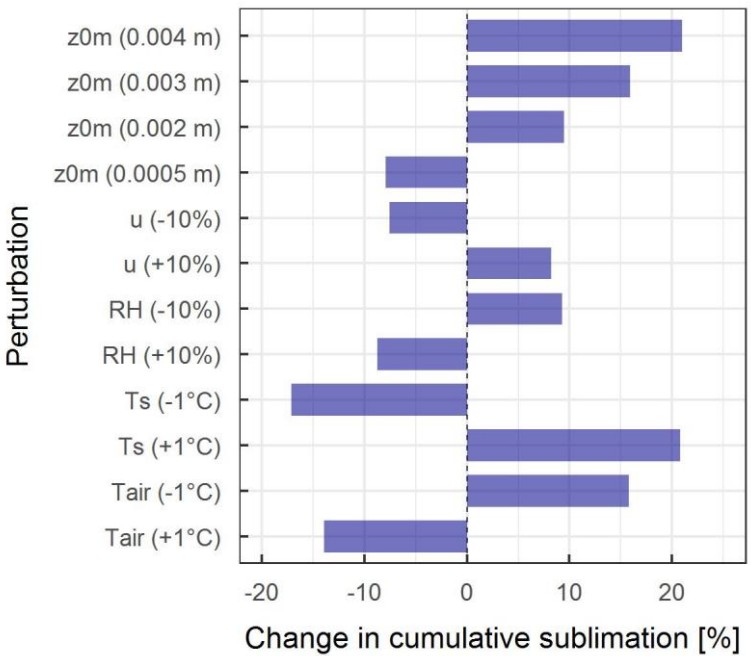

**Figure 14.** Calculated change in mean cumulative sublimation after applying perturbations to $T_{air}$ (± 1°C), $T_s$ (± 1°C), $u$ (± 10%), $RH$ (± 10%), snow $z_{0m}$ (0.0005 m, 0.002 m, 0.003 m, and 0.004 m).

**5.3 Comparison of sublimation rates with other HK/HMA glaciers**

This section discusses the existing sublimation rates/studies across the HMA glaciers/snow-covered sites compared to the AWS-M site on the Chhota Shigri Glacier (Table 4). The existing sublimation studies in the HK and HMA are not uniform with respect to the spatial and temporal scales, which makes it difficult to compare sublimation and associated processes consistently. However, it is worthwhile to use these existing sublimation datasets for comparison, not to conduct a thorough and rigorous comparison, but to qualitatively address the sublimation process in the region. The mean daily winter sublimation rate estimated in this study (1.1 ± 0.5 mm d$^{-1}$) is roughly similar to the mean sublimation (~0.2 to ~2 mm d$^{-1}$) on the other glacier/snow-covered sites across the HMA (Table 4). Sublimation rates during winter were slightly higher in the Pamir Range, e.g., Muztag Ata No. 15 (Zhu et al., 2018) and the Muji site (Zhu et al., 2020) compared to the inland/central Tibet region, e.g., Qiangtang No. 1 (Li et al., 2018) and the Dongkemadi site (Liang et al., 2018). This is likely due to the relatively drier atmospheric conditions in the Pamir Range than the central or eastern parts of Tibet (Table 4; also Liu et al., 2020). However, such spatial understanding needs more studies and direct measurements to confirm. The only in-situ lysimeter-based sublimation is available on the East Rongbuk Glacier measured at ~6500 m a.s.l. (Yang, 2010). Their measured sublimation rate was 1.9 mm d$^{-1}$ during late winter that is similar to the upper limit of our long-term daily sublimation rate. The only eddy-covariance measured sublimation rate during winter at the Yala Glacier was 1 mm d$^{-1}$, which is similar to the sublimation calculated at the AWS-M on the Chhota Shigri Glacier. At the Pindari Glacier AWS site (off-glacier at 3750 m a.s.l.), the

sublimation rate for a transient snow-cover was estimated to be ~0.3 mm d$^{-1}$ during winter (Singh et al., 2020). Sublimation rates calculated using bias-corrected ERA5 data for Dokriani (~1.2 mm d$^{-1}$) and Chhota Shigri (~0.7 mm d$^{-1}$) glaciers were

also similar to our study. Sublimation rates during the summer-monsoon season, in general, was lower than that of winter (Table 4; also Litt et al., 2019), which could be due to the warm and moist atmospheric conditions driven by the ISM. This also occurs in the westerlies dominated region such as Muztag Ata No. 15 Glacier in Pamir (Zhu et al., 2018) and the area of transition between the westerlies- and monsoon-dominated climate regimes such as Xiao Anglong Glacier in Upper Shiquanhe region (Zhu et al., 2021b). Despite high summer-monsoon humidity, sublimation is higher at higher altitude sites, such as in

the East Rongbuk Glacier site (6523 m a.sl.). This is most likely a result of the strong winds and low air vapour pressure at very high altitudes, which promote sublimation. The high moisture from ISM also impacts Tibetan glaciers, particularly those located in the northern slopes of the Himalaya (Zhu et al., 2021a; Liu et al., 2021) and central Tibet (Mölg et al., 2012; Li et al., 2018). The ratio of summer (June-September) sublimation to winter (October-May) is larger in the monsoon-dominated region such as Parlung No. 4 and Zhadang glaciers than that in the westerlies and the transition area, e.g., Xiao Anglong Glacier

(Zhu et al. 2020). In the Nepalese central Himalaya, we note a higher sublimation value of 2.4 and 1.8 mm d$^{-1}$, respectively on the Yala Glacier during the post- and pre-monsoon seasons (Table 4). Litt et al. (2019) also reported a significantly higher sublimation rate of 7.1 and 1.9 mm d$^{-1}$, respectively during post- and pre-monsoon seasons on the Mera Glacier in Nepal. Such higher sublimation rates on Yala and Mera glaciers are unique, particularly during post- and pre-monsoon seasons when air vapour pressure/specific humidity is higher than that in winter season (Shea et al., 2015; Perry et al., 2020). Nevertheless, such

higher sublimation can also be partially attributed to snow blowing/redistribution at such high-altitude sites (Barral et al., 2014; Wagnon et al., 2013; Huintjes et al., 2015b). Overall, dry air, low atmospheric pressure and high wind speeds are suitable conditions for sublimation, as reported from various high-altitude sites in the HMA (Matthews et al., 2020; Litt et al., 2019; Stigter et al., 2018; Zhu et al., 2018) and everywhere in the world (Wagnon et al., 1999; Cullen et al., 2007; Fyffe et al., 2021).

**Table 4. Compilation of sublimation rate across the HMA region. '*' refers to the evaporation values. 'Do' refers to the same method as in the row immediately above.**

| Site | Altitude (m a.sl.) | Region | Period of observation | Season approx. to Chhota Shigri | Surface | Method | $S$ (mm d$^{-1}$) | RH (%) | u (m s$^{-1}$) | Reference |
|---|---|---|---|---|---|---|---|---|---|---|
| **Tibetan Plateau** | | | | | | | | | | |
| Zhadang | 5665 | Nyainqentanglha Shan | 1 October to 31 May, 2008-2013 | Winter | Glacier-wide | Bulk-aerodynamic | 0.5 | 44 | 3.6 | Zhu et al. (2018) |
| Muztag Ata No. 15 | 4400 | Eastern Pamir | 1 October to 31 May, 2008-2013 | Winter | Glacier-wide | Do | 0.7 | 42 | 6.4 | Zhu et al. (2018) |
| Parlung | 4800 | Southeast TP | 1 October to 31 May, 2008-2013 | Winter | Glacier-wide | Do | 0.4 | 64 | 3.4 | Zhu et al. (2018) |
| Muji | 4685 | Northeast Pamir | 1 October to 31 May, 2011- 2017 | Winter | Glacier-wide | Do | 0.5 | 50 | 4 | Zhu et al. (2020) |
| Xiao Anglong | 5141 | Upper Shiquanhe (West Tibet) | 1 October to 31 May, 1968- 2019 | Winter | Glacier-wide | Do | 0.4 | ~35 | ~5 | Zhu et al. (2021b) |

| | | | | | | | | | | |
|---|---|---|---|---|---|---|---|---|---|---|
| Qiangtang No. 1 | 5882 | Inland TP | 1 October to 31 May, 2012-2016 | Winter | Glacier-wide | Do | 0.4 | 46 | 6.8 | Li et al. (2018) |
| Guliya Ice Cap | 6000 | Kunlun Shan | 1 October to 31 May, 2015-2016 | Winter | Glacier-wide | Do | 0.3 | 67 | 7.9 | Li et al. (2019) |
| Dongkemadi | 5600 | Central TP | 7 October 1992 to 4 May 1993 | Winter | Glacier ELA | Do | 0.2 | - | 4.3 | Liang et al. (2018) |
| August-one | 4817 | Qilian Mountains | Jan-May, Oct-Sept, 2016-2020 | Winter | Glacier | Do | 0.4 | 68 | 6.9 | Guo et al. (2021) |
| **Himalaya** | | | | | | | | | | |
| Pindari | 3750 | Central Himalaya | December 2016 to February 2017 | Winter | Medial moraine | Monin-Obukhov theory | ~0.3 | 55 | 1.2 | Singh et al. (2020) |
| Dokriani | ERA5 grid point | Do | 1 November 1979 – 30 October 2020 | Winter | Glacier-wide | Bulk-aerodynamic | ~1.2 | ~45 | ~7 | Srivastava and Azam, 2022 |
| Yala | 5350 | Do | 15 October 2015 to 20 April 2017 | Winter | Glacier/ablation zone | Eddy-covariance | 1 | ~40 | ~2.5 | Stigter et al. (2018) |
| Yala | 5330 | Do | 1 October to 15 November, 2012-2017 | Post-monsoon | Glacier/ablation zone | Bulk-aerodynamic | 2.4 | ~49 | ~1.8 | Litt et al. (2019) |
| Yala | 5330 | Do | 10 May to 5 June, 2012-2017 | Pre-monsoon | Glacier/ablation zone | Do | 1.8 | ~77 | ~1.9 | Do |
| Mera | 5360 | Do | 1 October to 15 November, 2013-2016 | Post-monsoon | Glacier/ablation zone | Do | 1.9 | ~46 | ~2.8 | Do |
| Mera | 5360 | Do | 10 May to 5 June, 2013-2016 | Pre-monsoon | Glacier/ablation zone | Do | 3.3 | ~72 | ~2.3 | Do |
| Lirung | 4250 | Do | 26 September to 12 October 2016 | Post-monsoon | Glacier debris | Eddy-covariance | 1.8-2.8[*] | ~60 | ~3 | Steiner et al. (2018) |
| South Col, Everest | 7945 | Do | 22 May to 31 October 2019 | Summer-monsoon | Ice-rock surface | Bulk-aerodynamic | ~0.8 | ~60 | 6.3 | Matthews et al. (2020) |
| East Rongbuk | ~6500 | Do | 28 April to 2 May 2008 | Pre-monsoon | Glacier | Lysimeter | 1.9 | - | - | Yang (2010) |
| East Rongbuk | 6523 | Do | 1 May to 22 July 2005 | Summer-monsoon | Glacier | Bulk-aerodynamic | 0.05-1.2 | 60 | 4.2 | Liu et al. (2021) |
| Xixibangma | 5900 | Do | 23 August to 29 September 1991 | Summer-monsoon | Glacier | Calculated | 0.02 | 36 | 5.9 | Aizen et al. (2002) |
| Naimona'nyi | 5543 | Do | 1 October 2010 to 31 May 2018 | Winter | Glacier-wide | Bulk-aerodynamic | 0.6 | 34 | 5.5 | Zhu et al. (2021a) |
| Chhota Shigri | 4670 | Western Himalaya | 1 Dec 2012 to 29 Jan 2013 | Winter | Glacier/ablation zone | Do | 0.8 | 44 | 4.9 | Azam et al. (2014a) |
| Chhota Shigri | ERA5 grid point | Do | 1 October 1979 – 30 September 2020 | Winter | Glacier-wide | Bulk-aerodynamic | 0.7 | ~40 | ~5.7 | Srivastava and Azam, 2022 |
| Chhota Shigri | 4863 | Do | 1 December to 30 April, 2009-2020 | Winter | Seasonal snow on moraine | Do | 1.1 | 43 | 5 | This study |

## 5.4 Sublimation fraction to winter snowfall and its importance

Sublimation is a substantial component of the surface mass balance and hydrological cycle in the HK glacierised catchments (Azam et al., 2021). Sublimation fractions have been reported in different ways, such as fractions of winter/annual snowfall or fractions of total ablation/mass balance. The cumulative sublimation at the AWS-M ranges from 85 mm w.e. to 172 mm w.e., with a long-term mean of 145 ± 25 mm w.e. for DJFMA during 2009-2020 (Table 3). The cumulative snowfall ranges from 402 mm w.e. to 971 mm w.e., with a mean of 574 ± 187 mm w.e. recorded at the glacier base camp and Keylong station (reliability of Keylong's precipitation data is discussed in Sect. 3.1) for DJFMA during 2009-2020 (Table 2). The cumulative sublimation loss accounts for 16-42% of the fraction of winter snowfall at the AWS-M site (Table 2). This mass loss is

substantial compared to other parts of the HK region. For example, in the central Himalayan Yala Glacier, sublimation loss was 21% of the snowfall for one winter season (Stigter et al., 2018). Similarly, sublimation loss was about 14-18% of the total snowfall in the Pheriche sub-catchment of the Dudh Koshi Basin in Nepal, based on distributed glaciohydrological model (Mimeau et al., 2019). Based on satellite-derived datasets, Gascoin (2021) showed that the sublimation ratio to snowfall can exceed 60% in the high-altitude areas in the north-western part of the Himalaya, e.g., Ladakh and Karakoram. The mean annual glacier-wide sublimation losses were around 20% of total annual ablation on Dokriani and Chhota Shigri glaciers over 1979-2020 based on glacier-wide SEB analysis using bias-corrected ERA5 datasets (Srivastava and Azam, 2022). In the Chinese Altai Mountain's Irtysh River Basin, sublimation accounts for 19% of the snowfall estimated through a physically based snow model (Wu et al., 2021). In the Tibetan Plateau, at the Zhadang Glacier, sublimation loss was 26% of the total mass loss annually (Huintjes et al., 2015a). At the August-One Glacier in the Qilian Mountains, evaposublimation accounts for 15% of the annual precipitation, with the major part during winter periods (Guo et al., 2021). In some sites of the Tibetan Plateau, sublimation fraction is considerably higher. For example, in the Muji Glacier in Pamir, the cold season's evaposublimation loss is > 70% of the corresponding snowfall (Zhu et al., 2020). In the Kunlun Mountains on the Guliya Ice Cap, glacier-wide sublimation loss was ~120% of the winter snowfall and ~50% of the annual snowfall (Zhu et al., 2022). On the Qiangtang No. 1 Glacier in inland Tibet, the sublimation and evaporation loss fraction were about 65-169% of the snowfall during 2012-2016, which is a significantly higher mass loss than gain (Li et al., 2018). Such a higher sublimation fraction at the Qiangtang No. 1 Glacier during non-melt seasons was associated with high wind speed (~7 m s$^{-1}$), lower $RH$ (~46%) and low annual precipitation (362-614 mm). This supports that the dry and windy environment fosters sublimation. Although there are limited observations available from various parts of the Himalaya and HMA, these observations show that the sublimation fraction to winter/annual snowfall/precipitation is higher in the north-western part of the HK and western Tibet (e.g., Zhu et al., 2020; Gascoin, 2021). This is likely due to the atmospheric condition of the north-western part of the HK and western Tibet which is drier than the eastern and central Himalaya. Dry atmospheric conditions favour higher sublimation than the wet due to high near-surface humidity gradients.

Sublimation is the largest mass loss component during winter. Nonetheless, the sum of winter snowfall may have significant uncertainties considering the under-catch of solid precipitation (Collier and Immerzeel, 2015; Shea et al., 2015; Doblas-Reyes et al., 2021) by the Geonor gauge at the glacier base camp and Keylong station due to strong winds. For example, the snowfall catch efficiency of a Geonor T-200B equipped with a single-Alter windshield (the one functional at the glacier base camp) could be about 50% or less at a wind speed of about 5 m s$^{-1}$ or higher (Wolff et al., 2015; see their Fig. 5). Despite the uncertainty in winter snowfall, our results indicate that sublimation loss during DJFMA is a significant component of winter mass distribution. Therefore, it is crucial to include sublimation in future surface mass balance and hydrological modelling in the region. We also stress the importance of reporting the sublimation estimates in a consistent and widely acceptable manner so that they can be directly compared between sites.

## 6 Conclusions and perspectives

In this study, we presented an 11-year record of observed meteorology, SEB and sublimation for DJFMA at 4863 m a.s.l. on the Chhota Shigri Glacier moraine in the western Himalaya. We investigated the role of turbulent heat fluxes in the SEB along with the influence of cloud cover and the sublimation to know their importance in winter mass distribution during 2009-2020.

The net short-wave radiation was the primary energy source of SEB. At the same time, turbulent heat fluxes ($H + LE$) significantly sink the energy, resulting in negative residual energy ($F_{surface}$) at the snow surface throughout DJFMA. Although net short-wave radiation was the largest contributor in the SEB across the HMA, we found a significant role of latent heat flux, contributing > 60% during the winter months. The moisture availability primarily controls the magnitude of latent heat flux, with considerable influence from snow surface temperature and wind speed. Interestingly, we found that the strong and cold winds, probably from the WDs storms, acts as an impediment of latent heat flux at the AWS-M site by setting up high moisture and cold temperature regime.

The large variability in the SEB components was directly related to cloud cover, which primarily affects incoming short-wave radiation reducing by 70% and incoming long-wave radiation raising by 25%. The cloud cover also controls the meteorological condition favourable for turbulent heat fluxes and reduce their magnitude by larger than 60%. The mean daily sublimation at the AWS-M was about three times lower on cloudy conditions than clear-sky due to the low incoming short-wave radiation and subsequent alteration in near-surface meteorological conditions. The mean daily sublimation was similar to the sublimation rates of other HK and HMA glaciers during winter. The vertical gradient of temperature and moisture along with surface temperature and wind speed emerged as the best predictors of sublimation based on the multiple linear regression analysis. The sensitivity analysis showed that sublimation is most sensitive to the changes in $z_{0m}$ and $T_s$ suggesting it is crucial for accurate SEB and sublimation. It is, however, slightly less sensitive to $T_{air}$ but it remains a matter of concern from a future warming perspective.

The cumulative DJFMA sublimation was $145 \pm 25$ mm w.e. a$^{-1}$, corresponding to 16-42% of the fraction of winter snowfall at the AWS-M site, which is relatively higher than observed in other sites across the HK region, with considerable interannual variations and is lower than a few of the Tibetan glacier sites. Hence, sublimation emerged as one of the significant mass balance components during winter, especially in a dry-cold-windy environment. However, sublimation estimates, and winter snowfall could be uncertain in the high-mountain sites, considering their sensitivity to meteorological forcing, surface roughness length, sensor inaccuracies and calculation errors.

Given the limitations, this 11-year dataset demonstrates how individual glacier-based long-term observations/studies can improve our understanding of local-scale meteorological factors that are affecting SEB and sublimation in the HK region. This study underscores the need for extensive measurements of high-quality, on-glacier weather data observation using the eddy-covariance technique and snowfall for robust region-wide modelling, and inclusion of sublimation schemes in glaciohydrological models.

**Code and Data availability**

The codes for SEB calculation and generating the figures are available at https://github.com/arindan/Winter-sublimation-at-the-Chhota-Shigri-Glacier-India (https://doi.org/10.5281/zenodo.6804947; Mandal et al., 2022). AWS-M data used in this study is available on request from the corresponding author.

**Supplementary Material**

The supplement related to this article is available online at:

**Author Contribution**

AM, TA, MFA and PW conceptualised the study. ALR supervised the study. AM performed the analysis, developed the figures, and wrote the paper. MS helped in SEB calculations. CS partly compiled the existing sublimation studies across the HMA. All authors contributed significantly to preparing the draft manuscript and discussion and supported the data analysis.

**Competing Interests**

The authors declare that they have no conflict of interest.

**Acknowledgements**

We thank Jawaharlal Nehru University, New Delhi, for providing all the facilities to carry out this work. The authors are greatly thankful to the field assistant B. Adhikari and other field supporters who have taken part in successive field trips in harsh conditions. The funding agencies and project collaborators who fully and partially supported this work are the
705 Department of Science and Technology (Govt. of India), IFCPAR/CEFIPRA, INDICE, GLACINDIA and CHARIS, MoES, SAC-ISRO. Thanks to Etienne Berthier for the Pléiades image provided under the Pléiades Glacier Observatory (PGO) initiative of the French Space Agency (CNES). AM is grateful to UGC-RGNF and DAAD Bi-nationally Supervised PhD Fellowship (Germany) for providing financial support for his PhD. MFA acknowledges the research grant from INSPIRE Faculty award (IFA-14-EAS-22) and Space Application Centre (ISRO). We thank Masashi Niwano (editor), Jakob Steiner and
710 the other (anonymous) reviewer for helping to improve the manuscript with their comments and suggestions.

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
