# Peer review of "11-year record of wintertime snow surface energy balance and sublimation at 4863 m a.s.l. on Chhota Shigri Glacier moraine (western Himalaya, India)"

_The Cryosphere, 2021_

## Referee Comment (RC1)

**Review of manuscript:**

**"11-year record of wintertime snow surface energy balance and sublimation at 4863 m a.s.l. on Chhota Shigri Glacier moraine (western Himalaya, India)" by Arindan Mandal and co-authers**

Submitted to The Cryosphere

General comments:

This paper analyzes the winter meteorological data and surface energy balance (SEB) on a lateral moraine of the Chhota Shigri Glacier in the western Himalaya during 2009-2019, and then explored the effects of cloud cover on winter energy balance and sublimation on that site. In addition, this paper presents long-term glacio-meteorological data in the western Himalayas, which is very important for studying the glacier mass balance changes on the Tibetan Plateau and the surrounding areas. This is an interesting paper, but needs major revisions before can be accepted for publications in TC. I have several comments that the authors should address.

Main comments:

1. Introduction: There are some studies discussing the energy balance and mass balance around HK regions and other regions on the Tibetan Plateau in recent years, such as Pamir and Tibet. Although the authors reviewed some studies, it is relatively simple. The authors should review more recent studies about energy balance and mass balance around the Tibetan Plateau, and pointed out the limitations of these studies.

2. I do not find how authors calibrate or evaluate their modeled results in this work. The parameters in the energy balance always need to be calibrated by some measured values. For example, the selected surface roughness lengths for momentum, temperature, and humidity, and the different formula of turbulent heat will impact the modeled H and LE. If the modeled results are calibrated, the data will be more credible. Why the author does not select surface temperature to

calibrate their model using the iteration method. In this work, the author can deduce that there is no snow cover at the AWS-M site when albedo is smaller than 0.4. This data can also be used to calibrate their modelled values. In addition, it seems that there are few studies about the glacier energy balance which delete G in their model. The AWS-M site remains snow-covered during winter and bare sand/sediment exposed during summer. Whether bare sand below the snow can provide more energy to heat the snow when compared to glacier ice below the snow? Or the author just focuses on the energy feature.

3. There are so many results in section 4. The author could shorten this section, because some studies have introduced the meteorological data at AWS-M in that glacier (Azam et al., 2016). Are there any special features from your data? Those special features are important. In addition, I hope that the author can discern the timescales for their results, such as diurnal cycle, seasonal cycle, and interannual timescales.

4. Line 23-25: The author does not discuss the influence of mid-latitude western disturbances on sublimation in the main text. The author can use the Reanalysis data (such as geopotential height and wind fields at 500 hPa or other heights from ERA5 or JRA55) to obtain the direct knowledge of circulation which can impact the sublimation and energy balance.

5. The author examines the role of cloud cover on SEB and turbulent heat fluxes based on clear-sky conditions and overcast conditions. However, this can be finished by using just two years of data. The relationship between CF and sublimation is small (Table 4). Thus, CF (or Sin) is not the main factor causing the interannual changes in sublimation in winter during 2009-2020. I strongly recommend that authors analyze the factors which control the interannual changes in sublimation in winter during 2009-2020 through correlation analysis. The author can explain interannual changes in sublimation from the view of energy balance. And the author should analyze the relationships between RH and sublimation, between albedo and sublimation, between Sin and sublimation, between Sout and sublimation, between Lin and sublimation, Lout and sublimation between D and Tair, between D and Ts,

and between D and RH. I guess that albedo is an important factor that contributes to the interannual changes in sublimation by changing Ts. The concrete results are depending on your further analysis.

6. Discussion: I sometimes feel confused about the sentences in the discussion. Take section 5.3 for example. The author said that sublimation during the summer-monsoon season was lower, which could be due to the ISM-driven warm and moist atmosphere in the southern slope of the HK region. However, sublimation is higher at very high altitudes despite high summer-monsoon humidity, e.g., East Rongbuk Glacier site (6523 m a.sl.). What is the main point of the author? When author compared their study with other studies, the author should note the spatial and temporal scales. Some studies used the glacier-wide values, while others used point values. Some studies used the low-altitude values, while others used the high-altitude values. Some studies used the annual values, while others used winter values. These data with different scales are incomparable. The author should select these data carefully.

Minor comments:

Line 32: wind-driven transport can cause accumulation in some sites.

Line 121-123: How do you get albedo in the night? Thus, what is your surface albedo threshold value in the night which is used to discern snow or bare-ground?

Line 153: Please explain the physical significance of Fsurface. If Fsurface is larger than 0, does melt occur at that time?

Line 209-210: Can you analyze the difference between infrared measured Ts and Ts derived from Lout? Please list the figure. Is the emissivity of bared-ground similar to that of snow cover? This is important for the author to calculate Ts from Lout.

Line 312-313: Which components in Rnet are more important in playing an essential role in governing the turbulent fluxes? And the author should indicate the timescale.

Line 313-314: I can not understand this sentence.

Line 325: What do you mean about the different colors of lines in Figure 6?

Line 359. Please add the "in the daytime" in the title of section 4.5.

Line 363-364 Why precipitation is higher in February and March than in January and April? High precipitation always means high cloud cover. This is different from your results of CF.

Line 376: 3 times lower?

Line 423: 145 ± 25 mm w.e. a-1?

Section 4.7: There is no section 4.7.2 in this part. The author can merge section 4.7 and section 4.7.1 as one part.

Line 451-452: I can not agree with the author, because we can not find that low Tair (-5°C and -10°C) corresponds to high Ts (0°C and -10°C) for the same time. From figure 14b, we can only find that sublimation was the larger when Tair ranged between -5°C and -10°C (compared to Tair in other values). This is similar to the Ts. Thus, the content in Line 451-452 is not correct.

Line 480-481: What is your timescale?

Line 544: Do you want to say that sublimation during the summer-monsoon season was lower than that during winter?

Line 545-547: The studies of Mölg et al. (2012) and Li et al. (2018) are in the south and central Tibet, respectively. They do not study the glaciers in the northern slope of the HK region.

Line 547-548: Can you explain the phenomenon that you found in these sentences?

Line 548-549: I do not find that the moisture content is relatively higher during post- and pre-monsoon on the Mera Glacier than that in winter in Table 5. And the altitudes are significantly different between the post- and pre-monsoon periods.

Line 550-551: What is the cause for the differences that the authors found in this sentence?

Line 553-555: I cannot understand what you want to say.

Line 560-564: These sentences have no relationship with the title 'Sublimation fraction to winter snowfall and its importance'.

Line 569: Why sublimate is higher in the northwestern part of the HK than that in the other parts of the HK region?

Line 579-580: Such a higher sublimation fraction? You mean that the sublimation

fraction is higher on Qiangtang No 1 Glacier than other glaciers on the TP. Have you compared the meteorological data at the Qiangtang No. 1 Glacier to that on other glaciers?

Line 610-611: This result disagreed with your description in section 5.4. Sublimation fraction to winter snowfall is higher on Qiangtang No 1 Glacier than that on Chhota Shigri Glacier.

Line 620: There are more than 10 published works about Chhota Shigri Glacier. However, the meteorological data for that glacier is still not open to scientists in the world.

---

## Author Comment (AC2)

**Response to Reviewer 1**

Below we provide our responses **(in red text)** point-by-point to each comment from the reviewer **(in black text)**. *Italic texts* are used to highlight specific changes in the updated manuscript.

**General comments:**

This paper analyzes the winter meteorological data and surface energy balance (SEB) on a lateral moraine of the Chhota Shigri Glacier in the western Himalaya during 2009-2019, and then explored the effects of cloud cover on winter energy balance and sublimation on that site. In addition, this paper presents long-term glacio- meteorological data in the western Himalayas, which is very important for studying the glacier mass balance changes on the Tibetan Plateau and the surrounding areas. This is an interesting paper, but needs major revisions before can be accepted for publications in TC. I have several comments that the authors should address.

We sincerely thank Reviewer 1 for evaluating our manuscript and giving suggestions to improve the quality of the manuscript. We have responded to your specific comments and outlined the changes that we have made in the revised manuscript. If the reviewers and the editor are satisfied with our responses, we will submit our revised manuscript. Below, we have highlighted (point-wise) the major revisions that we have made in the revised manuscript in response to your main comments:

- We have shortened the meteorological condition section by reducing the texts, figures (part of the figures have been shifted to the supplementary material) and tables in respective sections.

- Reorganized the presentation of meteorological and surface energy balance (SEB) analysis to account for different temporal scales, such as daily, hourly, seasonal, and inter-annual.

- Incorporated a large-scale wind/moisture circulation analysis using ERA5 500 hPa datasets to understand the influence of western disturbances (WDs) on sublimation.

- Sub-hourly and inter-annual correlation analysis, as well as multiple regression analysis of sublimation and meteorological variables, were included. Further, the discussion and interpretation were significantly revised, with a focus on sublimation factors.

- AWS data used in this study, codes for SEB calculation and figures are made open to the global community through open repository with DOI.

**Main comments:**

1. Introduction: There are some studies discussing the energy balance and mass balance around HK regions and other regions on the Tibetan Plateau in recent years, such as Pamir and Tibet.

Although the authors reviewed some studies, it is relatively simple. The authors should review more recent studies about energy balance and mass balance around the Tibetan Plateau, and pointed out the limitations of these studies.

We acknowledge that we reviewed only a few studies related to glacier surface energy balance (SEB) from the Pamir and Tibet regions in the Introduction section. It is because our main focus is to address the research gaps related to sublimation estimation and its role in glacier mass balance in the Himalaya-Karakoram (HK) region. Therefore, we mainly highlighted the importance of turbulent heat fluxes, understanding gaps for sublimation and its estimation methods used in the HK region and skipped/avoided to discuss findings/gaps related to glacier mass and energy balance.

To make the Introduction section more holistic and inclusive, we have included some of the recent glacier SEB (e.g., Li et al., 2019; Zhu et al., 2020) and sublimation (e.g., Guo et al., 2021) related studies from the Pamir and Tibet regions. Newly incorporated texts in the Introduction read as:

*Line No. 41-46:*
*'However, SEB studies on Tibetan glaciers are relatively more abundant (~17 investigated glaciers/ice-covered sites; Table S1), including direct turbulent heat flux measurements (Yang et al., 2011; Zhu et al., 2018) except in Pamir and Kunlun Mountains (Zhu et al., 2020). Glaciers in the Pamir area are extreme continental type, with cold temperature and low annual precipitation (Li et al., 2019), thus their SEB characteristics are expected to behave differently compared to the majority of HK glaciers which are alpine type, with relatively higher precipitation and temperature.'*

*Line No. 65-67:*
*'In the Muji Glacier in northeast Pamir, the cold season's evaposublimation loss is > 70% of the corresponding snowfall (Zhu et al., 2020). In the Qilian Mountains at the August-one Glacier (north-east Tibetan Plateau), evaposublimation loss is lower but accounts for about 15% of annual precipitation (Guo et al., 2021).'*

In addition, in Table 4 (revised manuscript) and the sublimation section (Sect. 5.3 and 5.4 in the revised manuscript), we specifically reviewed several studies from the Pamir and Tibet regions, comparing sublimation rates across the HK and Tibetan glacierised regions. Some of the texts are as follows:

*Line No. 720-723 (Sect. 5.3):*
*'Sublimation rates during winter were slightly higher in the Pamir region, e.g., Muztag Ata No. 1 (Zhu et al., 2018) and Muji site (Zhu et al., 2020) compared to the inland/central Tibet region, e.g., Qiangtang No. 1 (Li et al., 2018) and Dongkemadi site (Liang et al., 2018). This is likely*

*due to the relatively dry atmospheric condition in the Pamire region than the central or eastern parts of Tibet (Table 4; also Liu et al., 2020).'*

*Line No. 761-776 (Sect. 5.4):*
*'In the Tibetan Plateau, at the Zhadang Glacier, sublimation loss was 26% of the total annual mass loss (Huintjes et al., 2015a). At the August-one Glacier in the Qilian Mountains, evapo-sublimation accounts for 15% of the annual precipitation, with the major part during winter periods (Guo et al., 2021). In some sites of the Tibetan Plateau, sublimation fraction was considerably higher. For example, in the Muji Glacier in Pamir, cold season's evaposublimation loss was > 70% of the corresponding snowfall (Zhu et al., 2020). In the Kunlun Mountains at Guliya Ice Cap, glacier-wide sublimation loss was ~120% of the winter snowfall, whereas ~50% of the annual snowfall (Zhu et al., 2022). At the Qiangtang No 1 Glacier, inland Tibet, the sublimation and evaporation loss fraction was about 65-169% of the snowfall during 2012-2016 (Li et al., 2018). Such a higher sublimation fraction at the Qiangtang No 1 Glacier during non-melt seasons was associated with high wind speed (~7 m s$^{-1}$), lower RH (~46%) and low annual precipitation (362-614 mm). This supports that the dry and windy environment fosters sublimation in the HK region. Although there are no sufficient observations available from various parts of the Himalaya or HMA, sublimation fraction to snowfall/annual precipitation is higher in the northwestern part of the HK and western Tibet (e.g., Zhu et al., 2020; Gascoin, 2021). This is likely due to the atmospheric condition of the northwestern part of the HK and western Tibet which is drier than eastern and central Himalaya. Dry atmospheric conditions favor higher sublimation than the wet due to high near-surface humidity gradients.'*

2. I do not find how authors calibrate or evaluate their modeled results in this work. The parameters in the energy balance always need to be calibrated by some measured values. For example, the selected surface roughness lengths for momentum, temperature, and humidity, and the different formula of turbulent heat will impact the modeled H and LE. If the modeled results are calibrated, the data will be more credible. Why the author does not select surface temperature to calibrate their model using the iteration method. In this work, the author can deduce that there is no snow cover at the AWS-M site when albedo is smaller than 0.4. This data can also be used to calibrate their modelled values. In addition, it seems that there are few studies about the glacier energy balance which delete G in their model. The AWS-M site remains snow-covered during winter and bare sand/sediment exposed during summer. Whether bare sand below the snow can provide more energy to heat the snow when compared to glacier ice below the snow? Or the author just focuses on the energy feature.

We sincerely acknowledge the concern of Reviewer 1 for the calibration/validation of our SEB calculation, especially the bulk method for turbulent heat flux calculation.

We would like to mention that, in this work, we used the measured surface temperature ($T_s$; through an infrared radiometer) as an input to calculate the turbulent heat fluxes ($H$ and $LE$)

following the bulk method; therefore, we cannot compare the simulated $T_s$ and measured $T_s$ to calibrate/validate the bulk methods' performance, which is usually used for evaluating glacier SEB/bulk models.

Concerning the credibility of our SEB and bulk modelling approach, this method has been successfully applied on this glacier in an on-glacier site SEB experiment (Azam et al., 2014a) at 4670 m a.s.l. (~1 km away from the AWS-M; our study site). The SEB model result was validated using the observed surface melt ($r^2 = 0.98$), and also a significant correlation ($r^2 = 0.96$) was observed between the simulated $T_s$ and observed $T_s$ (Azam et al., 2014a).

To incorporate your concern, we tried to validate the sublimation rates using SR-50A data (snow height), which was available for a shorter period (Dec 2009 to Apr 2015). Our plan was to filter snow depth/height (SR-50A) data for periods with no snowfall longer than minimum one week and compare calculated sublimation rates and observed snowpack thickness change over such periods. However, due to inconsistency in the measured precipitation records from the Geonor gauge, the available data for this analysis was only for two years (DJFMA of 2012/13 and 2014/15) (Sect. 3.1). We examined SR-50A data for those two years, but we couldn't find any long enough periods without snowfalls. Figure R1 (below) shows Geonor precipitation data from DJFMA of 2012/13 and 2014/15. Therefore, comparing the SR-50A snow thickness change data to sublimation rates was not appropriate in this case. For the years when Geonor data was not available, we used daily precipitation from the neighbouring Indian Meteorological Department (IMD) Stations: Bhuntar (~50 km) and Manali (~40 km). In those station data also we do not find any long enough period without snowfall (figure not shown here). No daily precipitation data was available from the Keylong, closest IMD station (~60 km) from our study site. There was a possibility to apply the measured/assumed snow density to convert height change into mass loss and compare it with corresponding sublimation loss. However, considering data limitation, we are not in a position to conduct a direct validation of our bulk model.

[Figure]

**Figure R1.** Daily Geonor precipitation (measured at Chhota Shigri glacier base camp at 3850 m a.s.l.) and SR-50A (AWS-M; 4863 m a.s.l.) height change during DJFMA 2012/13 and 2014/15. Data shown only for DJFMA period.

The primary focus of this work is to estimate sublimation directly derived from *LE*. To analyse and quantify the meteorological variable's sensitivity to sublimation and a possible uncertainty in our bulk model, we conducted a sensitivity analysis of the calculated sublimation (Sect. 5.2), where we perturbed the calculated sublimation by changing meteorological variables (e.g., $T_{air}$ by ± 1°C, $T_s$ by ± 1°C, wind speed (*u*) by ± 10% and *RH* by ± 10%) and surface roughness lengths (0.0005 m, 0.002 m, 0.003 m and 0.004 m) to evaluate the range of sublimation.

In addition, the bulk method was compared with the direct eddy-covariance method over the snow surface at the Yala Glacier (Central Himalaya, Nepal) to evaluate the performance of the bulk method (Stigter et al., 2018). They found a good agreement between eddy-covariance and bulk method ($r^2$ = 0.88) in estimating sublimation rates, which shows the reliability of the bulk method over snow surface in the Himalayan site. However, as you also suggested that roughness lengths in the bulk model is very crucial to get an accurate result. Considering this and for better

accuracy, we have used the previously calculated snow surface roughness lengths already obtained on this glacier (0.001 m; Azam et al., 2014a), which was calculated using wind measurements at two different levels following a conventional logarithmic profile (e.g., Moore, 1983). These aspects are discussed in Sect. 3.2.2.

Based on the aforementioned discussion, we added a new sentence in the method section (Sect. 3.2.2. revised manuscript) highlighting the limitations of our bulk model validation. The new sentence reads as:

*Line No. 233-235:*
*'Due to data limitations, direct validation of the bulk model used in this study was not possible, but we trust our results based on Azam et al (2014a)'s bulk model validation on this glacier in 2012/13 and proved to deliver robust results compared to observations. We also conducted a sensitivity analysis of our bulk model including surface roughness lengths.'*

Regarding the ground/subsurface heat flux ($G$), we calculated $G$ at the AWS-M site following the method proposed by You et al. (2014) and Luce and Tarboton (2010). The diurnal variation of $G$ is shown below through a figure (Fig. R2) along with other major energy fluxes. The results show that $G$ is negligible about -0.4 ± 4.4 W m$^{-2}$ for DJFMA (2009-2020; n = 73624 half-hourly data points) compared to other energy fluxes. That means there is no significant energy coming from the ground/bare surface. In addition, considering the inadequate measurement of the subsurface heat measurements and relative information of $G$ in the HK region (Stigter et al., 2021), we neglected it in our SEB calculation. Also, since we focus on sublimation and its drivers/importance, $G$ is beyond the scope of this study.

[Figure]

**Figure R2.** Mean diurnal cycle of *G* (following You et al., 2014 and Luce and Tarboton, 2010) at the AWS-M for DJFMA (2009-2020) along with $R_{net}$, $H + LE$, and $F_{surface}$.

3. There are so many results in section 4. The author could shorten this section, because some studies have introduced the meteorological data at AWS-M in that glacier (Azam et al., 2016). Are there any special features from your data? Those special features are important. In addition, I hope that the author can discern the timescales for their results, such as diurnal cycle, seasonal cycle, and interannual timescales.

We thank the reviewer for the concern. To incorporate the reviewer's suggestion and shorten the meteorological condition section, we have revised most of the figures in this section. For instance, a part of the figures (e.g., Figure 2, 3, 5C and G, and 6D, 7B in original manuscript) have been shifted to the supplementary material. Table 2 (original manuscript) has also been moved to the supplementary material. Below we present the revised figures (e.g., Figure 2, 3, 5 and 6 in revised manuscript) for your reference. We merged Sect. 4.2 and 4.3 (original manuscript) into a single Sect. 4.1 (revised manuscript). We also reorganized the presentation of meteorological and surface energy balance (SEB) analysis to account for different temporal scales. Such as in Sect. 4.2 (revised manuscript) we presented the diurnal cycle of all meteorological and SEB variables, and seasonal and interannual variation of SEB components in Sect. 4.3 (revised manuscript).

We would also like to mention that Azam et al. (2016) did not focus on the SEB related details, for example, they discussed fewer variables (e.g., $T_{air}$, *RH*, *u*, $S_{in}$ and $S_{out}$). $T_s$, *albedo*, *q* and *CF* were missing in their study, which are important variables to understand SEB characteristics. Furthermore, the meteorological conditions presented by Azam et al. (2016) were based on only four years of datasets (2009-2013), but we updated it using 11-years long datasets in this work (2009-2020).

To discern the timescale of meteorological/SEB characteristics, we have performed the analysis at various temporal scales as suggested by the reviewer. For instance, first, in Sect 4.1 we presented the daily variations and ranges of all meteorological/radiation components. Second, in Sect. 4.2 we presented the diurnal cycle of all meteorological and SEB variables. Third, in Sect. 4.3 we presented the seasonal and interannual variation of SEB components with their statistical correlations. Further, in Sect. 4.6 we analysed sublimation considering various temporal scales, e.g., daily, sub-hourly, seasonal and interannual. Since sublimation and turbulent fluxes are our main interests, we investigated them in Sect. 4.4 and 4.5, focused on the impact of cloud cover in sublimation.

We invite the reviewer to go through the reorganised sections in the revised manuscript.

[Figure]

**Figure 2 (revised manuscript).** Monthly climatology of air ($T_{air}$) and surface temperature ($T_s$), relative humidity ($RH$), wind speed ($u$) and surface albedo ($\alpha_{acc}$) at the AWS-M for 2009-2020. DJFMA (1 December to 30 April) period is highlighted with a light blue rectangle in each panel. The shades around the line and scatter points correspond to one standard deviation (SD). Dashed lines in panel E refer to snow-surface albedo ($\alpha_{acc}$ = 0.4; red line) for SEB analysis and bare-surface albedo ($\alpha_{acc}$ = 0.2; black line). Daily values of $T_{air}$, $T_s$, $RH$, $u$ and albedo for the study period are shown in Fig. S2.

[Figure]

**Figure 3 (revised manuscript).** Windrose of the AWS-M for DJFMA (2009-2020). The frequency of wind direction is expressed as a percentage based on n = 69666 half-hourly data points.

[Figure]

**Figure 5 (revised manuscript).** The daily mean of short-wave radiation at the top of the atmosphere ($S_{TOA}$), short-wave incoming ($S_{in}$) and outgoing ($S_{out}$), cloud factor ($CF$), long-wave incoming ($L_{in}$) and outgoing ($L_{out}$), net radiation ($R_{net}$), turbulent sensible ($H$) and latent ($LE$) heat fluxes at the AWS-M for DJFMA, 2009-2020. $L_{in}$ and $R_{net}$ start from 1 December 2010. The black line highlights the mean of 2009-2020.

[Figure]

**Figure 6 (revised manuscript).** Mean monthly energy flux density of $R_{net}$, $H$, $LE$ and $F_{surface}$ for DJFMA, 2009-2020.

4. Line 23-25: The author does not discuss the influence of mid-latitude western disturbances on sublimation in the main text. The author can use the Reanalysis data (such as geopotential height and wind fields at 500 hPa or other heights from ERA5 or JRA55) to obtain the direct knowledge of circulation which can impact the sublimation and energy balance.

We thank the reviewer for the suggestion. We acknowledge that we briefly discussed the influence of westerlies on sublimation. However, in this work we intended to keep our analysis as observation data-based as possible, so we did not use any reanalysis dataset to conduct the spatial-scale wind circulation analysis. Therefore, a detailed large-scale analysis of the wind systems is beyond the scope of this study.

However, to incorporate the suggestion we have done a simple atmospheric circulation analysis using horizontal wind ($u$ and $v$) and vertically integrated moisture divergence (VIMD) from monthly ERA5 (0.25° grid) data at 500 hPa (Fig. S6 in the revised supplementary, a copy shown below). The figure depicts that, at 500 hPa, horizontal wind and moisture moves from the west and interacts with the western Himalayan relief/region during the DJFMA (2009-2020). We also noted that during the DJFMA months, there is a substantial amount of moisture divergence in the western Himalayan region, which corresponds to increased precipitation. This corroborates our idea that WDs events bring higher moisture and low temperatures into the region, which impede sublimation (discussed in Sect. 4.5 and Sect. 4.6 in revised manuscript). Since the manuscript is already long and our main focus is sublimation using observation (AWS-M) datasets, we would keep this Fig. S6 in the supplementary material.

**ERA5 mean wind and VIMD for DJMFA (2009-2020)**

[Figure]

**a** Dec  **b** Dec

**c** Jan  **d** Jan

**e** Feb  **f** Feb

**g** Mar  **h** Mar

**i** Apr  **j** Apr

**Fig. S6 (in supplementary file).** Mean horizontal wind (from *u* and *v* components) and vertically integrated moisture divergence (VIMD) at 500 hPa for DJFMA during 2009-2020 based on ERA5 data. ERA5 data was downloaded from the Climate Data Store, ECMWF (https://cds.climate.copernicus.eu). AWS-M location is shown with black square, with a label. Arrows in the wind plots refer to the direction of winds. Plots are generated in Python using several packages, mainly xarray, proplot, matplotlib. Plot template was taken from Lalande et al (2021). Note: the higher negative values (dark blue areas) of VIMD (i.e., large moisture convergence) refers to precipitation intensification in a particular region (https://apps.ecmwf.int/codes/grib/param-db/?id=213).

To further confirm the influence of WDs in sublimation, we have discussed the relationship based on observed datasets in Sect. 4.5 (revised manuscript) and also in Sect 4.6 (revised manuscript). Therein we used AWS-measured *u, RH, CF* and Geonor precipitation during the possible WDs events. We kept the analysis figure in supplementary material (Fig. S5 in the revised supplementary file; a copy shown below) because this is a short discussion supported by minimal analysis. From Fig. S5A we discern that strong winds (more than 10 m s$^{-1}$) often bring higher moisture (greater than 60-70% *RH*) during DJFMA and subsequent precipitation. We also note that higher precipitation events were associated with strong *u* (Fig. S5B) implying that those events were likely driven by WDs at the study site.

[Figure]

**Fig. S5 (in supplementary file).** (A) Relationship between relative humidity, wind speed and cloud factor, and (B) relative humidity, cloud factor and precipitation. The number of data points is mentioned on the respective panel. Precipitation was recorded at the glacier base camp at 3850 m a.s.l.

In addition to the large-scale wind/moisture circulation analysis (Fig. S6), we incorporated a literature review of large-scale circulation analyses on the influence of WDs over the western Himalayan region during the winter months. Newly added texts read as:

*Line No. 498-505 (Sect. 4.5):*
*'WDs events are most dominant during winter months around the Chhota Shigri region as observed based on the ERA5's horizontal wind fields and vertically integrated moisture divergence datasets at 500 hPa from 2009 to 2020 (Fig. S6). Zhu et al. (2021) and Liu et al. (2020) investigated the impact of WDs in the western Himalayan region using a large-scale circulation analysis based on ERA5's geopotential height and wind fields at 500 hPa and ERA Interim's atmospheric datasets (precipitation, vertically integrated water vapour transport and specific humidity), respectively. Both studies indicated that during the winter months in the western Himalaya and western Tibetan regions, WDs storm activities transport a significant amount of moisture and influence the precipitation.'*

*Line No. 555-559 (Sect. 4.6):*
*'Large-scale circulation studies based on moisture/source tracking approach confirms that the synoptic activity of WDs in the western Himalayan region during winter months intensifies not only the upper-troposphere disturbances (higher precipitation) but also their thermal structure through baroclinic processes (Baudouin et al., 2021; Canon et al., 2015). Thus, very strong and cold winds, with higher moisture through WDs impedes sublimation in the region.'*

5. The author examines the role of cloud cover on SEB and turbulent heat fluxes based on clear-sky conditions and overcast conditions. However, this can be finished by using just two years of data. The relationship between CF and sublimation is small (Table 4). Thus, CF (or Sin) is not the main factor causing the interannual changes in sublimation in winter during 2009-2020. I strongly recommend that authors analyze the factors which control the interannual changes in sublimation in winter during 2009-2020 through correlation analysis. The author can explain interannual changes in sublimation from the view of energy balance. And the author should analyze the relationships between RH and sublimation, between albedo and sublimation, between Sin and sublimation, between Sout and sublimation, between Lin and sublimation, Lout and sublimation between D and Tair, between D and Ts, and between D and RH. I guess that albedo is an important factor that contributes to the interannual changes in sublimation by changing Ts. The concrete results are depending on your further analysis.

We agree with Reviewer 1 that $CF/S_{in}$ is not the main factor for sublimation. Therefore, we did not use such statements anywhere in the manuscript. However, we did write as: *'Cloud cover, on the other hand, has a significant impact on the primary meteorological variables, particularly $S_{in}, T_s$ and $q_s$.'* in Line No. 536-537. The observation was based on (i) the correlation coefficient ($r$) analysis (Fig. 10; Sect. 4.5 in revised manuscript), (ii) difference in $LE$ magnitude in clear-sky

and overcast conditions (Sect. 4.4 in revised manuscript) and interannual correlation of sublimation and meteorological variations (Sect. 4.6 in revised manuscript; Table S4).

Concerning the main factors of sublimation, we note that sublimation is governed by a combined effect of different meteorological variables, primarily the vertical moisture ($q - q_s$) and temperature ($T_{air} - T_s$) difference/gradients, wind speed and the state of the surface boundary layer (stability). This is supported by multiple regression and variance analysis presented in Table 3 (revised manuscript; shown below). The multiple linear regressions analysis showed $q - q_s$, $T_{air} - T_s$, $u$ and $T_s$ together are the best sublimation predictors in clear-sky conditions (95%), overcast conditions (89%) and for all-data (without $CF$ filter; 92%). Considering two combined predictors, $q - q_s$ and $u$ explains the highest variance (> 80%) in sublimation for clear-sky, overcast and all-data conditions. However, individually, sublimation did not show strong correlation with any meteorological variables (Fig. 10 in revised manuscript, a copy below) except $q - q_s$, $T_{air} - T_s$, $T_s$ and $q_s$ which are the direct variables. All these correlation coefficients were based on half-hourly datasets for the daytime (between 09:00 and 16:00 IST).

Indeed, albedo is an important variable in sublimation, with a stronger correlation in clear-sky conditions ($r = -0.29$; Fig. 10 below). In overcast conditions, however, albedo has little impact on sublimation ($r = -0.02$).

Considering your suggestion, we developed an interannual correlation analysis based on cumulative sublimation and meteorological variables (n = 11 years; Table S4; a copy below). Inter-annual correlation analysis showed $T_s$ ($r = 0.85$; $p < 0.01$) correlates the highest with cumulative sublimation, followed by $S_{in}$ ($r = 0.79$; $p < 0.05$) and $RH > 80\%$ ($r = -0.76$; $p < 0.01$). This suggests that on an interannual scale, high $T_s$ (through higher $S_{in}$) and low near-surface moisture conditions supports sublimation.

Overall, we find that near-surface temperature ($T_{air} - T_s$) and moisture gradient ($q - q_s$), along with wind speed, were important factors in sublimation, while cloud cover shapes the meteorological variables. We have revised our discussion following the arguments presented above. We would like to invite the reviewer to go through over the revised manuscript sections (particularly Sect. 4.5, 4.6 and 5.1) for the meteorological factors of sublimation.

Below we highlighted the concluding sentences in different sections regarding the main sublimation factors:

*Line No. 507-509 (Sect. 4.5):*
*'Overall, we noted that at sub-hourly scale near-surface moisture availability (through $q - q_s$) plays a bigger role in determining LE magnitude, with the combined effects from several meteorological variables, particularly $q_s$, $T_s$ and $u$.'*

*Line No. 535-537 (Sect. 4.6):*
*'This suggests that on an interannual scale, high $T_s$ (through higher $S_{in}$) and low near-surface moisture conditions supports sublimation. Cloud cover, on the other hand, has a significant impact on the primary meteorological variables, particularly $S_{in}$, $T_s$ and $q_s$.'*

*Line No. 663-665 (Sect. 5.1):*
*'Overall, we conclude that near-surface moisture availability (through $q$ - $q_s$) plays a major role in governing LE magnitude at the AWS-M at different temporal scales, while moisture availability was influenced and conditioned by a number of meteorological variables, notably $S_{in}$, $u$, $q_s$, and $T_s$.'*

**Table 3 (in revised manuscript).** Summary of the multiple linear regression analysis (k-fold (k = 10) cross-validation) of sublimation rate and combined meteorological variables. Total n = 13217, 2708 and 2063 half-hourly data points for all-data, clear-sky and overcast conditions, respectively, between 09:00 and 16:00 IST for DJFMA (2009-2020). The *p*-value of $r^2$ was always < 0.001.

| Variable | $r^2$ cross-validation | | |
|---|---|---|---|
| | **All-data** | **Clear-sky** | **Overcast** |
| $T_s$, $u$ | 0.53 | 0.69 | 0.44 |
| $T_{air}$, $u$ | 0.10 | 0.17 | 0.30 |
| $q$, $u$ | 0.03 | 0.15 | 0.15 |
| $q_s$, $u$ | 0.58 | 0.71 | 0.47 |
| $u$, $T_{air}$-$T_s$ | 0.58 | 0.75 | 0.29 |
| $u$, $q$-$q_s$ | 0.86 | 0.85 | 0.84 |
| $q$, $u$, $T_{air}$ | 0.26 | 0.21 | 0.34 |
| $q$, $u$, $T_s$ | 0.79 | 0.82 | 0.71 |
| $q_s$, $u$, $T_{air}$ | 0.77 | 0.90 | 0.51 |
| $q_s$, $u$, $T_s$ | 0.59 | 0.71 | 0.48 |
| $T_{air}$-$T_s$, $q$-$q_s$, $u$ | 0.92 | 0.95 | 0.89 |
| $T_{air}$-$T_s$, $q$-$q_s$, $S_{in}$ | 0.85 | 0.85 | 0.67 |
| $T_{air}$-$T_s$, $q$-$q_s$, $L_{in}$ | 0.84 | 0.85 | 0.67 |
| $T_{air}$-$T_s$, $q$-$q_s$, $R_{net}$ | 0.85 | 0.86 | 0.70 |

[Figure]

**Figure 11 (in revised manuscript).** Pearson's correlation coefficient (*r*) matrix of various meteorological and SEB components at the AWS-M in clear-sky and overcast conditions between 09:00 and 16:00 IST, 2009-2020. Number (n) of half-hourly data points are shown on top of the panels.

**Table S4 (in supplementary file).** Interannual correlation coefficient (*r;* n = 11) between cumulative sublimation ($S_c$) and primary meteorological variables for 2009-2020. '*' refers to p < 0.05.

|  | *RH > 80%* | $T_s$ | $T_{air}$ | *u* | *RH* | $S_{in}$ | *CF* |
|---|---|---|---|---|---|---|---|
| $S_c$ | -0.76* | 0.85* | -0.15 | -0.10 | -0.50 | 0.79* | 0.56 |

6. Discussion: I sometimes feel confused about the sentences in the discussion. Take section 5.3 for example. The author said that sublimation during the summer- monsoon season was lower, which could be due to the ISM-driven warm and moist atmosphere in the southern slope of the HK region. However, sublimation is higher at very high altitudes despite high summer-monsoon humidity, e.g., East Rongbuk Glacier site (6523 m a.sl.). What is the main point of the author? When author compared their study with other studies, the author should note the spatial and temporal scales. Some studies used the glacier-wide values, while others used point values. Some studies used the low-altitude values, while others used the high- altitude values. Some studies used the annual values, while others used winter values. These data with different scales are incomparable. The author should select these data carefully.

We thank you for the comment. We rephrased the respective sentence in the revised manuscript. Now it reads as:

*Line No. 729-733:*
*'Sublimation rate during the summer-monsoon season, in general, was lower than that during winter (Table 4), which could be due to the warm and moist atmospheric conditions driven by the ISM. Despite high summer-monsoon humidity, sublimation is higher at very high altitude sites, such as the East Rongbuk Glacier site (6523 m a.sl.). At very high altitudes, this is most likely due to strong winds and low air vapour pressure.'*

Regarding the heterogeneous spatial and temporal scale of the comparison, we would like to highlight that sublimation is poorly investigated and understood across the HK region as compared to general glacier SEB studies. Therefore, available datasets are heterogeneous from the spatial and temporal scale point of view. For example, only a single study in the Himalaya (Stigter et al., 2018) and a few in Tibet (Guo et al., 2021; Zhu et al., 2020) have discussed sublimation in detail. Also, in some of the studies meteorological values are not clearly defined or shown (e.g., in Dongkemadi Glacier in central Tibet; Liang et al., 2018). Furthermore, most studies in the HK and Tibet regions have focused exclusively on the summer season, considering

the importance of summer SEB in melt modeling. Therefore, it is extremely hard to make an exhaustive comparison with consistent spatial or temporal scale based on limited available studies. This is the main reason for selecting all available studies and compare their values to draw a general overview of sublimation rates across HK and High Mountain Asia (HMA).

To clearly highlight the differences in spatial and temporal scales of existing sublimation studies, we added one sentence in the revised manuscript in the respective section (Sect. 5.3). It reads as:

*Line No. 715-718:*
*'The existing sublimation studies in the HK and HMA are not uniform in terms of spatial and temporal scales, which makes it difficult to compare sublimation and associated processes consistently. However, it is worthwhile to recall these existing sublimation datasets for comparison, not to conduct a thorough and rigorous comparison, but to qualitatively address the sublimation process in the region.'*

Considering your suggestion, we have revised the respective Table (Table 3 in revised manuscript) and texts slightly for a consistent/similar comparison of the sublimation rates based on available studies. We would like to invite the reviewer to go through the revised comparison section.

**Minor comments:**

Line 32: wind-driven transport can cause accumulation in some sites.

The sentence was framed from the ablation point of view. However, to give this sentence a bit more ablation perspective, we have revised it and now reads as:

*Line No. 32:*
*'..wind-driven transport/erosion—lead to the loss of snow and ice mass..'*

Line 121-123: How do you get albedo in the night? Thus, what is your surface albedo threshold value in the night which is used to discern snow or bare-ground?

We filtered the snow-covered period based on the daytime surface albedo ($\alpha_{acc}$) $\geq 0.4$. We revised the sentence for clarity and now it reads as:

*Line No. 32:*
*'We filtered the snow-covered period for SEB based on the daytime surface albedo threshold value above 0.4 at the AWS-M (the mean bare-ground/snow-free surface albedo was < 0.25 for July-August; 2009-2020).'*

Line 153: Please explain the physical significance of Fsurface. If Fsurface is larger than 0, does melt occur at that time?

We revised the respective section and included a dedicated sentence mentioning the physical significance of residual energy ($F_{surface}$). The sentence reads as:

*Line No. 172-174:*
*'When $F_{surface}$ is larger than 0 W m$^{-2}$ (towards positive), it will direct towards the surface/snowpack and warm it up until it reaches at melting point ($T_s = 0°C$), and then surplus $F_{surface}$ will cause melting (Hock, 2005).'*

Line 209-210: Can you analyze the difference between infrared measured Ts and Ts derived from Lout? Please list the figure. Is the emissivity of bared-ground similar to that of snow cover? This is important for the author to calculate Ts from Lout.

We have calculated the difference between infrared measured $T_s$ and $T_s$ derived from $L_{out}$ (shown in Table R1, below). The requested comparison figure is presented below (Fig. R3). $L_{out}$-based $T_s$ was derived using the Stefan-Boltzmann equation for the snow surface, with emissivity of 1 (following Hock and Holmgren, 2005; Wagnon et al., 2003). We observed the least root mean square error (RMSE = 0.23°C) and mean absolute error (MAE = 0.06°C) for emissivity = 1. We also observed that as emissivity decreases, RMSE and MAE increase considerably (Table R1).

We would also like to point out that we derived $T_s$ from $L_{out}$ only to compare it to the measured $T_s$. We did not use this $L_{out}$-based $T_s$ in any of our SEB/sublimation calculations, therefore it has no impact on our results.

**Table R1.** Comparison of RMSE and MAE for different snow emissivity.

| Emissivity | $r^2$ | RMSE [°C] | MAE [°C] |
|---|---|---|---|
| 1 | 0.99 | 0.23 | 0.06 |
| 0.99 | 0.99 | 14.09 | 14.08 |
| 0.98 | 0.99 | 27.39 | 27.37 |
| 0.97 | 0.99 | 39.98 | 39.95 |

Considering your above comment, we have revised the respective sentence in the revised manuscript and now it reads as:

*Line No. 222-224:*
*'$T_s$ was directly used from the measurement by an infrared radiometer (Table 1). The correlation between infrared measured $T_s$ and $T_s$ derived from $L_{out}$ (using Stefan-Boltzmann equation for the*

*snow surface with emissivity of 1 following Hock and Holmgren, 2005) was $r^2 = 0.99$ (p < 0.001) with RMSE = 0.23°C.'*

[Figure]

**Figure R3.** Comparison of half-hourly values of the infrared measured $T_s$ vs $L_{out}$-based $T_s$ at the AWS-M site.

Line 312-313: Which components in Rnet are more important in playing an essential role in governing the turbulent fluxes? And the author should indicate the timescale.

On an interannual scale, $S_{in}$ showed stronger indirect relationship with *LE* and *H* ($r$ = -0.80 and -0.61, respectively; $p$ < 0.05) than $L_{in}$ ($r$ = -0.36 and -0.39, respectively; not significant). Whereas, in half-hourly scale, in clear-sky conditions, $S_{in}$ and $L_{in}$ both have shown a nearly similar impact on *LE* ($r$ = -0.25, -0.26 and 0.29, respectively). In overcast conditions, impact of $S_{in}$ and $L_{in}$ equally rises ($r$ = -0.42 and -0.41). These analyses are further discussed in Sect 4.3 and Sect. 4.5 in the revised manuscript.

We have highlighted the timescales of our analysis in the respective sentence and sections. The revised texts read as:

*Line No. 420-421 (**Sect. 4.3. Seasonal and interannual variation of SEB components**):*
*'$S_{in}$ showed a stronger indirect relationship with LE and H (r = -0.80 and -0.61, respectively; p < 0.05) than $L_{in}$ (r = -0.36 and -0.39, respectively; not significant).'*

*Line No. 506-507 (**Sect. 4.5. Turbulent heat fluxes under different cloud conditions**):*
*'At sub-hourly scale, neither $R_{net}$ nor $S_{in}$ and $L_{in}$ can adequately explain turbulent fluxes in both overcast and clear-sky conditions (r = < 0.50; Fig. 11).'*

Line 313-314: I can not understand this sentence.

We have removed this sentence in the revised manuscript.

Line 325: What do you mean about the different colors of lines in Figure 6?

The different colour lines in Figure 6 (original manuscript) define different years from 2009/10 to 2019/20. In the revised manuscript, we have combined both Fig. 5 and 6 (original manuscript) into a single figure (Figure 5 in revised manuscript) and it contains a legend for all coloured lines. A copy of the revised figure (Figure 5) is shown under your main comments no. 3 (above).

Line 359. Please add the "in the daytime" in the title of section 4.5.

Revised it as suggested.

Line 363-364 Why precipitation is higher in February and March than in January and April? High precipitation always means high cloud cover. This is different from your results of CF.

We thank the reviewer for the question and concern. In Fig. 8 (in revised manuscript, a copy shown below), we showed that February was the second cloudy (overcast) month, which is consistent with February having the second highest precipitation amount (24% of winter; Table S3 in revised manuscript). In January, more hours were cloudy, but only accounted for 19% of the total winter precipitation. This could be partly explained by the average moisture content in January (0.8 g kg$^{-1}$) which was ~30% lower than in February (1.1 g kg$^{-1}$) (see Table S3 in revised manuscript). In addition, it should be worth mentioning here that Fig. 8 is based on n = 8191 half-hourly data points, which was extracted from n= 23903 half-hourly data points following clear-sky ($CF < 0.2$) and overcast ($CF > 0.8$) filters. Night values were neglected in Fig. 8 because our $CF$ calculation is based on $S_{in}$ data which was unavailable during night. Since, we do not have the cloud information from night and transition hours (for 16 hours), it is difficult to understand and correlate the precipitation value with cloud cover.

In addition, in-situ precipitation data from the Geonor station was available only for five discontinuous hydrological years, therefore only five years of precipitation data is not sufficient to discern the relationship between cloud cover/fraction and precipitation intensity in the region.

[Figure]

**Figure 8 (in revised manuscript).** Monthly fraction of clear-sky (CF ≤ 0.2) and overcast (CF ≥ 0.8) conditions at the AWS-M. Fraction percentage is calculated from n = 5810 clear-sky and n = 2381 overcast observations from total n = 23903 half-hourly values between 09:00 and 16:00 IST (DJFMA, 2009-2020).

Line 376: 3 times lower?

We modified the sentence and now reads as:

*Line No. 465-466:*
*'In clear-sky, the mean daytime H was -66 W $m^{-2}$ which is three times more negative than that in overcast conditions (-21 W $m^{-2}$).'*

Line 423: 145 ± 25 mm w.e. a-1?

Revised it as suggested.

Section 4.7: There is no section 4.7.2 in this part. The author can merge section 4.7 and section 4.7.1 as one part.

Thanks for the suggestion. We have merge it and renamed the section as:

*'4.6 Sublimation and its relationship with meteorological variables'*

Line 451-452: I can not agree with the author, because we can not find that low Tair (-5°C and -10°C) corresponds to high Ts (0°C and -10°C) for the same time. From figure14b, we can only find that sublimation was the larger when Tair ranged between -5°C and -10°C (compared to Tair in other values). This is similar to the Ts. Thus, the content in Line 451-452 is not correct.

Thanks for catching this issue. This sentence has been corrected and now reads as:

Line No. 551-553:
'Sublimation was the largest when $T_{air}$ ranged between -5°C and -10°C and also when $T_s$ ranged between 0°C and -10°C (Fig. 12; Fig. 13B and C). Whereas, sublimation was considerably lower when moisture availability was higher, $T_s$ was significantly lower, with very strong u (Fig. 12; Fig. 13).'

To show this observation clearly, we made two more meteorological clusters (i.e., $T_s$ > -10°C and $T_s$ < -10°C) in the existing Fig. 12 (in revised manuscript, a copy shown below). From Fig. 12 (bottom panels) it is clear that sublimation was almost half when $T_s$ < -10°C compared to $T_s$ > -10°C. This is also evident in Fig. 13 (in revised manuscript, a copy shown below).

[Figure]

**Figure 12 (in revised manuscript).** Half-hourly daytime (09:00-16:00) records of sublimation (red), wind speed (blue) and specific humidity (green) at the AWS-M for different clusters: no filter, $u > 10$ m sec$^{-1}$, $q > 2$ g kg$^{-1}$, $< 1$ g kg$^{-1}$, $T_s > -10°C$ and $T_s < -10°C$. Data period: DJFMA, 2009-2020. Number of data-points n=30257, 2347, 12295, 9762, 10552 and 12734 for no filter, $u > 10$ m sec$^{-1}$, $q > 2$ g kg$^{-1}$, $< 1$ g kg$^{-1}$, $T_s > -10°C$ and $T_s < -10°C$, respectively.

[Figure]

**Figure 13 (in revised manuscript).** Scatter plot of $u$, $q$, $T_{air}$, $T_s$, $CF$, $S_{in}$ and $L_{in}$ against sublimation rate at the AWS-M. The colour of the data points refers to the measured wind speed ($u$). Total n = 14088 half-hourly data points between 09:00 and 16:00 IST for DJFMA (2009-2020).

Line 480-481: What is your timescale?

Our analysis is based on half-hourly $LE$ datasets, however for a longer/seasonal perspective, we averaged it for daily, monthly, and seasonal (DJFMA) timescale as well. In the current section (Sect. 5.1 in revised manuscript), we have discussed $LE$ from an overall/holistic perspective to summarise the factor controlling $LE$ at the AWS-M site.

Line 544: Do you want to say that sublimation during the summer-monsoon season was lower than that during winter?

Yes. We revised it for a better read.

*Line No. 729-731:*
*'Sublimation rate during the summer-monsoon season, in general, was lower than that during winter (Table 4), which could be due to the warm and moist atmospheric conditions driven by the ISM'*

Line 545-547: The studies of Mölg et al. (2012) and Li et al. (2018) are in the south and central Tibet, respectively. They do not study the glaciers in the northern slope of the HK region.

Thanks for pointing this out. Previously we missed to cite the SEB studies which are from the northern slope of the Himalaya, for example, on Naimona'nyi and East Rongbuk glaciers (e.g., Zhu et al., 2021; Liu et al. 2021). We revised the sentence and now reads as:

*Line No. 733-735:*
*'The high moisture from ISM also impacts Tibetan glaciers, particularly those located in the northern slope of the Himalaya (Zhu et al., 2021; Liu et al., 2021) and central Tibet (Mölg et al., 2012; Li et al., 2018).'*

Line 547-548: Can you explain the phenomenon that you found in these sentences?

In this sentence, we did not intend to discuss any phenomenon, but to point out (from the existing studies) that sublimation rates in the central Himalaya are relatively higher during post-monsoon and pre-monsoon (for example in Yala and Mera glaciers; Table 4 in revised manuscript).

Although near-surface moisture (*RH*) is relatively higher in the central Himalaya during post-monsoon and pre-monsoon (because it is close to the Bay of Bengal) than in winter season. The sublimation rates are comparatively higher (Table 4; Yala and Mera glaciers). We assume that this is because of the high altitude location (for Yala it was > 5300 m a.s.l. and for Mera it was >5300 m a.s.l. and > 6500 m a.s.l.), where strong wind or snow blowing could have increased sublimation considerably.

We have revised the respective sentences and now reads as:

*Line No. 735-741:*
*'In the Nepalese central Himalaya, we note a higher sublimation value of 2.4 and 1.8 mm d$^{-1}$ on the Yala Glacier during the post- and pre-monsoon seasons (Table 4). Litt et al. (2019) also reported a significantly higher sublimation rate of 7.1 and 1.9 mm d$^{-1}$ during the post- and pre-monsoon on the Mera Glacier. Such higher sublimation rates on the Yala and Mera glaciers are unique, particularly during post- and pre-monsoon seasons when air vapour pressure/specific humidity is higher than that of winter season (Shea et al., 2015; Perry et al., 2020). Nevertheless, such higher sublimation can also be partially attributed to snow*

*blowing/redistribution at such high-altitude sites (Barral et al., 2014; Wagnon et al., 2013; Huintjes et al., 2015b).'*

Line 548-549: I do not find that the moisture content is relatively higher during post- and pre-monsoon on the Mera Glacier than that in winter in Table 5. And the altitudes are significantly different between the post- and pre-monsoon periods.

We have now updated Table 4 (in revised manuscript, a copy shown below) with *RH* and wind speed values for the Yala and Mera glaciers for the pre- and post-monsoon seasons. Table 4 shows *RH* for Yala and Mera glaciers are considerably higher (~70%) in pre-monsoon and close to 50% in post-monsoon (Litt et al., 2019).

Post-monsoon sublimation rate was not available for 6543 m a.s.l. AWS site from the Mera (Litt et al., 2019). So, to keep it consistent, now we have used the Mera Glacier sublimation rates from a single site: 5360 m a.s.l. where both pre- and post-monsoon seasons' sublimation rates are available. Updated Table 4 shown below:

**Table 4 (in revised manuscript).** Compilation of sublimation rate across the HMA region. '*' refers to the evaporation values. Do' refers to the same method as in the row immediately above.

| Site | Altitude (m a.s.l.) | Region | Period of observation | Season approx. to Chhota Shigri | Surface | Method | $S$ (mm d$^{-1}$) | RH (%) | u (m) | Reference |
|---|---|---|---|---|---|---|---|---|---|---|
| **Tibetan Plateau** | | | | | | | | | | |
| Zhadang | 5665 | Nyainqen tanglha Shan | 1 October to 31 May, 2008-2013 | Winter | Glacier-wide | Bulk-aerodynamic | 0.5 | 44 | 3.6 | Zhu et al. (2018) |
| Muztag Ata No. 15 | 4400 | Eastern Pamir | 1 October to 31 May, 2008-2013 | Winter | Glacier-wide | Do | 0.7 | 42 | 6.4 | Zhu et al. (2018) |
| Parlung | 4800 | Southeast TP | 1 October to 31 May, 2008-2013 | Winter | Glacier-wide | Do | 0.4 | 64 | 3.4 | Zhu et al. (2018) |
| Muji | 4685 | Northeast Pamir | 1 October to 31 May, 2011- 2017 | Winter | Glacier-wide | Do | 0.5 | 50 | 4 | Zhu et al. (2020) |
| Qiangtang No. 1 | 5882 | Inland TP | 1 October to 31 May, 2012-2016 | Winter | Glacier-wide | Do | 0.4 | 46 | 6.8 | Li et al. (2018) |
| Guliya Ice Cap | 6000 | Kunlun Shan | 1 October to 31 May, 2015-2016 | Winter | Glacier-wide | Do | 0.3 | 67 | 7.9 | Li et al. (2019) |
| Dongkem adi | 5600 | Central TP | 7 October 1992 to 4 May 1993 | Winter | Glacier ELA | Do | 0.2 | - | 4.3 | Liang et al. (2018) |
| August-one | 4817 | Qilian Mountains | Jan-May, Oct-Sept, 2016-2020 | Winter | Glacier | Do | 0.4 | 68 | 6.9 | Guo et al. (2021) |
| **Himalaya** | | | | | | | | | | |
| Pindari | 3750 | Central Himalaya | December 2016 to February 2017 | Winter | Medial moraine | Monin-Obukhov theory | ~0.3 | 55 | 1.2 | Singh et al. (2020) |
| Yala | 5350 | Central Himalaya | 15 October 2015 to 20 April 2017 | Winter | Glacier/ ablation zone | Eddy-covariance | 1 | ~40 | ~2.5 | Stigter et al. (2018) |
| Yala | 5330 | Do | 1 October to 15 November, 2012-2017 | Post-monsoon | Glacier/ ablation zone | Bulk-aerodynamic | 2.4 | ~49 | ~1.8 | Litt et al. (2019) |
| Yala | 5330 | Do | 10 May to 5 June, 2012-2017 | Pre-monsoo | Glacier/ ablation | Do | 1.8 | ~77 | ~1.9 | Do |

| Site | Elevation | Region | Date | Season | Surface | Method | | | | Reference |
|---|---|---|---|---|---|---|---|---|---|---|
| | | | | n | zone | | | | | |
| Mera | 5360 | Do | 1 October to 15 November, 2013-2016 | Post-monsoon | Glacier/ablation zone | Do | 1.9 | ~46 | ~2.8 | Do |
| Mera | 5360 | Do | 10 May to 5 June, 2013-2016 | Pre-monsoon | Glacier/ablation zone | Do | 3.3 | ~72 | ~2.3 | Do |
| Lirung | 4250 | Do | 26 September to 12 October 2016 | Post-monsoon | Glacier debris | Eddy-covariance | 1.8-2.8* | ~60 | ~3 | Steiner et al. (2018) |
| South Col, Everest | 7945 | Do | 22 May to 31 October 2019 | Summer-monsoon | Ice-rock surface | Bulk-aerodynamic | ~0.8 | ~60 | 6.3 | Matthews et al.(2020) |
| East Rongbuk | ~6500 | Do | 28 April to 2 May 2008 | Pre-monsoon | Glacier | Lysimeter | 1.9 | - | - | Yang (2010) |
| East Rongbuk | 6523 | Do | 1 May to 22 July 2005 | Summer-monsoon | Glacier | Bulk-aerodynamic | 0.05-1.2 | 60 | 4.2 | Liu et al. (2021) |
| Xixibangma | 5900 | Do | 23 August to 29 September 1991 | Summer-monsoon | Glacier | Calculated | 0.02 | 36 | 5.9 | Aizen et al. (2002) |
| Naimona'nyi | 5543 | Do | 1 October 2010 to 31 May 2018 | Winter | Glacier-wide | Bulk-aerodynamic | 0.6 | 34 | 5.5 | Zhu et al. (2021) |
| Chhota Shigri | 4670 | Western Himalaya | 1 Dec 2012 to 29 Jan 2013 | Winter | Glacier/ablation zone | Do | 0.8 | 44 | 4.9 | Azam et al. (2014a) |
| Chhota Shigri | 4863 | Do | 1 December to 30 April, 2009-2020 | Winter | Seasonal snow on moraine | Do | 1.1 | 43 | 5 | This study |

Line 550-551: What is the cause for the differences that the authors found in this sentence?

Thanks for pointing this out. Here we compared our study (for DJFMA) with wet/moist season's sublimation rates without any data/analysis from this study site. Therefore, we have removed the sentence from the revised manuscript.

Line 553-555: I cannot understand what you want to say.

We revise it, as:

*Line No. 741-744:*
*'Overall, dry air, low atmospheric pressure and high wind speeds are suitable conditions for sublimation, as reported from various high-altitude sites in the HMA (Matthews et al., 2020; Litt et al., 2019; Stigter et al., 2018; Zhu et al., 2018) and everywhere in the world (Wagnon et al., 1999; Cullen et al., 2007; Fyffe et al., 2021)'*

Line 560-564: These sentences have no relationship with the title 'Sublimation fraction to winter snowfall and its importance'.

Thanks for this suggestion. We have removed the sentences from the respective paragraph and revised it.

Line 569: Why sublimate is higher in the northwestern part of the HK than that in the other parts of the HK region?

It is because the atmospheric conditions of the northwestern part of the HK, as well as the west Tibet, are very dry and arid compared to other parts of the HK, such as the Eastern or Central Himalaya, where climate is more humid and monsoon precipitation is higher. Dry air and low atmospheric pressure create a steep near-surface moisture gradient, which fosters strong sublimation. To clarify this in the revised manuscript, we incorporated a dedicated sentence for this, and it reads as:

*Line No. 771-776:*
*'Although there are limited observations available from various parts of the Himalaya or HMA, the available findings show sublimation fraction to winter/annual snowfall/precipitation is higher in the northwestern part of the HK and western Tibet (e.g., Zhu et al., 2020; Gascoin, 2021). This is likely due to the atmospheric condition of the northwestern part of the HK and western Tibet which is drier than eastern and central Himalaya. Dry atmospheric conditions favor higher sublimation than the wet due to high near-surface humidity gradients.'*

Line 579-580: Such a higher sublimation fraction? You mean that the sublimation fraction is higher on Qiangtang No 1 Glacier than other glaciers on the TP. Have you compared the meteorological data at the Qiangtang No. 1 Glacier to that on other glaciers?

In this section we did not compare meteorological conditions of different glaciers. We intended to limit our discussion on the sublimation fractions and its variation across the region. In other glacier/areas, sublimation fraction was comparable (between ~16% and ~60%) and not very much contrasting, except at the Qiangtang No 1 Glacier which where sublimation fraction is 65-169%. Therefore, to briefly discuss the contrasting conditions at Qiangtang No 1, we presented the meteorological conditions (wind speed, *RH* and snowfall values from Table 3 in Li et al., 2018) in the discussion. This comparison briefly points out the contrast, which we thought to be interesting for the readers.

Line 610-611: This result disagreed with your description in section 5.4. Sublimation fraction to winter snowfall is higher on Qiangtang No 1 Glacier than that on Chhota Shigri Glacier.

Thanks for pointing this out. We revised the sentence and now reads as:

*Line No. 807-809:*
*'The cumulative DJFMA sublimation was 145 ± 25 mm w.e. a⁻¹, corresponding to 16-42% of the fraction of winter snowfall at the AWS-M site, which is relatively higher than that observed in*

*other studies across the HK region, with considerable interannual variations and lower than a few of the Tibetan sites.'*

Line 620: There are more than 10 published works about Chhota Shigri Glacier. However, the meteorological data for that glacier is still not open to scientists in the world.

We have uploaded AWS-M data used in this study in Zenodo along with the codes used in SEB calculation and generating the figures. The citable open-access link (https://doi.org/10.5281/zenodo.6609605; Mandal et al., 2022) is now provided in the revised manuscript.

---

## Author Comment (AC3)

**Response to Reviewer 2**

Below we provide our responses **(in red text)** point-by-point to each comment from the reviewer **(in black text)**. *Italic texts* are used to highlight specific changes in the updated manuscript.

The authors present a very clear study on multiple winters worth of energy balance data from a site in the Western Himalaya. They show the consistent importance of sublimation during snow cover times and find results that generally match well with previous studies in the field. The work is very timely and the numbers found here will guide research conducted on the larger scale that is not able to include the process on a distributed scale. The paper is very clearly written, well supported with data and clear Figures that leave only very few general comments from my side which I detail below and which I hope you can address. I have a number of minor comments at the end. I applaud the authors for the field work that this work is based on as well as the clear way of presenting the results. It is important work and I think this should be an important paper in the TC library in future.

We sincerely thank Reviewer 2 for evaluating our manuscript, suggestions, and the positive feedback on our study. We have responded to your specific comments and outlined the changes that we have made in the revised manuscript. If the reviewers and the editor are satisfied with our responses, we will submit our revised manuscript. Below, we have highlighted (point-wise) the major revisions that we have made in the revised manuscript in response to your main comments:

- Included a detailed discussion on the influence of cloud cover on sublimation, based on correlation analysis and comparison of existing cloud and sublimation studies from the Himalaya/Tibet region,

- Included a future perspective on sublimation sensitivities to meteorological variables, which may be useful to readers in getting an idea of future sublimation and subsequent changes in terms of SEB of snow/glacier surfaces,

- In addition, the result and discussion sections have been significantly revised, with new text and restructuring in response to Reviewer 1's suggestions.

**General:**

In the Discussion I would expect more discussion of the role of cloud cover, which as you note is important but to me has a surprisingly low correlation and obviously wind plays a very different role in these regimes (your Figure 12). Could you compare the relative cloud cover to the other sites, or at least the ones from (Guo et al., 2021; Stigter et al., 2018). Not to cite here as still in review but (Conway et al., 2022) also provides some new great insights in this direction. I would

hope to learn here how different I can expect my sublimation rates to be when I work in a different regime of overcast conditions.

We thank Reviewer 2 for the suggestions. We have considered the suggested studies to compare with our results and expanded the discussion section (Sect. 5.1). Some of the newly incorporated texts discussing the respective aspect are mentioned below.

Conway et al. (2022) focused on the melt seasons SEB and associated meteorological characteristics, but they also discussed the complex interaction between cloud cover and overall SEB of the glaciers across various sites, including four glacier sites in the Himalaya. The findings of Conway et al. (2022) are consistent with our findings in general. They also point out that at most of their study sites, increased cloud cover decreases the magnitude of *LE* and *$F_{surface}$*. At very high altitude sites (e.g., Mera, Zongo) they found that *LE* is still negative (that means sublimation) in overcast conditions (at *CF* > 0.7 mean melt-season's *LE* was ~-60 W m$^{-2}$; Fig. A5 in Conway et al., 2022). Overall, their findings show that in overcast conditions, near-surface meteorology (particularly near-surface vapour pressure and relative humidity) is significantly altered which limits higher magnitude of radiation and turbulent heat fluxes. Cloud cover, on the other hand, has little impact on wind speed and *$T_{air}$* at most sites, including the Chhota Shigri Glacier (Fig. 8 in Conway et al. 2022). Although the climatic setting varies greatly across the Himalayan region, and cloud cover's influence is complex, we highlight that overcast conditions lower the magnitude of sublimation, as shown by our study and Conway et al. (2022). Following your suggestion as Conway et al. (2022) is still in review we did not cite it in this study.

*Line No. 653-663:*
*'We note the importance of cloud cover in modulating the surface atmosphere at the AWS-M site which favours sublimation, however, the correlation coefficient between CF and LE was poor (r = -0.09 and -0.16 in clear-sky and overcast conditions, respectively; Fig. 10). This is most likely due to the complex influence of cloud cover on meteorological variables, particularly $S_{in}$ and $L_{in}$. Cloud cover reduces $S_{in}$, which impedes sublimation, but at the same time it also increases $L_{in}$, which promotes sublimation partly by raising $T_s$. This is well-supported by the higher correlations between sublimation and $S_{in}$ and $L_{in}$, particularly in overcast conditions (Fig. 10). Although Stigter et al. (2018) did not discuss the correlation between sublimation and cloud cover/factor at the Yala Glacier, they did indicate that sublimation was negligible or about zero on overcast days when humidity was higher. This is supported by the poor correlation of determination ($r^2 = 0.08$) between sublimation and RH at the Yala Glacier. Guo et al. (2021) also did not obtain a statistical relationship between sublimation and cloud cover, but they also noted a weak sublimation rate during cloudy months due to high moisture and warm conditions.'*

Table 2: In text you say max T_a is 0.1, in Table 0.0

Thanks for pointing this out. We have revised Table 2 (original manuscript) with maximum $T_{air}$ as 0.1°C. Following Reviewer 1's suggestion, we have shifted Table 2 (original manuscript) to supplementary material (as Table S3).

Table 4: R2 for u is 0? I am also surprised that CF seems to be more correlated to sublimation in the transition phase than in overcast or clear sky condition. Can that be explained? I would have expected a higher correlation under overcast condition.

In the previous version of the manuscript, we showed Pearson's correlation coefficient ($r$) as well as the coefficient of determination ($r^2$) through linear regression analysis. Since we already have a dedicated analysis and figure showing $r$ (Figure 10; revised manuscript; Sect. 4.5), we planned to remove $r^2$/linear regression between sublimation and meteorological variables from Table 3 (revised manuscript; note: Table 3 was Table 4 in original manuscript). We only kept the multiple linear regression analysis in Table 3 (revised manuscript) to show the readers how a combined effect of meteorological variables influences sublimation. This way we still have the correlation analysis between sublimation and meteorological variation and discussion (Fig. 10; Sect. 4.5 dedicatedly) while skipping the discussion of $r^2$ for the same relationships. Using $r$ and $r^2$ for the same relationship is a little confusing and difficult to follow for the readers.

Indeed, the relationship between $u$ and sublimation is weak in both clear-sky and cloudy conditions ($r$ = 0.37 and 0.33 in clear-sky and overcast, respectively; Fig. 10 in revised manuscript). The absence of strong correlation between sublimation and $u$ is expected because a supportable condition for an enhanced sublimation was created by a combination of meteorological variables, primarily the vertical moisture and temperature gradient, wind speed and the state of the surface boundary layer (stability) (please refer to Sect. 5.1 in revised manuscript). The weak correlation between $u$ and sublimation can be partly explained by the very heterogeneous wind speed at the AWS-M. For example, available observation from various studies showed that $u$ generally decreased in overcast conditions (e.g., Stigter et al., 2018; also in Conway et al., 2022 in several glacier sites). However, in overcast conditions we often had higher $u$ (Fig. 9 and Fig. 11; revised manuscript) due to westerly activities (discussed in Sect. 4.5 and 4.6; revised manuscript). This heterogeneity was the cause of weak correlation between $u$ and sublimation in part. In this regards, new study by Fugger et al. (2022) also reported that the relationship between $LE$ and meteorological variables was highly unpredictable, and $u$ failed to explain the variability of $LE$/sublimation at five on-site glacier studies in the central and eastern Himalaya (see their Fig. 9A).

Correlation between sublimation and $CF$ was also weak ($r$ = -0.09 and -0.16 in clear-sky and overcast conditions, respectively; Fig. 10 in revised manuscript). This is likely due to the complex influence of cloud cover on meteorological variables, particularly $S_{in}$ and $L_{in}$. For

instance, cloud cover reduces $S_{in}$, which impedes sublimation, but at the same time it also increases $L_{in}$, which promotes sublimation partly by raising $T_s$.

We do observe a slightly higher correlation between sublimation and $CF$ in overcast conditions ($r$ = -0.16) than clear-sky ($r$ = -0.09), but not that significant.

To give a thought to your concern (based on our observation in the original manuscript) that $CF$ was more correlated in the transition phase, we analysed this relationship a bit further. We analysed the sublimation correlations for three more cloud conditions by binning $CF$ for three more categories within the transition phase (i.e., $CF > 0.2 <= 0.4$; $CF > 0.4 <= 0.6$; $CF > 0.6 <= 0.8$). In those categories, we also did not find any strong correlation between sublimation rates and CF. The $r$ values were similar as in clear-sky and overcast conditions (not shown here). This is partially reflected in Fig. 13 (revised manuscript; a copy shown below).

[Figure]

**Figure 13 (in revised manuscript).** Scatter plot of $u$, $q$, $T_{air}$, $T_s$, $CF$, $S_{in}$ and $L_{in}$ against sublimation rate at the AWS-M. The colour of the data points refers to the measured wind speed ($u$). Total n = 14088 half-hourly data points between 09:00 and 16:00 IST for DJFMA (2009-2020).

Based on the above argument on the weak relationship of $u$ and $CF$ with sublimation, we revised our discussion. We would like to invite the reviewer to go through the revised manuscript (Sect. 5.1; revised manuscript). The newly incorporated texts are highlighted below:

*Line No. 644-653 (Sect. 5.1):*
*'Stigter et al. (2018) and Guo et al. (2021) noted a stronger direct relationship between LE and u, which does not agree with the present study. This could be partly explained by the very heterogeneous wind speed at the AWS-M (Fig. 13). For example, the available observations from different sites showed that u generally decreases in overcast conditions (e.g., Stigter et al., 2018; Guo et al., 2021). However, at the AWS-M, u was often higher in overcast conditions (Fig. 9; Fig. S5) due to westerly activities (discussed in Sect. 4.5 and 4.6). Very high u maintains a neutral stratification of the boundary layer resulting in a lower LE magnitude. This heterogeneity is likely the cause of weak correlation between u and sublimation in part. However, the highest multiple regression variance in combination with u (~90%; Table 3) in clear-sky and overcast conditions emphasise the importance of u in driving LE/sublimation. Fugger et al. (2022) also noted that the relationship between LE and meteorological variables is highly unpredictable, and u fails to explain the variability of LE at five on-glacier sites in the central and eastern Himalaya (see their Fig. 9A).'*

*Line No. 653-659 (Sect. 5.1):*
*'We note the importance of cloud cover in modulating the surface atmosphere at the AWS-M site which favours sublimation, however, the correlation between CF and LE was poor (r = -0.09 and -0.16 in clear-sky and overcast conditions, respectively; Fig. 10). This is most likely due to the complex influence of cloud cover on meteorological variables, particularly $S_{in}$ and $L_{in}$. Cloud cover reduces $S_{in}$, which impedes sublimation, but at the same time it also increases $L_{in}$, which promotes sublimation partly by raising $T_s$. This is well-supported by the higher correlations between sublimation and $S_{in}$ and $L_{in}$, particularly in overcast conditions (Fig. 10).'*

L505ff/Figure 15: This is interesting – could you expand here what that means for a potential future change especially of T_air? Also in the text you mention the big sensitivity to T_s, but that under melting condition won't change much. It seems to be equally (or just slightly less) sensitive to T_air though, which likely will change. That seems important to me for future consideration.

Thank you for the suggestion. The future perspective of the sensitivities is interesting and worth expanding. Following your suggestion, we have expanded the discussion. The newly incorporated texts are presented below. We invite you to go through the revised respective section (Sect. 5.2; revised manuscript).

*Line No. 686-696 (Sect. 5.2):*
*'Another important aspect of sensitivity to meteorological variables is related to the future atmospheric warming and its consequences to sublimation. $T_s$ exhibited a higher sublimation sensitivity than $T_{air}$ (Fig. 14), but under melting conditions it will not change much because the temperature of snow/ice surface cannot rise above the melting point ($T_s$ = 0°C). However, relative potential changes in $T_{air}$ are likely to be higher across the globe including in the*

*Himalayan region (Hock et al., 2019; Krishnan et al., 2019). Therefore, sublimation sensitivity with respect to $T_{air}$ could be a major concern in the future, due to the expected warming. Considering a future $T_{air}$ increase of ~0.3 ± 0.2°C decade$^{-1}$ for the Himalayan region (Ren et al., 2017; Krishnan et al., 2019), a crude estimate suggests a ~5% decrease in sublimation per decade from snow/glacier surfaces. This could probably be attributed to a lower energy sink through LE, which will boost the efficiency of $S_{in}/R_{net}$ resulting in a more surface melt. However, since sublimation is a process driven by the combined effect of multiple meteorological variables, it remains to be seen how the sensitivity of a single variable influences the overall sublimation and associated processes.'*

**Minor comments:**

L20: replace 'consequently' with 'resulting in'

Done.

L21: 'largest fraction' or 'proportion'

We think 'proportion' would be a better choice. Thanks for the suggestion. Done.

L24: 'to the region'

Done.

L26: sublimation is a variable, not a parameter; remove the two 'the' articles

Done, thanks.

L40: 'more abundant'

Done.

L53: 'The contribution …is …'

Done.

L:57: 'poorly understood'

Done.

L71: Technically it has been applied (Sakai et al., 2004) although they did not term it sublimation and on this debris cover (as in (Steiner et al., 2018)) it is more an evaporative process. But this is a grey area, and at least our attempt to measure sublimation over snow with a pan lysimeter have simply been unsuccessful because they freeze and can't measure properly. You also later mention the PhD thesis by Yang (2010).

Thanks for the information.

L101: 'radiation', no need for a plural here

Done.

Table1: The superscript a at the bottom is missing. Also again I would use 'radiation' in singular

We will make sure the superscript is there, thanks. Changed it to 'radiation'.

L134: 'single-Alter-shielded'

Done.

L164: you use 'net radiation' here but earlier used net all-wave radiation'. I would go throughout for the shorter version.

We choose net radiation across the manuscript following your suggestion.

L166: The two sentences should be conjoined with comma or you need to restructure syntax

We have revised it following your suggestion.

L189f and in general: no need to include [in …] with the units

Done, we remove [in ...] here and elsewhere.

L229: remove 'equation by' or 'the equation by'

Done.

L292: I would leave 'snow cover' in singular

Done.

L299: does not

We removed this sentence from the revised manuscript considering Reviewer 1's suggestion to shorten the respective section.

L310: maybe rather 'down to'

Done, thanks.

L322: 'such a high contribution'

Done.

L336: remove 'thin'

Done.

Figure 11: Nice figure and just a pedantic comment – can you make Tair-Ts instead of Tair_Ts in the axis label? Also you introduce D here but only introduce it much later in the text (L447). Make sure to somehow introduce it earlier, otherwise as a reader I need to go looking forward in the text, which is awkward. The question is though why you show it at all here as it is just the reverse from q-qs – you could consider to just remove the column/row in both subfigures.

Thanks for pointing this out and the suggestion. We have revised the figure (Figure 10; revised manuscript; a copy shown below) as suggested and removed $D$ from the figure (Figure 10). Also, considering $D$ is already included in $LE$ equation, we have decided not to use $D$ at all, across the manuscript and therefore, revised the respective sections accordingly.

[Figure]

**Figure 10 (in revised manuscript).** Pearson's correlation coefficient (*r*) matrix of various meteorological and SEB components at the AWS-M in clear-sky and overcast conditions between 09:00 and 16:00 IST, 2009-2020. Number (n) of half-hourly data points are shown on top of the panels.

L433: 'restrict'

We removed this sentence from the revised manuscript.

L453: It is quite clear that D is directly positively related to LE as it is the main part of the equation/definition, so it can't really be any other way. I would remove this sentence.

We have removed this as suggested.

L523: remove one 'in this study'

Done.

L525/L540: maybe 'similar' or 'comparable' instead of 'identical'

Done. We choose 'similar'.

L577: 'with the major part'

Done.

L581: 'This supports …'

Done.

L600: 'impediment'

Done, thanks.

L603: maybe 'reducing by 70%' and 'raising by 25%'

Done, thanks.

L604: Bit confusing – restraining to what? Also '50% cloud fraction' to be clear.

We revised the respective sentences for clarity. Now the sentence reads as:

*Line No.: 798-799*
*'The cloud cover also restrains the meteorological condition favourable for turbulent heat fluxes and reduces their magnitude by more than 50%.'*

L607: remove 'were'

Done.

L608: 'suggesting it is crucial for …'

Done.

L612f: remove 'significantly' – that is a hard term and you don't really show that here. I would also remove the part behind the semi-colon. That is always a given and a bit redundant. And you say the same in the following sentences already.

Done, we have revised it as suggested.

L620: Please provide this for the final version. It is a pity if such a statement remains without a link in a final publication.

We have uploaded AWS-M data used in this study in Zenodo along with the codes used in SEB calculation and generating the figures. The citable open-access link (https://doi.org/10.5281/zenodo.6609605; Mandal et al., 2022) is now provided in the revised manuscript.

**References**

Conway, J. P., Abermann, J., Andreassen, L. M., Azam, M. F., Cullen, N. J., Fitzpatrick, N., Giesen, R., Langley, K., MacDonell, S., Mölg, T., Radic, V., Reijmer, C. H., and Sicart, J.-E.: Cloud forcing of surface energy balance from *in-situ* measurements in diverse mountain glacier environments, The Cryosphere Discuss. [preprint], https://doi.org/10.5194/tc-2022-24, in review, 2022.

Dimri, A. P., Niyogi, D., Barros, A. P., Ridley, J., Mohanty, U. C., Yasunari, T., and Sikka, D. R.: Western Disturbances: A review, Reviews of Geophysics, 53, 225–246, https://doi.org/10.1002/2014RG000460, 2015.

Fugger, S., Fyffe, C. L., Fatichi, S., Miles, E., McCarthy, M., Shaw, T. E., Ding, B., Yang, W., Wagnon, P., Immerzeel, W., Liu, Q., and Pellicciotti, F.: Understanding monsoon controls on the energy and mass balance of glaciers in the Central and Eastern Himalaya, The Cryosphere, 16, 1631–1652, https://doi.org/10.5194/tc-16-1631-2022, 2022.

Guo, S., Chen, R., Han, C., Liu, J., Wang, X., and Liu, G.: Five-year analysis of evaposublimation characteristics and its role on surface energy balance SEB on a midlatitude continental glacier, Earth and Space Science, e2021EA001901, https://doi.org/10.1029/2021EA001901, 2021.

Hock, R., Rasul, G., Adler, C., Cáceres, B., Gruber, S., Hirabayashi, Y., Jackson, M., Kääb, A., Kang, S., Kutuzov, S., Milner, A., Molau, U., Morin, S., Orlove, B. ,and Steltzer, H. I.: Chapter 2: High

Mountain Areas, IPCC Special Report on the Ocean and Cryosphere in a Changing Climate (SROCC), 131–202, 2019.

Krishnan, R., Shrestha, A. B., Ren, G., Rajbhandari, R., Saeed, S., Sanjay, J., Syed, Md. A., Vellore, R., Xu, Y., You, Q., and Ren, Y.: Unravelling Climate Change in the Hindu Kush Himalaya: Rapid Warming in the Mountains and Increasing Extremes, in: The Hindu Kush Himalaya Assessment, edited by: Wester, P., Mishra, A., Mukherji, A., and Shrestha, A. B., Springer International Publishing, Cham, 57–97, https://doi.org/10.1007/978-3-319-92288-1_3, 2019.

Pratap, B., Sharma, P., Patel, L., Singh, A. T., Gaddam, V. K., Oulkar, S., and Thamban, M.: Reconciling High Glacier Surface Melting in Summer with Air Temperature in the Semi-Arid Zone of Western Himalaya, Water, 11, 1561, https://doi.org/10.3390/w11081561, 2019.

Ren, Y. Y., Ren, G. Y., Sun, X. B., Shrestha, A. B., You, Q. L., Zhan, Y. J., ... & Wen, K. M.: Observed changes in surface air temperature and precipitation in the Hindu Kush Himalayan region over the last 100-plus years. Advances in Climate Change Research, *8*(3), 148-156, 2017.

Stigter, E. E., Litt, M., Steiner, J. F., Bonekamp, P. N. J., Shea, J. M., Bierkens, M. F. P., and Immerzeel, W. W.: The Importance of Snow Sublimation on a Himalayan Glacier, Front. Earth Sci., 6, 108, https://doi.org/10.3389/feart.2018.00108, 2018.

---

## Author Response (AR1)

**Response to Reviewer 1**

Below we provide our responses **(in red text)** point-by-point to each comment from the reviewer **(in black text)**. *Italic texts* are used to highlight specific changes in the updated manuscript.

**General comments:**

This paper analyzes the winter meteorological data and surface energy balance (SEB) on a lateral moraine of the Chhota Shigri Glacier in the western Himalaya during 2009-2019, and then explored the effects of cloud cover on winter energy balance and sublimation on that site. In addition, this paper presents long-term glacio- meteorological data in the western Himalayas, which is very important for studying the glacier mass balance changes on the Tibetan Plateau and the surrounding areas. This is an interesting paper, but needs major revisions before can be accepted for publications in TC. I have several comments that the authors should address.

We sincerely thank the Reviewer 1 for evaluating our manuscript and giving suggestions to improve the quality of the manuscript. All specific comments have been addressed in detail below, and we have also highlighted (point-wise) the major revisions that we have carried out in the revised manuscript in response to major comments:

- We have shortened the meteorological condition section by reducing the texts, figures (part of the figures have been shifted to the supplementary material) and tables in respective sections.

- Reorganized the presentation of meteorological and surface energy balance (SEB) analysis to account for different temporal scales, such as daily, hourly, seasonal, and inter-annual.

- Incorporated a large-scale wind/moisture circulation analysis using ERA5 500 hPa datasets to understand the influence of western disturbances (WDs) on sublimation.

- Sub-hourly and inter-annual correlation analysis, as well as multiple regression analysis of sublimation and meteorological variables, were included. Further, the discussion and interpretation were significantly revised, with a focus on sublimation factors.

**Main comments:**

1. Introduction: There are some studies discussing the energy balance and mass balance around HK regions and other regions on the Tibetan Plateau in recent years, such as Pamir and Tibet. Although the authors reviewed some studies, it is relatively simple. The authors should review more recent studies about energy balance and mass balance around the Tibetan Plateau, and pointed out the limitations of these studies.

We acknowledge that we reviewed only a few studies related to glacier surface energy balance (SEB) from the Pamir and Tibet regions in the Introduction section. It is because our main focus is to address the research gaps related to sublimation estimation and its role in glacier mass balance in the Himalaya-Karakoram (HK) region. Therefore, we mainly highlighted the importance of turbulent heat fluxes, understanding gaps for sublimation and its estimation methods used in the HK region and skipped/avoided to discuss findings/gaps related to glacier mass and energy balance.

To make the Introduction section more inclusive, we have included some of the recent glacier SEB (e.g., Li et al., 2019; Zhu et al., 2020) and sublimation (e.g., Guo et al., 2021) related studies from the Pamir and Tibet regions. Newly incorporated texts in the Introduction read as:

*Line No. 41-45:*
*'However, SEB studies on Tibetan glaciers are relatively more abundant (~17 investigated glaciers/ice-covered sites; Table S1), including direct turbulent heat flux measurements (Yang et al., 2011; Zhu et al., 2018) except in Pamir and Kunlun Mountains (Zhu et al., 2020). Glaciers in the Pamir Range are extreme continental type, with cold temperature and low annual precipitation (Li et al., 2019), thus their SEB characteristics is expected to behave differently than majority of HK glaciers which are alpine type, with relatively higher precipitation and temperature.'*

*Line No. 68-71:*
*'In the Muji Glacier in northeast Pamir, cold season's evaposublimation loss is > 70% of the corresponding snowfall (Zhu et al., 2020). In the Qilian Mountains at the August-One Glacier in north-east Tibetan Plateau, evaposublimation loss is lower but accounts for about 15% of annual precipitation (Guo et al., 2021).'*

In addition, in Table 4 (revised manuscript) and the sublimation section (Sect. 5.3 and 5.4 in the revised manuscript), we specifically reviewed several studies from the Pamir and Tibet regions, comparing sublimation rates across the HK and Tibetan glacierised regions. Some of the texts are as follows:

*Line No. 585-588 (Sect. 5.3):*
*'Sublimation rates during winter were slightly higher in the Pamir Range, e.g., Muztag Ata No. 1 (Zhu et al., 2018) and the Muji site (Zhu et al., 2020) compared to the inland/central Tibet region, e.g., Qiangtang No. 1 (Li et al., 2018) and the Dongkemadi site (Liang et al., 2018). This is likely due to the relatively drier atmospheric conditions in the Pamir Range than the central or eastern parts of Tibet (Table 4; also Liu et al., 2020).'*

*Line No. 630-644 (Sect. 5.4):*

*'In the Tibetan Plateau, at the Zhadang Glacier, sublimation loss was 26% of the total mass loss annually (Huintjes et al., 2015a). At the August-One Glacier in the Qilian Mountains, evaposublimation accounts for 15% of the annual precipitation, with the major part during winter periods (Guo et al., 2021). In some sites of the Tibetan Plateau, sublimation fraction is considerably higher. For example, in the Muji Glacier in Pamir, the cold season's evaposublimation loss is > 70% of the corresponding snowfall (Zhu et al., 2020). In the Kunlun Mountains on the Guliya Ice Cap, glacier-wide sublimation loss was ~120% of the winter snowfall and ~50% of the annual snowfall (Zhu et al., 2022). On the Qiangtang No. 1 Glacier in inland Tibet, the sublimation and evaporation loss fraction were about 65-169% of the snowfall during 2012-2016, which is a significantly higher mass loss than gain (Li et al., 2018). Such a higher sublimation fraction at the Qiangtang No. 1 Glacier during non-melt seasons was associated with high wind speed (~7 m s$^{-1}$), lower RH (~46%) and low annual precipitation (362-614 mm). This supports that the dry and windy environment fosters sublimation. Although there are limited observations available from various parts of the Himalaya and HMA, these observations show that the sublimation fraction to winter/annual snowfall/precipitation is higher in the north-western part of the HK and western Tibet (e.g., Zhu et al., 2020; Gascoin, 2021). This is likely due to the atmospheric condition of the north-western part of the HK and western Tibet which is drier than the eastern and central Himalaya. Dry atmospheric conditions favour higher sublimation than the wet due to high near-surface humidity gradients.'*

2. I do not find how authors calibrate or evaluate their modeled results in this work. The parameters in the energy balance always need to be calibrated by some measured values. For example, the selected surface roughness lengths for momentum, temperature, and humidity, and the different formula of turbulent heat will impact the modeled H and LE. If the modeled results are calibrated, the data will be more credible. Why the author does not select surface temperature to calibrate their model using the iteration method. In this work, the author can deduce that there is no snow cover at the AWS-M site when albedo is smaller than 0.4. This data can also be used to calibrate their modelled values. In addition, it seems that there are few studies about the glacier energy balance which delete G in their model. The AWS-M site remains snow-covered during winter and bare sand/sediment exposed during summer. Whether bare sand below the snow can provide more energy to heat the snow when compared to glacier ice below the snow? Or the author just focuses on the energy feature.

We sincerely acknowledge the concern of Reviewer 1 for the calibration/validation of our SEB calculation, especially the bulk method for turbulent heat flux calculation.

We would like to mention that, in this work, we used the measured surface temperature ($T_s$; through an infrared radiometer) as an input to calculate the turbulent heat fluxes ($H$ and $LE$) following the bulk method. Therefore, we cannot compare the simulated $T_s$ and measured $T_s$ to calibrate/validate

the bulk methods' performance, which is usually used for evaluating glacier SEB/bulk models when direct $T_s$ measurements are not available and $T_s$ is modelled.

Concerning the credibility of our SEB and bulk modelling approach, this method has been successfully applied on this glacier in an on-glacier site SEB experiment (Azam et al., 2014a) at 4670 m a.s.l. (~1 km away from the AWS-M; our study site). The SEB model result was validated using the observed surface/stake melt ($r^2 = 0.98$), and also a significant correlation ($r^2 = 0.96$) was observed between the simulated $T_s$ and observed $T_s$ (Azam et al., 2014a).

To incorporate your concern, we attempted to validate the sublimation rates using SR-50A data (snow height), which was available for a shorter period (Dec 2009 to Apr 2015). Our plan was to filter snow depth/height (SR-50A) data for periods with no snowfall longer than minimum one week and compare calculated sublimation rates and observed snowpack thickness change over such periods. However, due to inconsistency in the measured precipitation records from the Geonor gauge, the available data for this analysis was only for two years (DJFMA of 2012/13 and 2014/15) (Sect. 3.1). We examined SR-50A data for those two years, but we couldn't find any long enough periods without snowfalls. Figure R1 (below) shows Geonor precipitation data from DJFMA of 2012/13 and 2014/15. Therefore, comparing the SR-50A snow thickness change data to sublimation rates was not appropriate in this case. For the years when Geonor data was not available, we used daily precipitation from the neighbouring Indian Meteorological Department (IMD) Stations: Bhuntar (~50 km) and Manali (~40 km). In those station data also we do not find any longer period without snowfall (figure not shown here). The daily precipitation data was not available from Keylong, the closest IMD station (~60 km) from our study site. There was a possibility to apply the measured/assumed snow density to convert height change into mass loss and compare it with corresponding sublimation loss. However, considering the limitation in data availability, we could not conduct a direct validation of our bulk model.

[Figure]

**Figure R1.** Daily Geonor precipitation (measured at Chhota Shigri glacier base camp at 3850 m a.s.l.) and SR-50A (AWS-M; 4863 m a.s.l.) height change during DJFMA 2012/13 and 2014/15. Data shown only for DJFMA period.

The primary focus of this work is to estimate sublimation directly derived from *LE*. To analyse and quantify the meteorological variable's sensitivity to sublimation and a possible uncertainty in our bulk model, we conducted a sensitivity analysis of the calculated sublimation (Sect. 5.2), where we perturbed the calculated sublimation by changing meteorological variables (e.g., $T_{air}$ by $\pm 1°C$, $T_s$ by $\pm 1°C$, wind speed (*u*) by $\pm 10\%$ and *RH* by $\pm 10\%$) and surface roughness lengths (0.0005 m, 0.002 m, 0.003 m and 0.004 m) to evaluate the range of sublimation.

In addition, the bulk method was compared with the direct eddy-covariance method over the snow surface at the Yala Glacier (Central Himalaya, Nepal) to evaluate the performance of the bulk method (Stigter et al., 2018). They found a good agreement between eddy-covariance and bulk method ($r^2 = 0.88$) in estimating sublimation rates, which shows the reliability of the bulk method over snow surface in the Himalayan site. However, the reviewer also suggested that roughness

lengths in the bulk model is very crucial to get an accurate result. Considering this and for better accuracy, we have used the previously calculated snow surface roughness lengths already obtained from this glacier (0.001 m; Azam et al., 2014a), which was calculated using wind measurements at two different levels following a conventional logarithmic profile (e.g., Moore, 1983). These aspects are discussed in Sect. 3.2.2.

Based on the aforementioned discussion, we have modified statements in the method section (Sect. 3.2.2. revised manuscript) highlighting the limitations in our bulk model validation. Now the sentence reads as:

*Line No. 235-238:*
*'Due to the limitations in the data availability, direct validation of the bulk model used in this study was not possible, therefore, our results are based on Azam et al (2014a)'s bulk model validation done on this glacier in 2012/13 and it proved to deliver robust results compared to observations. We also conducted a sensitivity analysis of our bulk model including surface roughness lengths (Sect. 5.2).'*

Regarding the ground/conductive heat flux ($G$), we calculated $G$ at the AWS-M site following the method proposed by You et al. (2014) and Luce and Tarboton (2010). The diurnal variation of $G$ is shown below through a figure (Fig. R2) along with other major energy fluxes. The results show that $G$ is negligible and is about -0.4 ± 4.4 W m$^{-2}$ for DJFMA (2009-2020; n = 73624 half-hourly data points) compared to other energy fluxes. That shows that there is no significant energy coming from the ground/bare surface. In addition, considering the inadequate measurement of the subsurface heat flux and relative information of $G$ in the HK region (Stigter et al., 2021), we did not consider it in our SEB calculation. Further, our focus in this study is sublimation and its drivers/importance, hence the modelling $G$ is beyond the scope of this study.

[Figure]

**Figure R2.** Mean diurnal cycle of $G$ (following You et al., 2014 and Luce and Tarboton, 2010) at the AWS-M for DJFMA (2009-2020) along with $R_{net}$, $H + LE$, and $F_{surface}$.

3. There are so many results in section 4. The author could shorten this section, because some studies have introduced the meteorological data at AWS-M in that glacier (Azam et al., 2016). Are there any special features from your data? Those special features are important. In addition, I hope that the author can discern the timescales for their results, such as diurnal cycle, seasonal cycle, and interannual timescales.

We thank the reviewer for the concern. To incorporate the reviewer's suggestion and shorten the meteorological condition section, we have revised most of the figures in this section. For instance, a part of the figures (e.g., Figure 2, 3, 5C and G, and 6D, 7B in original manuscript) have been shifted to the supplementary material. Table 2 (original manuscript) has also been moved to the supplementary material (Table S3). We have modified/revised the figures (e.g., Figure 2, 3, 5 and 6 in revised manuscript) for your reference. We merged Sect. 4.2 and 4.3 (original manuscript) into a single Sect. 4.1 (revised manuscript). We also reorganized the presentation of meteorological and surface energy balance (SEB) analysis that account for different temporal scales. Such as in Sect. 4.2 (revised manuscript) we presented the diurnal cycle of all meteorological and SEB variables, and seasonal and interannual variation of SEB components in Sect. 4.3 (revised manuscript).

We would also like to mention that Azam et al. (2016) did not focus on the SEB related details, for example, they discussed fewer variables (e.g., $T_{air}$, $RH$, $u$, $S_{in}$ and $S_{out}$). $T_s$, *albedo*, $q$ and *CF* were missing in their study, which are important variables to understand SEB characteristics. Furthermore, the meteorological conditions presented by Azam et al. (2016) were based on only four years of datasets (2009-2013), but we have used 11-years long datasets in this study (2009-2020).

To discern the timescale of meteorological/SEB characteristics, we have performed the analysis at various temporal scales as suggested by the reviewer. For instance, first, in Sect 4.1 we presented the daily variations and ranges of all meteorological/radiation components. Second, in Sect. 4.2 we presented the diurnal cycle of all meteorological and SEB variables. Third, in Sect. 4.3 we presented the seasonal and interannual variation of SEB components with their statistical correlations. Further, in Sect. 4.6 we analysed sublimation considering various temporal scales, e.g., daily, sub-hourly, seasonal and interannual. Since sublimation and turbulent fluxes are our main interests, we have investigated them in detail in Sect. 4.4 and 4.5, and focused more on the impact of cloud cover in sublimation. We believe the manuscript is now in much clearer and logical shape due to these analysis results. We thank the reviewer for the same.

We invite the reviewer to go through the reorganised sections (Sect. 4.1, 4.2 and 4.3) in the revised manuscript.

[Figure]

**Figure 2 (revised manuscript).** Monthly climatology of air ($T_{air}$) and surface temperature ($T_s$), relative humidity ($RH$), wind speed ($u$) and surface albedo ($\alpha_{acc}$) at the AWS-M for 2009-2020. DJFMA (1 December to 30 April) period is highlighted with a light blue rectangle in each panel. The shades around the line and scatter points correspond to one standard deviation (SD). Dashed lines in panel E refer to snow-surface albedo ($\alpha_{acc} = 0.4$; red line) for SEB analysis and bare-surface albedo ($\alpha_{acc} = 0.2$; black line). Daily values of $T_{air}$, $T_s$, $RH$, $u$ and albedo for the study period are shown in Fig. S2. Mean yearly values of different variables are provided in Table S4.

[Figure]

**Figure 3 (revised manuscript).** Windrose of the AWS-M for DJFMA (2009-2020). The frequency of wind direction is expressed as a percentage based on n = 69666 half-hourly data points.

[Figure]

**Figure 5 (revised manuscript).** The daily mean of short-wave radiation at the top of the atmosphere ($S_{TOA}$), short-wave incoming ($S_{in}$) and outgoing ($S_{out}$), cloud factor ($CF$), long-wave incoming ($L_{in}$) and outgoing ($L_{out}$), net radiation ($R_{net}$), turbulent sensible ($H$) and latent ($LE$) heat fluxes at the AWS-M for DJFMA, 2009-2020. $L_{in}$ and $R_{net}$ start from 1 December 2010. The black line highlights the mean of 2009-2020.

[Figure]

**Figure 6 (revised manuscript).** Mean monthly energy flux density of $R_{net}$, $H$, $LE$ and $F_{surface}$ for DJFMA, 2009-2020.

4. Line 23-25: The author does not discuss the influence of mid-latitude western disturbances on sublimation in the main text. The author can use the Reanalysis data (such as geopotential height and wind fields at 500 hPa or other heights from ERA5 or JRA55) to obtain the direct knowledge of circulation which can impact the sublimation and energy balance.

We thank the reviewer for the suggestion. We acknowledge that we briefly discussed the influence of westerlies on sublimation. However, in this work we intended to keep our analysis as observation data-based as possible, so we did not use any reanalysis dataset to conduct the spatial-scale wind circulation analysis. Therefore, a detailed large-scale analysis of the wind systems is beyond the scope of this study.

However, to incorporate the suggestion we have done a simple atmospheric circulation analysis using horizontal wind ($u$ and $v$) and vertically integrated moisture divergence (VIMD) from monthly ERA5 (0.25° grid) data at 500 hPa (Fig. S6 in the revised supplementary, a copy shown below). The figure depicts that, at 500 hPa, horizontal wind and moisture moves from the west and interacts with the western Himalayan relief/region during the DJFMA (2009-2020). We also noted that during the DJFMA months, there is a substantial amount of moisture divergence in the western Himalayan region, which corresponds to increased precipitation. This corroborates our idea that WDs events bring higher moisture and low temperatures into the region, which impede sublimation (discussed in Sect. 4.5 and Sect. 4.6 in revised manuscript). Since the manuscript is already lengthy and our main focus is sublimation using observation (AWS-M) datasets, we would keep this Fig. S6 in the supplementary material.

**ERA5 mean wind and VIMD for DJMFA (2009-2020)**

[Figure]

**Fig. S6 (in supplementary file).** Mean horizontal wind (from *u* and *v* components) and vertically integrated moisture divergence (VIMD) at 500 hPa for DJFMA during 2009-2020 based on ERA5 data. ERA5 data was downloaded from the Climate Data Store, ECMWF (https://cds.climate.copernicus.eu). AWS-M location is shown with black square, with a label. Arrows in the wind plots refer to the direction of winds. Plots are generated in Python using several packages, mainly xarray, proplot, matplotlib. Plot template was taken from Lalande et al (2021). Note: the higher negative values (dark blue areas) of VIMD (i.e., large moisture convergence) refers to precipitation intensification in a particular region (https://apps.ecmwf.int/codes/grib/param-db/?id=213).

To further confirm the influence of WDs in sublimation, we have further discussed the relationships based on observed dataset in Sect. 4.5 (revised manuscript) and also in Sect 4.6 (revised manuscript). Therein we used AWS-measured *u*, *RH*, *CF* and Geonor precipitation during the possible WDs events. We kept the analysis figure in supplementary material (Fig. S5 in the revised supplementary file; a copy shown below) since it is a short discussion supported by minimal analysis. From Fig. S5A we discern that strong winds (more than 10 m s$^{-1}$) often bring higher moisture (greater than 60-70% *RH*) during DJFMA and subsequent precipitation. We also noted that higher precipitation events were associated with strong *u* (Fig. S5B) implying that those events were likely to be driven by WDs at the study site.

[Figure]

**Fig. S5 (in supplementary file).** (A) Relationship between relative humidity, wind speed and cloud factor, and (B) relative humidity, cloud factor and precipitation. The number of data points is mentioned on the respective panel. Precipitation was recorded at the glacier base camp at 3850 m a.s.l.

In addition to the large-scale wind/moisture circulation analysis (Fig. S6), we incorporated a literature review of large-scale circulation analyses on the influence of WDs over the western Himalayan region during the winter months. Newly added texts read as:

*Line No. 410-414 (Sect. 4.5):*
*'WDs events are most dominant during winter months around the Chhota Shigri region. This was observed from the ERA5's horizontal wind fields and vertically integrated moisture divergence datasets at 500 hPa from 2009 to 2020 (Fig. S6). Zhu et al. (2021) and Liu et al. (2020) also indicated that during the winter months in the western Himalaya and western Tibetan regions, WDs storm activities transport a significant amount of moisture and influence the precipitation.'*

*Line No. 463-466 (Sect. 4.6):*
*'Large-scale circulation studies based on the moisture/source tracking approach confirms that the synoptic activity of WDs in the western Himalayan region during winter months intensifies not only the upper-troposphere disturbances (higher precipitation) but also their thermal structure through baroclinic processes (Baudouin et al., 2021; Cannon et al., 2015). Thus, very strong and cold winds with higher moisture from WDs impedes sublimation in the region.'*

5. The author examines the role of cloud cover on SEB and turbulent heat fluxes based on clear-sky conditions and overcast conditions. However, this can be finished by using just two years of data. The relationship between CF and sublimation is small (Table 4). Thus, CF (or Sin) is not the main factor causing the interannual changes in sublimation in winter during 2009-2020. I strongly recommend that authors analyze the factors which control the interannual changes in sublimation in winter during 2009-2020 through correlation analysis. The author can explain interannual changes in sublimation from the view of energy balance. And the author should analyze the relationships between RH and sublimation, between albedo and sublimation, between Sin and sublimation, between Sout and sublimation, between Lin and sublimation, Lout and sublimation between D and Tair, between D and Ts, and between D and RH. I guess that albedo is an important factor that contributes to the interannual changes in sublimation by changing Ts. The concrete results are depending on your further analysis.

We agree with Reviewer 1 that $CF/S_{in}$ is not the main factor for sublimation. Therefore, we did not use such statements anywhere in the manuscript. However, we did mentioned: *'Cloud cover, on the other hand, has a significant impact on the primary meteorological variables, particularly $S_{in}$, $T_s$ and $q_s$.'* in Line No. 444-445. The observation was based on (i) the correlation coefficient (*r*) analysis (Fig. 10; Sect. 4.5 in revised manuscript), (ii) difference in *LE* magnitude in clear-sky and overcast conditions (Sect. 4.4 in revised manuscript) and interannual correlation of sublimation and meteorological variations (Sect. 4.6 in revised manuscript; Table S5).

Concerning the main factors of sublimation, we note that the sublimation is governed by a combined effect of different meteorological variables, primarily the vertical moisture $(q - q_s)$ and temperature $(T_{air} - T_s)$ difference/gradients, wind speed and the state of the surface boundary layer (stability). This is supported by multiple regression and variance analysis presented in Table 3 (revised manuscript; shown below). The multiple linear regressions analysis showed $q - q_s$, $T_{air} - T_s$, $u$ and $T_s$ together are the best sublimation predictors in clear-sky conditions (95%), overcast conditions (89%) and is applicable for all-data (without $CF$ filter; 92%). Considering two combined predictors, $q - q_s$ and $u$ explains the highest variance (> 80%) in sublimation for clear-sky, overcast and all-data conditions. However, individually, sublimation did not show strong correlation with any meteorological variables (Fig. 10 in revised manuscript, a copy below) except $q - q_s$, $T_{air} - T_s$, $T_s$ and $q_s$ which are the direct variables. All these correlation coefficients were based on half-hourly datasets for the daytime (between 09:00 and 16:00 IST).

Indeed, albedo is an important variable in sublimation, with a stronger correlation in clear-sky conditions ($r = -0.29$; Fig. 10 below). In overcast conditions, however, albedo has little impact on sublimation ($r = -0.02$).

Considering your suggestion, we developed an interannual correlation analysis based on cumulative sublimation and meteorological variables (n = 11 years; Table S5; a copy below). Inter-annual correlation analysis showed $T_s$ ($r = 0.85$; $p < 0.01$) correlates the highest with cumulative sublimation, followed by $S_{in}$ ($r = 0.79$; $p < 0.05$) and $RH > 80\%$ ($r = -0.76$; $p < 0.01$). This suggests that on an interannual scale, high $T_s$ (through higher $S_{in}$) and low near-surface moisture conditions supports sublimation.

Overall, we find that near-surface temperature $(T_{air} - T_s)$ and moisture gradients $(q - q_s)$, along with wind speed, are important factors in sublimation, while cloud cover shapes the meteorological variables. We have revised our discussion following the results discussed above. We would like to invite the reviewer to go through the revised manuscript sections (particularly Sect. 4.5, 4.6 and 5.1) for the meteorological factors role in sublimation.

Below we highlighted the concluding sentences in different sections regarding the main sublimation factors:

*Line No. 417-419 (Sect. 4.5):*
*'Overall, we noted that at the sub-hourly scale near-surface moisture availability (through q - $q_s$) plays a bigger role in determining the magnitude of LE, with the combined effects from several meteorological variables, particularly $q_s$, $T_s$ and u.'*

*Line No. 443-445 (Sect. 4.6):*

*'This suggests that on an interannual scale, high $T_s$ (through higher $S_{in}$) and low near-surface moisture conditions support sublimation. Cloud cover, on the other hand, has a significant impact on the primary meteorological variables, particularly $S_{in}$, $T_s$ and $q_s$.'*

*Line No. 531-533 (Sect. 5.1):*

*'Overall, we conclude that near-surface moisture availability (through $q$ - $q_s$) plays a major role in governing the magnitude of LE at the AWS-M at different temporal scales, while moisture availability was influenced and conditioned by a number of meteorological variables, notably $S_{in}$, $u$, $q_s$, and $T_s$.'*

**Table 3 (in revised manuscript).** Summary of the multiple linear regression analysis (k-fold (k = 10) cross-validation) of sublimation rate and combined meteorological variables. Total n = 13217, 2708 and 2063 half-hourly data points for all-data, clear-sky and overcast conditions, respectively, between 09:00 and 16:00 IST for DJFMA (2009-2020). The *p*-value of $r^2$ was always < 0.001.

| Variable | $r^2$ cross-validation | | |
|---|---|---|---|
| | **All-data** | **Clear-sky** | **Overcast** |
| $T_s$, $u$ | 0.53 | 0.69 | 0.44 |
| $T_{air}$, $u$ | 0.10 | 0.17 | 0.30 |
| $q$, $u$ | 0.03 | 0.15 | 0.15 |
| $q_s$, $u$ | 0.58 | 0.71 | 0.47 |
| $u$, $T_{air}$-$T_s$ | 0.58 | 0.75 | 0.29 |
| $u$, $q$-$q_s$ | 0.86 | 0.85 | 0.84 |
| $q$, $u$, $T_{air}$ | 0.26 | 0.21 | 0.34 |
| $q$, $u$, $T_s$ | 0.79 | 0.82 | 0.71 |
| $q_s$, $u$, $T_{air}$ | 0.77 | 0.90 | 0.51 |
| $q_s$, $u$, $T_s$ | 0.59 | 0.71 | 0.48 |
| $T_{air}$-$T_s$, $q$-$q_s$, $u$ | 0.92 | 0.95 | 0.89 |
| $T_{air}$-$T_s$, $q$-$q_s$, $S_{in}$ | 0.85 | 0.85 | 0.67 |
| $T_{air}$-$T_s$, $q$-$q_s$, $L_{in}$ | 0.84 | 0.85 | 0.67 |
| $T_{air}$-$T_s$, $q$-$q_s$, $R_{net}$ | 0.85 | 0.86 | 0.70 |

[Figure]

**Figure 11 (in revised manuscript).** Pearson's correlation coefficient (*r*) matrix of various meteorological and SEB components at the AWS-M in clear-sky and overcast conditions between 09:00 and 16:00 IST, 2009-2020. Number (n) of half-hourly data points are shown on top of the panels.

**Table S5 (in supplementary file).** Interannual correlation coefficient ($r$; n = 11) between cumulative sublimation ($S_c$) and primary meteorological variables for 2009-2020. '*' refers to p < 0.05.

| | RH > 80% | $T_s$ | $T_{air}$ | $u$ | RH | $S_{in}$ | CF |
|---|---|---|---|---|---|---|---|
| $S_c$ | -0.76* | 0.85* | -0.15 | -0.10 | -0.50 | 0.79* | 0.56 |

6. Discussion: I sometimes feel confused about the sentences in the discussion. Take section 5.3 for example. The author said that sublimation during the summer- monsoon season was lower, which could be due to the ISM-driven warm and moist atmosphere in the southern slope of the HK region. However, sublimation is higher at very high altitudes despite high summer-monsoon humidity, e.g., East Rongbuk Glacier site (6523 m a.sl.). What is the main point of the author? When author compared their study with other studies, the author should note the spatial and temporal scales. Some studies used the glacier-wide values, while others used point values. Some studies used the low-altitude values, while others used the high- altitude values. Some studies used the annual values, while others used winter values. These data with different scales are incomparable. The author should select these data carefully.

We thank you for the comment. We rephrased the respective sentence in the revised manuscript. Now it reads as:

*Line No. 596-600:*
*'Sublimation rate during the summer-monsoon season, in general, was lower than that of winter (Table 4; also Litt et al., 2019), which could be due to the warm and moist atmospheric conditions driven by the ISM. Despite high summer-monsoon humidity, sublimation is higher at higher altitude sites, such as in the East Rongbuk Glacier site (6523 m a.sl.). This is most likely a result of the strong winds and low air vapour pressure at very high altitudes, which promote sublimation.'*

Regarding the heterogeneous spatial and temporal scale of our comparison, we would like to highlight that sublimation is poorly investigated and understood across the HK region as compared to general glacier SEB studies. The available datasets are heterogeneous from the spatial and temporal scale point of view in the HK region. For example, only a single study in the Himalaya (Stigter et al., 2018) and a few studies in Tibet (Guo et al., 2021; Zhu et al., 2020) have discussed sublimation in detail. Also, in some of the studies meteorological values are not clearly defined or shown (e.g., in Dongkemadi Glacier in central Tibet; Liang et al., 2018). Furthermore, most studies in the HK and Tibet regions have focused exclusively on the summer season, considering the importance of summer SEB in melt modelling. Therefore, it is extremely hard to make an exhaustive comparison with consistent spatial or temporal scale based on limited available studies. This is the main reason that we have selected all available studies in this region and compare those

values to draw a general overview on the sublimation rates across HK and High Mountain Asia (HMA).

In order to clearly highlight the differences in spatial and temporal scales of existing sublimation studies, we added a few sentences in the revised manuscript in the respective section (Sect. 5.3). It reads as:

*Line No. 580-583:*
*'The existing sublimation studies in the HK and HMA are not uniform with respect to the spatial and temporal scales, which makes it difficult to compare sublimation and associated processes consistently. However, it is worthwhile to use these existing sublimation datasets for comparison, not to conduct a thorough and rigorous comparison, but to qualitatively address the sublimation process in the region.'*

Considering your suggestion, we have revised the respective Table (Table 4 in revised manuscript) and texts slightly for a consistent/similar comparison of the sublimation rates based on available studies. We would like to invite the reviewer to go through the revised comparison section.

**Minor comments:**
Line 32: wind-driven transport can cause accumulation in some sites.

The sentence was framed from the ablation point of view. However, to give this sentence a bit more ablation perspective, we have revised it and now reads as:

*Line No. 33:*
*'..wind-driven transport/erosion—lead to the loss of snow and ice mass..'*

Line 121-123: How do you get albedo in the night? Thus, what is your surface albedo threshold value in the night which is used to discern snow or bare-ground?

We filtered the snow-covered period based on the daytime surface albedo ($\alpha_{acc}$) ≥ 0.4. We revised the sentence for clarity and now it reads as:

*Line No. 33-34:*
*'We filtered the snow-covered period for SEB based on the daytime surface albedo threshold value above 0.4 at the AWS-M (the mean bare-ground/snow-free surface albedo was lesser than 0.25 for July-August; 2009-2020).'*

Line 153: Please explain the physical significance of Fsurface. If Fsurface is larger than 0, does melt occur at that time?

We revised the respective section and included a dedicated sentence mentioning the physical significance of residual energy ($F_{surface}$). The sentence reads as:

*Line No. 175-176:*
*'When $F_{surface}$ is larger than 0 W m$^{-2}$ (towards positive), it will get directed towards the surface/snowpack and warm it up until it reaches the melting point ($T_s = 0°C$), and then surplus $F_{surface}$ will cause melting (Hock, 2005).'*

Line 209-210: Can you analyze the difference between infrared measured Ts and Ts derived from Lout? Please list the figure. Is the emissivity of bared-ground similar to that of snow cover? This is important for the author to calculate Ts from Lout.

We have calculated the difference between infrared measured $T_s$ and $T_s$ derived from $L_{out}$ (shown in Table R1, below). The requested comparison figure is presented below (Fig. R3). $L_{out}$-based $T_s$ was derived using the Stefan-Boltzmann equation for the snow surface, with emissivity of 1 (following Hock and Holmgren, 2005; Wagnon et al., 2003). We observed the least root mean square error (RMSE = 0.23°C) and mean absolute error (MAE = 0.06°C) for emissivity = 1. We also observed that as emissivity decreases, RMSE and MAE increase considerably (Table R1).

We would also like to point out that we derived $T_s$ from $L_{out}$ only to compare it to the measured $T_s$. We did not use this $L_{out}$-based $T_s$ in any of our SEB/sublimation calculations, therefore it has no impact on our results.

**Table R1.** Comparison of RMSE and MAE for different snow emissivity.

| Emissivity | $r^2$ | RMSE [°C] | MAE [°C] |
|---|---|---|---|
| 1 | 0.99 | 0.23 | 0.06 |
| 0.99 | 0.99 | 14.09 | 14.08 |
| 0.98 | 0.99 | 27.39 | 27.37 |
| 0.97 | 0.99 | 39.98 | 39.95 |

Considering your above comment, we have revised the respective sentence in the revised manuscript and now it reads as:

*Line No. 224-226:*
*'$T_s$ was directly used from the measurement by an infrared radiometer (Table 1). The correlation between infrared measured $T_s$ and $T_s$ derived from $L_{out}$ (using Stefan-Boltzmann equation for the snow surface with emissivity of 1 following Hock and Holmgren, 2005) was $r^2 = 0.99$ (p < 0.001) with RMSE = 0.23°C.'*

[Figure]

**Figure R3.** Comparison of half-hourly values of the infrared measured $T_s$ vs $L_{out}$-based $T_s$ at the AWS-M site.

Line 312-313: Which components in Rnet are more important in playing an essential role in governing the turbulent fluxes? And the author should indicate the timescale.

On an interannual scale, $S_{in}$ showed stronger indirect relationship with *LE* and *H* ($r$ = -0.80 and -0.61, respectively; $p < 0.05$) than $L_{in}$ ($r$ = -0.36 and -0.39, respectively; not significant). Whereas, in half-hourly scale, in clear-sky conditions, $S_{in}$ and $L_{in}$ both have shown a nearly similar impact on *LE* ($r$ = -0.25, -0.26 and 0.29, respectively). In overcast conditions, impact of $S_{in}$ and $L_{in}$ equally rises ($r$ = -0.42 and -0.41). These analyses are further discussed in Sect 4.3 and Sect. 4.5 in the revised manuscript.

We have highlighted the timescales of our analysis in the respective sentence and sections. The revised texts read as:

*Line No. 348-349 (**Sect. 4.3. Seasonal and interannual variation of SEB components**):*
*'$S_{in}$ showed stronger indirect relationship with LE and H ($r$ = -0.80 and -0.61, respectively; $p < 0.05$) than $L_{in}$ ($r$ = -0.36 and -0.39, respectively; not significant).'*

*Line No. 415-417 (**Sect. 4.5. Turbulent heat fluxes under different cloud conditions**):*
*'At the sub-hourly scale, neither $R_{net}$ nor $S_{in}$ and $L_{in}$ can adequately explain turbulent fluxes in both overcast and clear-sky conditions ($r$ = $< 0.50$; Fig. 11).'*

Line 313-314: I can not understand this sentence.

We have removed this sentence in the revised manuscript.

Line 325: What do you mean about the different colors of lines in Figure 6?

The different colour lines in Figure 6 (original manuscript) define different years from 2009/10 to 2019/20. In the revised manuscript, we have combined both Fig. 5 and 6 (original manuscript) into a single figure (Figure 5 in revised manuscript) and it contains a legend for all coloured lines. A copy of the revised figure (Figure 5) is shown under your main comments no. 3 (above).

Line 359. Please add the "in the daytime" in the title of section 4.5.

Revised it as suggested.

Line 363-364 Why precipitation is higher in February and March than in January and April? High precipitation always means high cloud cover. This is different from your results of CF.

We thank the reviewer for the question and concern. In Fig. 8 (in revised manuscript, a copy shown below), we showed that February was the second cloudy (overcast) month, which is consistent with February having the second highest precipitation amount (24% of winter; Table S3 in revised manuscript). In January, more hours were cloudy, but only accounted for 19% of the total winter precipitation. This could be partly explained by the average moisture content in January (0.8 g kg$^{-1}$) which was ~30% lower than in February (1.1 g kg$^{-1}$) (see Table S3 in revised manuscript). In addition, it should be worth mentioning here that Fig. 8 is based on n = 8191 half-hourly data points, which was extracted from n= 23903 half-hourly data points following clear-sky ($CF < 0.2$) and overcast ($CF > 0.8$) filters. Night values were neglected in Fig. 8 because our $CF$ calculation is based on $S_{in}$ data which are not available during night. Since, we do not have the cloud information in nights and transition hours (for total 16 hours), it is difficult to understand and correlate the precipitation value with cloud cover.

In addition, in-situ precipitation data from the Geonor station was available only for five discontinuous hydrological years, besides that the five years of precipitation data is not sufficient to discern the relationship between cloud cover/fraction and precipitation intensity in the region.

[Figure]

**Figure 8 (in revised manuscript).** Monthly fraction of clear-sky (CF ≤ 0.2) and overcast (CF ≥ 0.8) conditions at the AWS-M. Fraction percentage is calculated from n = 5810 clear-sky and n = 2381 overcast observations from total n = 23903 half-hourly values between 09:00 and 16:00 IST (DJFMA, 2009-2020).

Line 376: 3 times lower?

We modified the sentence and now reads as:

*Line No. 377-378:*
*'In clear-sky, the mean daytime H was -66 W m$^{-2}$ which is three times more negative compared to overcast conditions (-21 W m$^{-2}$).'*

Line 423: 145 ± 25 mm w.e. a-1?

Revised it as suggested.

Section 4.7: There is no section 4.7.2 in this part. The author can merge section 4.7 and section 4.7.1 as one part.

Thanks for the suggestion. We have merge it and renamed the section as:

*'4.6 Sublimation and its relationship with meteorological variables'*

Line 451-452: I can not agree with the author, because we can not find that low Tair (-5°C and -10°C) corresponds to high Ts (0°C and -10°C) for the same time. From figure14b, we can only

find that sublimation was the larger when Tair ranged between -5°C and -10°C (compared to Tair in other values). This is similar to the Ts. Thus, the content in Line 451-452 is not correct.

Thanks for catching this issue. This sentence has been corrected and now reads as:

*Line No. 459-461:*
*'Sublimation was the largest when $T_{air}$ ranged between -5°C and -10°C and also when $T_s$ ranged between 0°C and -10°C (Fig. 12; Fig. 13B and C). Whereas, sublimation was considerably lower when moisture availability was higher, $T_s$ was significantly lower, with very strong u (Fig. 12; Fig. 13).'*

To show this observation clearly, we made two more meteorological clusters (i.e., $T_s$ > -10°C and $T_s$ < -10°C) in the existing Fig. 12 (in revised manuscript, a copy shown below). From Fig. 12 (bottom panels) it is clear that sublimation was almost half when $T_s$ < -10°C compared to $T_s$ > -10°C. This is also evident in Fig. 13 (in revised manuscript, a copy shown below).

[Figure]

**Figure 12 (in revised manuscript).** Half-hourly daytime (09:00-16:00) records of sublimation (red), wind speed (blue) and specific humidity (green) at the AWS-M for different clusters: no filter, $u > 10$ m sec$^{-1}$, $q > 2$ g kg$^{-1}$, $< 1$ g kg$^{-1}$, $T_s > -10°C$ and $T_s < -10°C$. Data period: DJFMA, 2009-2020. Number of data-points n=30257, 2347, 12295, 9762, 10552 and 12734 for no filter, $u > 10$ m sec$^{-1}$, $q > 2$ g kg$^{-1}$, $< 1$ g kg$^{-1}$, $T_s > -10°C$ and $T_s < -10°C$, respectively.

[Figure]

**Figure 13 (in revised manuscript).** Scatter plot of $u$, $q$, $T_{air}$, $T_s$, $CF$, $S_{in}$ and $L_{in}$ against sublimation rate at the AWS-M. The colour of the data points refers to the measured wind speed ($u$). Total n = 14088 half-hourly data points between 09:00 and 16:00 IST for DJFMA (2009-2020).

Line 480-481: What is your timescale?

Our analysis is based on half-hourly $LE$ datasets, however for a longer/seasonal perspective, we averaged it for daily, monthly, and seasonal (DJFMA) timescale as well. In the current section (Sect. 5.1 in revised manuscript), we have discussed $LE$ from an overall/holistic perspective to summarise the factor controlling $LE$ at the AWS-M site.

Line 544: Do you want to say that sublimation during the summer-monsoon season was lower than that during winter?

Yes. We revised it for a better read.

*Line No. 596-597:*
*'Sublimation rate during the summer-monsoon season, in general, was lower than that of winter (Table 4; also Litt et al., 2019), which could be due to the warm and moist atmospheric conditions driven by the ISM.'*

Line 545-547: The studies of Mölg et al. (2012) and Li et al. (2018) are in the south and central Tibet, respectively. They do not study the glaciers in the northern slope of the HK region.

Thanks for pointing this out. Previously we missed to cite the SEB studies which are from the northern slope of the Himalaya, for example, on Naimona'nyi and East Rongbuk glaciers (e.g., Zhu et al., 2021; Liu et al. 2021). We revised the sentence and now reads as:

*Line No. 600-601:*
*'The high moisture from ISM also impacts Tibetan glaciers, particularly those located in the northern slopes of the Himalaya (Zhu et al., 2021; Liu et al., 2021) and central Tibet (Mölg et al., 2012; Li et al., 2018).'*

Line 547-548: Can you explain the phenomenon that you found in these sentences?

In this sentence, we did not intend to discuss any phenomenon, but to point out (from the existing studies) that sublimation rates in the central Himalaya are relatively higher during post-monsoon and pre-monsoon (for example in Yala and Mera glaciers; Table 4 in revised manuscript).

Although near-surface moisture (*RH*) is relatively higher in the central Himalaya during post-monsoon and pre-monsoon (maybe because it is close to the Bay of Bengal) than in winter season. The sublimation rates are comparatively higher (Table 4; Yala and Mera glaciers). We assume it because of its high-altitude location (for Yala it was > 5300 m a.s.l. and for Mera it was >5300 m a.s.l. and > 6500 m a.s.l.), where strong wind or snow blowing could have increased sublimation considerably.

We have revised the respective sentences and now reads as:

*Line No. 601-608:*
*'In the Nepalese central Himalaya, we note a higher sublimation value of 2.4 and 1.8 mm d$^{-1}$, respectively on the Yala Glacier during the post- and pre-monsoon seasons (Table 4). Litt et al. (2019) also reported a significantly higher sublimation rate of 7.1 and 1.9 mm d$^{-1}$, respectively during post- and pre-monsoon seasons on the Mera Glacier in Nepal. Such higher sublimation rates on Yala and Mera glaciers are unique, particularly during post- and pre-monsoon seasons when air vapour pressure/specific humidity is higher than that in winter season (Shea et al., 2015; Perry et al., 2020). Nevertheless, such higher sublimation can also be partially attributed to snow blowing/redistribution at such high-altitude sites (Barral et al., 2014; Wagnon et al., 2013; Huintjes et al., 2015b).'*

Line 548-549: I do not find that the moisture content is relatively higher during post- and pre-monsoon on the Mera Glacier than that in winter in Table 5. And the altitudes are significantly different between the post- and pre-monsoon periods.

We have now updated Table 4 (in revised manuscript, a copy shown below) with *RH* and wind speed values for the Yala and Mera glaciers for the pre- and post-monsoon seasons. Table 4 shows *RH* for Yala and Mera glaciers are considerably higher (~70%) in pre-monsoon and close to 50% in post-monsoon (Litt et al., 2019).

Post-monsoon sublimation rate was not available for 6543 m a.s.l. AWS site from the Mera (Litt et al., 2019). So, to keep it consistent, now we have used the Mera Glacier sublimation rates from a single site: 5360 m a.s.l. where both pre- and post-monsoon seasons' sublimation rates are available. Updated Table 4 shown below:

**Table 4 (in revised manuscript).** Compilation of sublimation rate across the HMA region. '*' refers to the evaporation values. Do' refers to the same method as in the row immediately above.

| Site | Altitude (m a.s.l.) | Region | Period of obser-vation | Season approx. to Chhota Shigri | Surface | Method | $S$ (mm d$^{-1}$) | RH (%) | u (m s$^{-1}$) | Reference |
|---|---|---|---|---|---|---|---|---|---|---|
| **Tibetan Plateau** | | | | | | | | | | |
| Zhadang | 5665 | Nyainqen-tanglha Shan | 1 October to 31 May, 2008-2013 | Winter | Glacier-wide | Bulk-aerody-namic | 0.5 | 44 | 3.6 | Zhu et al. (2018) |
| Muztag Ata No. 15 | 4400 | Eastern Pa-mir | 1 October to 31 May, 2008-2013 | Winter | Glacier-wide | Do | 0.7 | 42 | 6.4 | Zhu et al. (2018) |
| Parlung | 4800 | Southeast TP | 1 October to 31 May, 2008-2013 | Winter | Glacier-wide | Do | 0.4 | 64 | 3.4 | Zhu et al. (2018) |
| Muji | 4685 | Northeast Pamir | 1 October to 31 May, 2011- 2017 | Winter | Glacier-wide | Do | 0.5 | 50 | 4 | Zhu et al. (2020) |
| Qiangtang No. 1 | 5882 | Inland TP | 1 October to 31 May, 2012-2016 | Winter | Glacier-wide | Do | 0.4 | 46 | 6.8 | Li et al. (2018) |
| Guliya Ice Cap | 6000 | Kunlun Shan | 1 October to 31 May, 2015-2016 | Winter | Glacier-wide | Do | 0.3 | 67 | 7.9 | Li et al. (2019) |
| Dongkemadi | 5600 | Central TP | 7 October 1992 to 4 May 1993 | Winter | Glacier ELA | Do | 0.2 | - | 4.3 | Liang et al. (2018) |
| August-one | 4817 | Qilian Mountains | Jan-May, Oct-Sept, 2016-2020 | Winter | Glacier | Do | 0.4 | 68 | 6.9 | Guo et al. (2021) |
| **Himalaya** | | | | | | | | | | |
| Pindari | 3750 | Central Himalaya | December 2016 to February 2017 | Winter | Medial mo-raine | Monin-Obu-khov theory | ~0.3 | 55 | 1.2 | Singh et al. (2020) |
| Dokriani | ERA5 grid point | Do | 1 November 1979 – 30 October 2020 | Winter | Glacier-wide | Bulk-aerody-namic | ~1.2 | ~45 | ~7 | Srivastava and Azam, 2022 |
| Yala | 5350 | Do | 15 October 2015 to 20 April 2017 | Winter | Glacier/ab-lation zone | Eddy-covari-ance | 1 | ~40 | ~2.5 | Stigter et al. (2018) |
| Yala | 5330 | Do | 1 October to 15 November, 2012-2017 | Post-monsoon | Glacier/ab-lation zone | Bulk-aerody-namic | 2.4 | ~49 | ~1.8 | Litt et al. (2019) |
| Yala | 5330 | Do | 10 May to 5 June, 2012-2017 | Pre-mon-soon | Glacier/ab-lation zone | Do | 1.8 | ~77 | ~1.9 | Do |
| Mera | 5360 | Do | 1 October to 15 November, 2013-2016 | Post-monsoon | Glacier/ab-lation zone | Do | 1.9 | ~46 | ~2.8 | Do |
| Mera | 5360 | Do | 10 May to 5 June, 2013-2016 | Pre-mon-soon | Glacier/ab-lation zone | Do | 3.3 | ~72 | ~2.3 | Do |

| | | | | | | | | | | |
|---|---|---|---|---|---|---|---|---|---|---|
| Lirung | 4250 | Do | 26 September to 12 October 2016 | Post-monsoon | Glacier debris | Eddy-covariance | 1.8-2.8* | ~60 | ~3 | Steiner et al. (2018) |
| South Col, Everest | 7945 | Do | 22 May to 31 October 2019 | Summer-monsoon | Ice-rock surface | Bulk-aerodynamic | ~0.8 | ~60 | 6.3 | Matthews et al. (2020) |
| East Rongbuk | ~6500 | Do | 28 April to 2 May 2008 | Pre-monsoon | Glacier | Lysimeter | 1.9 | - | - | Yang (2010) |
| East Rongbuk | 6523 | Do | 1 May to 22 July 2005 | Summer-monsoon | Glacier | Bulk-aerodynamic | 0.05-1.2 | 60 | 4.2 | Liu et al. (2021) |
| Xixibangma | 5900 | Do | 23 August to 29 September 1991 | Summer-monsoon | Glacier | Calculated | 0.02 | 36 | 5.9 | Aizen et al. (2002) |
| Naimona'nyi | 5543 | Do | 1 October 2010 to 31 May 2018 | Winter | Glacier-wide | Bulk-aerodynamic | 0.6 | 34 | 5.5 | Zhu et al. (2021) |
| Chhota Shigri | 4670 | Western Himalaya | 1 Dec 2012 to 29 Jan 2013 | Winter | Glacier/ablation zone | Do | 0.8 | 44 | 4.9 | Azam et al. (2014a) |
| Chhota Shigri | ERA5 grid point | Do | 1 October 1979 – 30 September 2020 | Winter | Glacier-wide | Bulk-aerodynamic | 0.7 | ~40 | ~5.7 | Srivastava and Azam, 2022 |
| Chhota Shigri | 4863 | Do | 1 December to 30 April, 2009-2020 | Winter | Seasonal snow on moraine | Do | 1.1 | 43 | 5 | This study |

Line 550-551: What is the cause for the differences that the authors found in this sentence?

Thanks for pointing this out. Here we compared our study (for DJFMA) with wet/moist season's sublimation rates without any data/analysis from this study site. Therefore, we have removed the sentence from the revised manuscript.

Line 553-555: I cannot understand what you want to say.

We revise it, as:

*Line No. 608-610:*
*'Overall, dry air, low atmospheric pressure and high wind speeds are suitable conditions for sublimation, as reported from various high-altitude sites in the HMA (Matthews et al., 2020; Litt et al., 2019; Stigter et al., 2018; Zhu et al., 2018) and everywhere in the world (Wagnon et al., 1999; Cullen et al., 2007; Fyffe et al., 2021).'*

Line 560-564: These sentences have no relationship with the title 'Sublimation fraction to winter snowfall and its importance'.

Thanks for this suggestion. We have removed the sentences from the respective paragraph and revised it.

Line 569: Why sublimate is higher in the northwestern part of the HK than that in the other parts of the HK region?

It is because the atmospheric conditions of the northwestern part of the HK, as well as the west Tibet, are very dry and arid compared to other parts of the HK, such as the Eastern or Central Himalaya, where climate is more humid and monsoon precipitation is higher. Dry air and low

atmospheric pressure create a steep near-surface moisture gradient, which fosters strong sublimation. To clarify this in the revised manuscript, we incorporated a dedicated sentence for this, and it reads as:

*Line No. 640-644:*
*'Although there are limited observations available from various parts of the Himalaya and HMA, these observations show that the sublimation fraction to winter/annual snowfall/precipitation is higher in the north-western part of the HK and western Tibet (e.g., Zhu et al., 2020; Gascoin, 2021). This is likely due to the atmospheric condition of the north-western part of the HK and western Tibet which is drier than the eastern and central Himalaya. Dry atmospheric conditions favour higher sublimation than the wet due to high near-surface humidity gradients.'*

Line 579-580: Such a higher sublimation fraction? You mean that the sublimation fraction is higher on Qiangtang No 1 Glacier than other glaciers on the TP. Have you compared the meteorological data at the Qiangtang No. 1 Glacier to that on other glaciers?

In this section we did not compare meteorological conditions of different glaciers. We intended to limit our discussion on the sublimation fractions and its variation across the region. In other glacier/areas, sublimation fraction was comparable (between ~16% and ~60%) and are not very much contrasting, except at the Qiangtang No 1 Glacier which where sublimation fraction is 65-169%. Therefore, to briefly discuss the contrasting conditions at Qiangtang No 1, we presented their meteorological conditions (wind speed, *RH* and snowfall values from Table 3 in Li et al., 2018) in the discussion. This comparison briefly points out the contrast, which we thought to be interesting for the readers.

Line 610-611: This result disagreed with your description in section 5.4. Sublimation fraction to winter snowfall is higher on Qiangtang No 1 Glacier than that on Chhota Shigri Glacier.

Thanks for pointing this out. We revised the sentence and now reads as:

*Line No. 675-677:*
*'The cumulative DJFMA sublimation was $145 \pm 25$ mm w.e. a$^{-1}$, corresponding to 16-42% of the fraction of winter snowfall at the AWS-M site, which is relatively higher than observed in other sites across the HK region, with considerable interannual variations and is lower than a few of the Tibetan glacier sites.'*

Line 620: There are more than 10 published works about Chhota Shigri Glacier. However, the meteorological data for that glacier is still not open to scientists in the world.

We are not allowed to share the datasets in open platforms because the data belongs to the several funding agencies (Department of Science and Technology (Govt. of India), MoES, SAC-ISRO, and several bilateral projects: Indo-French, Indo-Norwegian). We oversighted this while responding to the reviewer comments in the open discussion forum. However, we can share the datasets on a request basis. The codes for the SEB calculation and figures are uploaded and shared through the open platform Zenodo (https://doi.org/10.5281/zenodo.6804947; Mandal et al., 2022).

**Response to Reviewer 2**

Below we provide our responses **(in red text)** point-by-point to each comment from the reviewer **(in black text)**. *Italic texts* are used to highlight specific changes in the updated manuscript.

The authors present a very clear study on multiple winters worth of energy balance data from a site in the Western Himalaya. They show the consistent importance of sublimation during snow cover times and find results that generally match well with previous studies in the field. The work is very timely and the numbers found here will guide research conducted on the larger scale that is not able to include the process on a distributed scale. The paper is very clearly written, well supported with data and clear Figures that leave only very few general comments from my side which I detail below and which I hope you can address. I have a number of minor comments at the end. I applaud the authors for the field work that this work is based on as well as the clear way of presenting the results. It is important work and I think this should be an important paper in the TC library in future.

We sincerely thank the Reviewer 2 for evaluating our manuscript, suggestions, and the positive feedback on our study. All specific comments have been addressed in detail below, and we have also highlighted (point-wise) the major revisions that we have carried out in the revised manuscript in response to major comments:

- Included a detailed discussion on the influence of cloud cover on sublimation, based on correlation analysis and comparison of existing cloud and sublimation studies from the Himalaya/Tibet region,

- Included a future perspective on sublimation sensitivities to meteorological variables, which may be useful to readers in getting an idea of future sublimation and subsequent changes in terms of SEB of snow/glacier surfaces,

- In addition, the result and discussion sections have been significantly revised, with new text and restructuring in response to Reviewer 1's suggestions.

**General:**

In the Discussion I would expect more discussion of the role of cloud cover, which as you note is important but to me has a surprisingly low correlation and obviously wind plays a very different role in these regimes (your Figure 12). Could you compare the relative cloud cover to the other sites, or at least the ones from (Guo et al., 2021; Stigter et al., 2018). Not to cite here as still in review but (Conway et al., 2022) also provides some new great insights in this direction. I would hope to learn here how different I can expect my sublimation rates to be when I work in a different regime of overcast conditions.

We thank Reviewer 2 for the suggestions. We have considered the suggested studies to compare with our results and expanded the discussion section (Sect. 5.1). Some of the newly incorporated texts discussing the respective aspect are mentioned below.

Conway et al. (2022) focused on the melt seasons SEB and associated meteorological characteristics, but they also discussed the complex interaction between cloud cover and overall SEB of the glaciers across various sites, including four glacier sites in the Himalaya. The findings of Conway et al. (2022) are consistent with our findings in general. They also point out that at most of their study sites, increased cloud cover decreases the magnitude of $LE$ and $F_{surface}$. At very high-altitude sites (e.g., Mera, Zongo) they found that $LE$ is still negative (that means sublimation) in overcast conditions (at $CF > 0.7$, mean melt-season's $LE$ was ~-60 W m$^{-2}$; Fig. A5 in Conway et al., 2022). Overall, their findings show that in overcast conditions, near-surface meteorology (particularly near-surface vapour pressure and relative humidity) is significantly altered which limits higher magnitude of radiation and turbulent heat fluxes. Cloud cover, on the other hand, has little impact on wind speed and $T_{air}$ at most sites, including the Chhota Shigri Glacier (Fig. 8 in Conway et al. 2022). Although the climatic setting varies greatly across the Himalayan region, and cloud cover's influence is complex, we highlight that overcast conditions lower the magnitude of sublimation, as shown by our study and Conway et al. (2022). We have cited Conway et al. (2022) in the Discussion.

*Line No. 520-531:*
*'We note the importance of cloud cover in modulating the surface atmosphere at the AWS-M site which favours sublimation, however, the correlation coefficient between CF and LE was poor (r = -0.09 and -0.16 in clear-sky and overcast conditions, respectively; Fig. 10). This is most likely due to the complex influence of cloud cover on meteorological variables, particularly $S_{in}$ and $L_{in}$. Cloud cover reduces $S_{in}$, which impede sublimation, but at the same time it also increases $L_{in}$, which promotes sublimation partly by raising the $T_s$. This is well-supported by the higher correlations between sublimation and $S_{in}$ and $L_{in}$, particularly in overcast condition (Fig. 10). Although Stigter et al. (2018) did not discuss the correlation between sublimation and cloud cover/factor at the Yala Glacier, they did indicate that sublimation was negligible or about zero on overcast days when humidity was higher. This is supported by the poor correlation of determination ($r^2 = 0.08$) between sublimation and RH at the Yala Glacier. Guo et al. (2021) also did not obtain a statistical relationship between sublimation and cloud cover, but they also noted a weak sublimation rate during cloudy months due to high moisture and warm conditions. Conway et al. (2022) also found that an increase in cloud cover decreases the magnitude of LE at four on-glacier Himalayan sites, including the Chhota Shigri Glacier.'*

Table 2: In text you say max T_a is 0.1, in Table 0.0

Thanks for pointing this out. We have revised Table 2 (original manuscript) with maximum $T_{air}$ as 0.1°C. Following Reviewer 1's suggestion, we have shifted Table 2 (original manuscript) to supplementary material (as Table S3).

Table 4: R2 for u is 0? I am also surprised that CF seems to be more correlated to sublimation in the transition phase than in overcast or clear sky condition. Can that be explained? I would have expected a higher correlation under overcast condition.

In the previous version of the manuscript, we showed Pearson's correlation coefficient ($r$) as well as the coefficient of determination ($r^2$) through linear regression analysis. Since we already have a dedicated analysis and figure showing $r$ (Figure 10; revised manuscript; Sect. 4.5), we planned to remove $r^2$/linear regression between sublimation and meteorological variables from Table 3 (revised manuscript; note: Table 3 was Table 4 in original manuscript). We only kept the multiple linear regression analysis in Table 3 (revised manuscript) to show the readers how a combined effect of meteorological variables influences sublimation. This way we still have the correlation analysis between sublimation and meteorological variation and discussion (Fig. 10; Sect. 4.5 dedicatedly) while skipping the discussion of $r^2$ for the same relationships. Using $r$ and $r^2$ for the same relationship is a little confusing and difficult to follow for the readers.

Indeed, the relationship between $u$ and sublimation is weak in both clear-sky and cloudy conditions ($r = 0.37$ and $0.33$ in clear-sky and overcast, respectively; Fig. 10 in revised manuscript). The absence of strong correlation between sublimation and $u$ is expected because a supportable condition for an enhanced sublimation was created by a combination of meteorological variables, primarily the vertical moisture and temperature gradient, wind speed and the state of the surface boundary layer (stability) (please refer to Sect. 5.1 in revised manuscript). The weak correlation between $u$ and sublimation can partly be explained by the highly heterogeneous wind speed at the AWS-M. For example, available observation from various studies showed that $u$ generally decreased in overcast conditions (e.g., Stigter et al., 2018; also in Conway et al., 2022 in several glacier sites). However, in overcast conditions we often had higher $u$ (Fig. 9 and Fig. 11; revised manuscript) due to westerly activities (discussed in Sect. 4.5 and 4.6; revised manuscript). This heterogeneity was the cause of weak correlation between $u$ and sublimation in part. In this regards, new study by Fugger et al. (2022) also reported that the relationship between $LE$ and meteorological variables was highly unpredictable, and $u$ failed to explain the variability of $LE$/sublimation at five on-site glacier studies in the central and eastern Himalaya (see their Fig. 9A).

Correlation between sublimation and $CF$ was also weak ($r = -0.09$ and $-0.16$ in clear-sky and overcast conditions, respectively; Fig. 10 in revised manuscript). This is likely due to the complex influence of cloud cover on meteorological variables, particularly $S_{in}$ and $L_{in}$. For instance, cloud

cover reduces $S_{in}$, which impedes sublimation, but at the same time it also increases $L_{in}$, which promotes sublimation partly by raising $T_s$.

We do observe a slightly higher correlation between sublimation and *CF* in overcast conditions ($r$ = -0.16) than clear-sky ($r$ = -0.09), but not that significant.

To give a thought to your concern (based on our observation in the original manuscript) that *CF* was more correlated in the transition phase, we analysed this relationship a bit further. We analysed the sublimation correlations for three more cloud conditions by binning *CF* for three more categories within the transition phase (i.e., $CF > 0.2 <= 0.4$; $CF > 0.4 <= 0.6$; $CF > 0.6 <= 0.8$). In those categories, we also did not find any strong correlation between sublimation rates and CF. The $r$ values were similar as in clear-sky and overcast conditions (not shown here). This is partially reflected in Fig. 13 (revised manuscript; a copy shown below).

[Figure]

**Figure 13 (in revised manuscript).** Scatter plot of $u$, $q$, $T_{air}$, $T_s$, $CF$, $S_{in}$ and $L_{in}$ against sublimation rate at the AWS-M. The colour of the data points refers to the measured wind speed ($u$). Total n = 14088 half-hourly data points between 09:00 and 16:00 IST for DJFMA (2009-2020).

Based on the above argument on the weak relationship of $u$ and *CF* with sublimation, we revised our discussion. We would like to invite the reviewer to go through the revised manuscript (Sect. 5.1; revised manuscript). The newly incorporated texts are highlighted below:

*Line No. 507-520 (Sect. 5.1):*

*'Stigter et al. (2018) and Guo et al. (2021) also reported a similar process where an integrated effect was responsible for higher sublimation in the Yala and August-One glaciers. The integrated effect of different meteorological variables in supporting sublimation also explains the weak correlation between LE/sublimation and u (r = < 0.40; Fig. 10). Stigter et al. (2018) and Guo et al. (2021) noted a strong direct relationship between LE and u, which does not agree with the present study. This could be partly explained by the highly heterogeneous u at the AWS-M (Fig. 13). For example, the available observations from different sites showed that u generally decreases in overcast conditions (e.g., Stigter et al., 2018; Guo et al., 2021; Conway et al., 2022). However, at the AWS-M, u was often higher in overcast conditions (Fig. 9; Fig. S5) due to westerly activities (discussed in Sect. 4.5 and 4.6). Very high u maintains a neutral stratification of the boundary layer resulting in a lower magnitude of LE. This heterogeneity is likely the cause of the weak correlation between u and sublimation in part. However, the highest multiple regression variance in combination with u (~90%; Table 3) in clear-sky and overcast conditions emphasise the importance of u in driving LE/sublimation. Fugger et al. (2022) also observed that the relationship between LE and meteorological variables is highly unpredictable, and u fails to explain the variability of LE at five on-glacier sites in the central and eastern Himalaya (see their Fig. 9A).'*

*Line No. 520-525 (Sect. 5.1):*

*'We note the importance of cloud cover in modulating the surface atmosphere at the AWS-M site which favours sublimation, however, the correlation coefficient between CF and LE was poor (r = -0.09 and -0.16 in clear-sky and overcast conditions, respectively; Fig. 10). This is most likely due to the complex influence of cloud cover on meteorological variables, particularly $S_{in}$ and $L_{in}$. Cloud cover reduces $S_{in}$, which impede sublimation, but at the same time it also increases $L_{in}$, which promotes sublimation partly by raising the $T_s$. This is well-supported by the higher correlations between sublimation and $S_{in}$ and $L_{in}$, particularly in overcast condition (Fig. 10).'*

L505ff/Figure 15: This is interesting – could you expand here what that means for a potential future change especially of T_air? Also in the text you mention the big sensitivity to T_s, but that under melting condition won't change much. It seems to be equally (or just slightly less) sensitive to T_air though, which likely will change. That seems important to me for future consideration.

Thank you for the suggestion. The future perspective of the sensitivities is interesting and worth expanding. Following your suggestion, we have expanded the discussion. The newly incorporated texts are presented below. We invite you to go through the revised respective section (Sect. 5.2; revised manuscript).

*Line No. 553-563 (Sect. 5.2):*

*'Another important aspect of the sensitivity to meteorological variables is related to the future atmospheric warming and its consequences to sublimation. $T_s$ exhibited a higher sublimation*

*sensitivity than Tair (Fig. 14), but under melting condition Ts will not change much because the temperature of the snow/ice surface cannot rise above the melting point (Ts = 0°C). However, relative potential changes in Tair are likely to be higher across the globe including in the Himalayan region (Hock et al., 2019; Krishnan et al., 2019). Therefore, sublimation sensitivity with respect to Tair could be a major concern in future, due to the expected warming. Considering a future Tair increase of ~0.3 ±0.2°C decade-1 for the Himalayan region (Ren et al., 2017; Krishnan et al., 2019), a crude estimate suggests a ~5% decrease in sublimation per decade from the snow/glacier surfaces. This could probably attribute to a lower energy sink through LE, which will boost the efficiency of Sin/Rnet resulting in more surface melt. However, since sublimation is a process driven by the combined effect of multiple meteorological variables, it remains to be seen how the sensitivity of a single variable influences the overall sublimation and associated processes.'*

**Minor comments:**

L20: replace 'consequently' with 'resulting in'

Done.

L21: 'largest fraction' or 'proportion'

We think 'proportion' would be a better choice. Thanks for the suggestion. Done.

L24: 'to the region'

Done.

L26: sublimation is a variable, not a parameter; remove the two 'the' articles

Done, thanks.

L40: 'more abundant'

Done.

L53: 'The contribution …is …'

Done.

L:57: 'poorly understood'

Done.

L71: Technically it has been applied (Sakai et al., 2004) although they did not term it sublimation and on this debris cover (as in (Steiner et al., 2018)) it is more an evaporative process. But this is a grey area, and at least our attempt to measure sublimation over snow with a pan lysimeter have simply been unsuccessful because they freeze and can't measure properly. You also later mention the PhD thesis by Yang (2010).

Thanks for the information.

L101: 'radiation', no need for a plural here

Done.

Table1: The superscript a at the bottom is missing. Also again I would use 'radiation' in singular

We will make sure the superscript is there, thanks. Changed it to 'radiation'.

L134: 'single-Alter-shielded'

Done.

L164: you use 'net radiation' here but earlier used net all-wave radiation'. I would go throughout for the shorter version.

We choose net radiation across the manuscript following your suggestion.

L166: The two sentences should be conjoined with comma or you need to restructure syntax

We have revised it following your suggestion.

L189f and in general: no need to include [in …] with the units

Done, we remove [in ...] here and elsewhere.

L229: remove 'equation by' or 'the equation by'

Done.

L292: I would leave 'snow cover' in singular

Done.

L299: does not

We removed this sentence from the revised manuscript considering Reviewer 1's suggestion to shorten the respective section.

L310: maybe rather 'down to'

Done, thanks.

L322: 'such a high contribution'

Done.

L336: remove 'thin'

Done.

Figure 11: Nice figure and just a pedantic comment – can you make Tair-Ts instead of Tair_Ts in the axis label? Also you introduce D here but only introduce it much later in the text (L447). Make sure to somehow introduce it earlier, otherwise as a reader I need to go looking forward in the text, which is awkward. The question is though why you show it at all here as it is just the reverse from q-qs – you could consider to just remove the column/row in both subfigures.

Thanks for pointing this out and the suggestion. We have revised the figure (Figure 10; revised manuscript; a copy shown below) as suggested and removed *D* from the figure (Figure 10). Also, considering *D* is already included in *LE* equation, we have decided not to use *D* at all, across the manuscript and therefore, revised the respective sections accordingly.

[Figure]

**Figure 10 (in revised manuscript).** Pearson's correlation coefficient (*r*) matrix of various meteorological and SEB components at the AWS-M in clear-sky and overcast conditions between 09:00 and 16:00 IST, 2009-2020. Number (n) of half-hourly data points are shown on top of the panels.

L433: 'restrict'

We removed this sentence from the revised manuscript.

L453: It is quite clear that D is directly positively related to LE as it is the main part of the equation/definition, so it can't really be any other way. I would remove this sentence.

We have removed this as suggested.

L523: remove one 'in this study'

Done.

L525/L540: maybe 'similar' or 'comparable' instead of 'identical'

Done. We choose 'similar'.

L577: 'with the major part'

Done.

L581: 'This supports …'

Done.

L600: 'impediment'

Done, thanks.

L603: maybe 'reducing by 70%' and 'raising by 25%'

Done, thanks.

L604: Bit confusing – restraining to what? Also '50% cloud fraction' to be clear.

We revised the respective sentences for clarity. Now the sentence reads as:

*Line No.: 666-667:*
*'The cloud cover also controls the meteorological condition favourable for turbulent heat fluxes and reduce their magnitude by larger than 50%.'*

L607: remove 'were'

Done.

L608: 'suggesting it is crucial for …'

Done.

L612f: remove 'significantly' – that is a hard term and you don't really show that here. I would also remove the part behind the semi-colon. That is always a given and a bit redundant. And you say the same in the following sentences already.

Done, we have revised it as suggested.

L620: Please provide this for the final version. It is a pity if such a statement remains without a link in a final publication.

The codes for the SEB calculation and figures are uploaded and shared through the open platform Zenodo (https://doi.org/10.5281/zenodo.6804947; Mandal et al., 2022).

Dear Suryanarayanan,

We thank you for taking interest in our work and commenting on it. Below we respond (in red text) to your concerns (in black text).

1. The definition of daytime seems to vary throughout the text, for example, in Figure 10, Table 3, Figure 13. I would guess the daytime time period was chosen solely based on solar angle and shadow effects. Why is this the case?

In Figure 13 (revised manuscript), the daytime was defined between 08:00 and 17:00 IST. Previously we thought that if we could show maximum day-hours to show how sublimation varies in different meteorological clusters. However, following your concern and to keep it uniform, we intend to make this daytime range consistent and therefore, considered daytime as 09:00-16:00 IST throughout the manuscript.

2. Beyond daytime, what is the value of the cloud factor? Is it not defined then?

We do not have cloud factor ($CF$) information for the night time because our $CF$ calculation is based on incoming short-wave radiation ($S_{in}$) data (please refer to Sect. 3.3; revised manuscript) which is unavailable during night. In addition, $S_{in}$ data is often uncertain during early morning and dusk hours when solar angles are flatter. Therefore, to avoid the steep valley wall's shading effect during the morning and evening time, we restricted our $CF$ calculation for the daytime only (09:00-16:00 IST). Therefore, we could not define $CF$ beyond daytime hours.

3. In Table 5, what does the abbreviation "Do" mean?

Thanks for catching this issue. In the original manuscript we forgot to mention this in the table caption. 'Do' refers to the same method as in the row immediately above.

---

## Referee Report (RR1)

**Review of manuscript:**

**"11-year record of wintertime snow surface energy balance and sublimation at 4863 m a.s.l. on Chhota Shigri Glacier moraine (western Himalaya, India)" by Arindan Mandal and co-authers**

Submitted to The Cryosphere

General comments:

I appreciate that the authors made a big revision to the manuscript and gave their kindly responses to my comments. The paper is much improved, although some of the limitations and concerns raised in the reviews remain. Below, I have listed a few minor points that could still be addressed but I believe that the paper is now basically suitable for publication.

**Minor comments:**

Line 20: How do you get that cloud cover also restricts the turbulent heat fluxes by around **50%** in this work? I do not find such data (50%) in the main text. The author could add some sentences to explain why cloud cover restricts the turbulent heat fluxes. Line 22-24: Please add the interpretation that why a strong control of cloud cover in shaping favorable conditions for turbulent latent heat flux.

Line 44-46: Li et al. (2019) have modeled the glacier mass balance and energy balance in the western Kunlun Mountains (not in the Pamir). And the similar questions in some other sites also need to check.

Section 3.2: How do you discern snow or bare-ground in the night? This is related to calculating turbulent heat fluxes at the night.

Section 4.5: The title of this section needs to be modified because this section explains the relationship between climate and turbulent heat fluxes and cloud cover is not the key factor impacting the changes in LE.

The manuscript is not concise and too wordy. There are some similar contents in Section 4.5, 4.6 and 5.1.

Line 456-457: Do you mean overcast conditions cause the neutral stability of the

surface boundary layer? Please explain it. Similar phenomena occurred in sections 4.4, 4.5, 4.7, and 5.1. The author could explain how cloud cover impacts climate conditions in the main text.

Line 474-475: Please explain this sentence.

Line 531-532: Cloud cover, on the other hand, has a significant impact on the primary meteorological variables, particularly Sin, Ts and qs. Several sentences can be added to simply explain the cause for this point.

Line 691: Muztag Ata No.1 is changed as Muztag Ata No.15.

Line 705-706: Sublimation rate during the June-September, in general, was lower than that of October-May. This also occurs in the westerlies region (such as Muztag Ata No.15 Glacier) and the area of transition between the westerlies- and monsoon-dominated climate regimes (such as Xiao Anglong Glacier). However, the ratio of June-September sublimation to October-May sublimation is larger in the monsoon region than that in the westerlies and the transition area (Zhu et al. 2020).

---

## Referee Report (RR2)

The authors have done an excellent job in addressing the concerns raised by reviewers. I have a few minor issues to be addressed below before publication.

L52: '…applications of SEB remain rare to date in the …'

L63: maybe write 'fluxes' instead of 'flux'

L68: '…observed to be up to 66% …'

L93: 'Special attention is given to …'

L98: Like with Glacier/Basin etc 'Valley' should also be capitalized if referring to a specific one.

L111: Why was 'were' replaced by 'was'? Data is always plural! Same elsewhere (e.g. l166).

L138: 'less than'

L181: '(positive)' but actually I think you can leave the whole bracket away

L318: 'are shown' – maybe give a good read through once on where you should have plural and were singular, this occurs quite often.

L550: Remove 'whereas' – better not to start sentences with it.

L667: 'This could …' I do not quite understand what you want to say here? 'This could possibly lead to a lower energy sink through the LE flux, which will …'?

Figure 12: I know this is always a bit challenging, but it would be good if the legend does not overlay any data visualization in the top left panel.

L705: 'rates'

L732: 'This is a big issue …' Again I don't get what that sentence is supposed to say. What is the 'issue' here? Honestly I think that sentence is not required.

L809: 'schemes'

---

## Author Response (AR2)

Below we have provided our responses **(in red text)** point-by-point to each comment from the reviewers **(in black text)**. *Italic texts* are used to highlight specific changes in the updated manuscript.

**Response to Reviewer 1**

**General comments:**

I appreciate that the authors made a big revision to the manuscript and gave their kindly responses to my comments. The paper is much improved, although some of the limitations and concerns raised in the reviews remain. Below, I have listed a few minor points that could still be addressed but I believe that the paper is now basically suitable for publication.

We sincerely thank Reviewer 1 for the comments and suggestion in our revised manuscript. We have addressed nearly all the concerns raised by Reviewer 1 and made necessary changes in the revised manuscript. We would like to invite to the revised sections of the manuscript.

**Minor comments:**

Line 20: How do you get that cloud cover also restricts the turbulent heat fluxes by around 50% in this work? I do not find such data (50%) in the main text. The author could add some sentences to explain why cloud cover restricts the turbulent heat fluxes.

This has been discussed in Sect. 4.4. In the previous manuscript, we didn't mention this in terms of '%' reduction. In the revised manuscript we have added a few lines mentioning turbulent heat flux reduction in '%' and a brief explanation for that. Revised and new sentences are shown below:

Line No. 377-387 (in the revised manuscript; track-changed version)

*'Turbulent heat fluxes were generally higher in clear-sky conditions due to higher instability of the surface boundary layer (Fig. 9). In clear-sky, the mean daytime H was -66 W m$^{-2}$ which is three times more negative compared to overcast conditions (-21 W m$^{-2}$), corresponding to 68% reduction of H in overcast conditions than in clear-sky. Similarly, the mean daytime LE was also higher in clear-sky, with -136 W m$^{-2}$ compared to -47 W m$^{-2}$ in overcast conditions (65% reduction). The reduced magnitude of turbulent heat fluxes in overcast/cloudy conditions was due to the neutral stability of the surface boundary layer (Fig. 9B; $R_{ib} \approx 0$). In neutral stability conditions, cold temperature ($T_{air} - T_s$ close to 0) restricts the magnitude of H and LE (Fig. 9). In clear-sky conditions, more negative LE was due to the surface's intense heating ($T_{air} - T_s < 0°C$), which creates a stronger vertical moisture gradient ($q - q_s$) than overcast conditions. $F_{surface}$ showed a slight daytime variation during clear-sky, but no significant variation in overcast conditions.'*

Line 22-24: Please add the interpretation that why a strong control of cloud cover in shaping favorable conditions for turbulent latent heat flux.

The interpretation is presented in detail in Sect 5.1. The texts are shown below:

Line No. 508-512

*'For example, cloud cover shapes the prevailing weather conditions at the study site by influencing the stability of the surface boundary layer (Fig. 9). In a stable stratification ($T_{air}$ - $T_s$ > 0°C), the snow surface remains cooler than the air, which attributes to a gentle near-surface moisture gradient and a lower LE, whereas in an unstable stratification ($T_{air}$ - $T_s$ < 0°C), steep near-surface moisture gradient results in a high negative LE.'*

Also, to mention this in the abstract briefly, we revised the respective sentence (in the abstract) as:

Line No. 22-23

*'Sublimation rates were three times higher in clear-sky conditions than overcast, indicating a strong role of cloud cover in shaping favourable conditions for turbulent latent heat flux by modulating the near-surface boundary layer conditions.'*

Line 44-46: Li et al. (2019) have modeled the glacier mass balance and energy balance in the western Kunlun Mountains (not in the Pamir). And the similar questions in some other sites also need to check.

Thanks for pointing this out. We have revised the sentence and shown below:

Line No. 43-46

*'Glaciers in the Pamir and Kunlun Mountains are extreme continental type, with cold temperature and low annual precipitation (Zhu et al., 2020; Li et al., 2019), thus their SEB characteristics are expected to behave differently than majority of HK glaciers which are alpine type, with relatively higher precipitation and temperature.'*

Section 3.2: How do you discern snow or bare-ground in the night? This is related to calculating turbulent heat fluxes at the night.

We simply assumed snow-cover at night if the snow-cover existed during the daytime considering no $S_{in}$ in the night and significantly cooler temperature. To properly handle this, we discarded data from the entire day when a single half-hourly albedo value was lower than the snow-cover albedo threshold (0.4).

Moreover, the night-time turbulent heat fluxes were significantly lower (below ± 20 W m$^{-2}$; Fig. 6 in the revised manuscript) compared to daytime values (close to 100 W m$^{-2}$). Considering the low/negligible values of turbulent heat fluxes at night and possible uncertainty in night-time albedo, we restricted our analysis to the daytime hours only (09:00-16:00 Indian Standard Time).

The snow-cover filtration method based on albedo threshold is discussed in Sect. 3.1. Below we copied a few lines from the respective section:

Line No. 133-136

*'We filtered the snow-covered period for SEB based on the daytime surface albedo threshold value above 0.4 at the AWS-M (the mean bare-ground/snow-free surface albedo was lesser than 0.25 for July-August; 2009-2020). Additionally, we discarded the data of 74 days (2975 data points) out of a total of 1664 days (76248 data points; DJFMA 2009-2020) when albedo was below 0.4 (refer to Table S2 for snow-free dates).'*

Section 4.5: The title of this section needs to be modified because this section explains the relationship between climate and turbulent heat fluxes and cloud cover is not the key factor impacting the changes in LE.

We modified the title of the respective section (Sect. 4.5) to **"Relationship of turbulent heat fluxes and meteorological variables under different cloud conditions"**.

We kept 'under different cloud conditions' considering that the section discusses the relationship of turbulent fluxes for both clear-sky and overcast conditions.

The manuscript is not concise and too wordy. There are some similar contents in Section 4.5, 4.6 and 5.1.

We removed some of the similar explanations/sentences from the respective sections. We invite Reviewer 1 to go through the revised sections.

Line 456-457: Do you mean overcast conditions cause the neutral stability of the surface boundary layer? Please explain it.

Yes. In overcast conditions, due to significantly lower $S_{in}$, temperature (both $T_{air}$ and $T_s$) remains lower, with stronger wind (Fig. 9B). Due to cold temperature, vertical temperature difference ($T_{air} - T_s$) remains lower close to zero which results in $R_{ib} \approx 0$ (near-neutral stability; Fig. 9B). This phenomenon is also visible in Fig. 11A and 11D, where in overcast conditions (blue dots), $T_{air} - T_s$ values lie close to zero corresponding to a reduced magnitude of turbulent heat fluxes. To clearly show this phenomenon we have added $T_s$ in Fig. 9 (revised plot shown below).

Considering your 1ˢᵗ comment and this one, we have revised the respective Sect. 4.4 and added a few lines explaining how overcast conditions cause the neutral stability and impacts turbulent heat fluxes. Kindly refer to the 1ˢᵗ comment above. Below we have showed the revised and newly added lines:

Line No. 371-372

*'Due to comparatively lower temperature (both $T_{air}$ and $T_s$) and higher u in overcast conditions, the surface boundary layer remains near-neutral ($R_{ib}$ close to 0 due to low vertical temperature difference; $T_{air}$ - $T_s$).'*

Line No. 380-385

*'The reduced magnitude of turbulent heat fluxes in overcast/cloudy conditions was due to the neutral stability of the surface boundary layer (Fig. 9B; $R_{ib}$ values close to 0). In neutral stability conditions, cold temperature ($T_{air} – T_s$ close to 0) restricts the magnitude of H and LE (Fig. 9).'*

[Figure]

**Fig. 9 (Revised):** Daytime (09:00-16:00 IST) diurnal cycle of $T_{air}$, $T_s$, $u$, $R_{ib}$ and SEB components under the clear-sky ($CF \leq 0.2$) and overcast ($CF \geq 0.8$) conditions.

Similar phenomena occurred in sections 4.4, 4.5, 4.7, and 5.1. The author could explain how cloud cover impacts climate conditions in the main text.

Kindly refer to the previous comment and the 1st comment where we have explained this aspect and mentioned the necessary changes we have made in the revised manuscript.

Line 474-475: Please explain this sentence.

We have revised the sentence for a clearer explanation.

Line No. 397-400

*'Similarly, LE was strongly correlated with $T_{air}$ - $T_s$ in clear-sky (r = 0.84; p < 0.001) but moderately correlated in overcast conditions (r = 0.50; p < 0.001), which suggests that the vertical temperature difference significantly controls the near-surface vertical moisture gradient (one of the primary drivers of LE). This attributes to a significantly higher negative LE in clear-sky than in overcast conditions (Fig. 11).'*

Line 531-532: Cloud cover, on the other hand, has a significant impact on the primary meteorological variables, particularly Sin, Ts and qs. Several sentences can be added to simply explain the cause for this point.

Thank you for the suggestion. Here, we refer the readers to the section (Fig. 9; Sect 4.4) where we explicitly discussed the impact of cloud cover in primary meteorological variables and SEB components. We could have added a few sentences here, however, such explanation has already been discussed in Sect. 4.4, with sole focus on cloud cover impact on daytime meteorology and SEB components. Therefore, considering the potential repetition/similar texts, we avoided to add new sentences here. The revised sentence shown below:

Line No. 449-451

*'Cloud cover, on the other hand, has a significant impact on the primary meteorological variables, particularly $S_{in}$, $T_s$ and $q_s$ (Fig. 9; Sect. 4.4).'*

Line 691: Muztag Ata No.1 is changed as Muztag Ata No.15.

Corrected. Thanks.

Line 705-706: Sublimation rate during the June-September, in general, was lower than that of October-May. This also occurs in the westerlies region (such as Muztag Ata No.15 Glacier) and the area of transition between the westerlies- and monsoon-dominated climate regimes (such as Xiao Anglong Glacier). However, the ratio of June-September sublimation to October-May sublimation is larger in the monsoon region than that in the westerlies and the transition area (Zhu et al. 2020).

Thanks for the point. We have included the suggested text almost as it is in the discussion.

Line No. 604-607

*'This also occurs in the westerlies dominated region such as Muztag Ata No. 15 Glacier in Pamir (Zhu et al., 2018) and the area of transition between the westerlies- and monsoon-dominated climate regimes such as Xiao Anglong Glacier in Upper Shiquanhe region (Zhu et al., 2021b).'*

Line No. 611-613

*'The ratio of summer (June-September) sublimation to winter (October-May) is larger in the monsoon-dominated region such as Parlung No. 4 and Zhadang glaciers than that in the westerlies and the transition area, e.g., Xiao Anglong Glacier (Zhu et al. 2020).'*

**Response to Reviewer 2 (Jakob Steiner)**

The authors have done an excellent job in addressing the concerns raised by reviewers. I have a few minor issues to be addressed below before publication.

We would like to thank Jakob Steiner for his comments and suggestion in our revised manuscript. We have addressed all the concerns pointed by him and made necessary changes in the revised manuscript.

L52: '…applications of SEB remain rare to date in the …'

Revised as suggested.

L63: maybe write 'fluxes' instead of 'flux'

Done.

L68: '…observed to be up to 66% …'

Revised as suggested.

L93: 'Special attention is given to …'

Revised as suggested.

L98: Like with Glacier/Basin etc 'Valley' should also be capitalized if referring to a specific one.

Thanks, Done.

L111: Why was 'were' replaced by 'was'? Data is always plural! Same elsewhere (e.g. l166).

We have corrected it everywhere in the manuscript.

L138: 'less than'

Revised as suggested.

L181: '(positive)' but actually I think you can leave the whole bracket away

Thanks. Removed the text within brackets.

L318: 'are shown' – maybe give a good read through once on where you should have plural and were singular, this occurs quite often.

Thanks for catching this. We have corrected it across the manuscript.

L550: Remove 'whereas' – better not to start sentences with it.

We revised it as suggested. Now the sentence reads as:

*'Sublimation was considerably lower when moisture availability was higher, $T_s$ was significantly lower, with very strong u (Fig. 12; Fig. 13).'*

L667: 'This could …' I do not quite understand what you want to say here? 'This could possibly lead to a lower energy sink through the LE flux, which will …'?

Thanks. We have revised the sentence as suggested:

*'This could possibly lead to a lower energy sink through the LE flux, which will boost the efficiency of $S_{in}/R_{net}$ resulting in more surface melt.'*

Figure 12: I know this is always a bit challenging, but it would be good if the legend does not overlay any data visualization in the top left panel.

Thanks. We have revised the figure as suggested and shown below:

[Figure]

L705: 'rates'

Done.

L732: 'This is a big issue …' Again I don't get what that sentence is supposed to say. What is the 'issue' here? Honestly I think that sentence is not required.

We removed the sentence as suggested.

L809: 'schemes'

Done.